

# On the mechanisms of warming the mid-Pliocene and the inference of a hierarchy of climate sensitivities with relevance to the understanding of climate futures

Deepak Chandan[1] and W. Richard Peltier[1]

[1]Department of Physics, University of Toronto, 60 St. George Street, Toronto, Ontario, M5S1A7

*Correspondence to:* Deepak Chandan (dchandan@atmosp.physics.utoronto.ca)

**Abstract.** We present results from our investigation into the physical mechanisms through which the mid-Pliocene, with an atmospheric $p\mathrm{CO_2}$ of only ∼400 ppmv, could have supported the same magnitude of global warmth as that which has been projected for the climate at the end of the 21$^{\text{st}}$ century when $p\mathrm{CO_2}$ is expected to be three times higher. These mechanisms explore changes to the radiative properties of the surface, the clouds, greenhouse gases and changes to the meridional heat transport. Furthermore, we provide a mid-Pliocene perspective on ongoing efforts to understand the climate system's sensitivity at various timescales and using multiple lines of evidence. The similarities in the boundary conditions between the mid-Pliocene and the present day, together with the globally elevated temperatures, make the mid-Pliocene an ideal palaeo time period from which to derive inferences of climate sensitivity and assess the impacts of various timescale-dependent feedback processes. We assess a hierarchy of climate sensitivities of increasing complexity in order to explore the response of the climate over a very large range of timescales. The various sensitivities that we calculate provide insight on not only how the climate responds to a given forcing over a short timescale, but also on intermediate and very-long timescales. The latter category includes the impact of the feedback from the glacial isostatic adjustment of the Earth's surface in response to the melting of the polar ice sheets. Our inference of the intermediate timescale climate sensitivity suggests that the projected warming by 2300 CE, inferred using Earth System Models of Intermediate Complexity on the basis of an extension to the RCP4.5 emission scenario in which atmospheric $p\mathrm{CO_2}$ stabilizes at roughly twice the PI level in year 2150 CE, could be underestimated by ∼ 1 °C due to the absence of ice sheet based feedbacks.

## 1 Introduction

The most recent assessment report by the Intergovernmental Panel on Climate Change (IPCC, 2013) expresses a high level of confidence that the global surface temperature at the end of the 21$^{\text{st}}$ century would be greater than 2 °C warmer than the 1850–1900 reference level in the two most extreme warming scenarios: RCP6 and RCP8.5. These levels of warming are driven by an increase in the radiative forcing to $6\,\mathrm{Wm^{-2}}$ in RCP6 (Fujino et al., 2006; Hijioka et al., 2008) and $8.5\,\mathrm{Wm^{-2}}$ in RCP8.5 (Riahi et al., 2007, 2011), corresponding to atmospheric carbon dioxide partial-pressures of ($p\mathrm{CO_2}$) ∼ 1370 ppmv and ∼ 850 ppmv respectively.





The mid-Pliocene time period ($\sim 3.3$ Mya) is also expected to have been roughly 2–3 °C warmer than the prein-dustrial (PI; Haywood and Valdes, 2004; Haywood et al., 2013; Kamae et al., 2016), and up to $3.8$ °C warmer (Chandan and Peltier, 2017) using the latest reconstruction for the mid-Pliocene boundary conditions. Atmosphere only modelling driven with proxy inferred sea surface temperatures also yield a mid-Pliocene that is 2–3 °C warmer than the PI (Haywood et al., 2013). These levels of warming are nearly identical to those forecasted with the RCP6 and RCP8.5 scenarios, yet, in contrast to those scenarios the mid-Pliocene warming took place under substantially less radiative forcing. Proxy inferences for $pCO_2$ yield values in the range of 300–400 ppmv (Seki et al., 2010; Martínez-Botí et al., 2015; Pagani et al., 2010; Badger et al., 2013), with some estimates pushing the lower bound to 200 ppmv (Bartoli et al., 2011; Tripati et al., 2009). A $pCO_2$ of 400 ppmv is typically used in coordinated efforts for the modelling of the mid-Pliocene, such as in the Pliocene Model Intercomparison Project phase 1 (PlioMIP; Haywood et al., 2011) and phase 2 (PlioMIP2; Haywood et al., 2016). See Chandan and Peltier (2017) for a recent review of proxy based inferences of various climatological attributes of the mid-Pliocene.

There have been other hothouses during the recent Cenozoic, but the mid-Pliocene stands out among them as a unique 'natural analogue' for the future, warmer climate. The recent interglacials were warmer than the mid-Pliocene, but they were ephemeral, and the climate quickly returned to a glacial state. In contrast, the mid-Pliocene was a much longer period of sustained warmth. More importantly, with regards to the future climate, the primary forcing during the interglacials was the increase in northern hemisphere summer insolation whereas the primary forcing during the mid-Pliocene was the elevated $pCO_2$. The mid-Pliocene also has the advantage of having had a continental configuration nearly identical to the present. As one travels further back in time to other hothouse worlds such as the middle-Miocene climate optimum, the Oligocene warming or the mid-Eocene climate optimum, the surface boundary conditions begin to increasingly depart from the present day.

Although the mid-Pliocene was in a state of quasi-equilibrium, its pervasive warmth was set against the backdrop of a world that was undergoing a profound transformation which began in the mid-Eocene and ended with the onset of the Pleistocene glacial-interglacial cycles $\sim 1$ Mya. During this period, the global mean surface temperature was dropping (Hansen et al., 2013; Zachos et al., 2008), the atmospheric $pCO_2$ was declining (Zhang et al., 2013), ice sheets were beginning to form and grow in size in the polar regions (Zachos et al., 2001) and as a result the global mean sea level was falling (Haq et al., 1987; Hansen et al., 2013). Therefore, in Earth's history, the climate advanced from a hotter state to the (relatively cooler) mid-Pliocene, i.e. the climate evolved to the mid-Pliocene on a *cooling trajectory*.

But, for the purposes of understanding the future climate, we are actually interested in the transition from the present to a warmer climate. Therefore, it would appear to be worthwhile to examine this late Cenozoic transition in reverse. If we applied the changes known to have occurred in the climate system over the past 3 Myr in reverse using an appropriate climate model, we would reasonably expect to recover a 'mid-Pliocene-like' state. We would only have to reverse the $pCO_2$ history since the surface temperature increase, the decrease in the polar ice sheets and the rise in the sea level are all just feedbacks, among several others, operating on various timescales. The resulting





quasi-equilibrium, mid-Pliocene-like state would be a function of the forcing and the internal feedbacks that are active on this *warming trajectory*. Therefore, this final state will invariably contain some information (an imprint) regarding the nature of the feedbacks and the climate system's response to those feedbacks, and response to the forcing itself.

The feedbacks that would operate in this expository scenario would also operate during the contemporary warming that is being driven by an increase in atmospheric $pCO_2$. Some of these feedbacks are already well underway, and some will 'come online' at different timescales in the future (see Sect. 4.4). If it were possible to extract some insight into the sensitivities of the Earth system to these feedbacks on the basis of the system's *warm approach to the mid-Pliocene*, then we could provide an independent constraint to the ongoing efforts to quantify, using a diversity

of methods, the multiple sensitivities that characterize the potential of the climate to warm in the future. We build upon the work of PALAEOSENS Project Members (2012) and show how we can deduce these sensitivities using the suite of PlioMIP2 simulations. We therefore draw attention to the fact that the significance of the mid-Pliocene for understanding the future of the climate rests not only in its 'warm state' persona but also in the fact that as a climate simulation approaches a mid-Pliocene-like state on a warming trajectory, the simulation will evolve under the

influence of the same mechanisms, forcings and feedbacks that the future of our climate will evolve. Provided, the resulting climate is a faithful representation of the mid-Pliocene, which can be assessed on the basis of data-model comparison, we can have confidence in the integrity of these results.

    Therefore, in several ways the mid-Pliocene is in an arguably privileged position to inform us about the warm climate of the future. In recognition of this advantage, in this paper we seek to conduct two fundamental analyses

into the mid-Pliocene climate: first, we want to gain a deeper understanding into the origin of the warming within the mid-Pliocene simulation of Chandan and Peltier (2017) that was driven using the latest inference of the mid-Pliocene boundary conditions. It was found that with these new boundary conditions, it becomes possible for a climate model to, for the first time, reasonably accurately capture features of the proxy-inferred enhanced warming in the high-latitudes during the mid-Pliocene. Therefore, understanding why these results were able to do such a good job

is manifestly of interest to understanding the mid-Pliocene climate and it can also guide strategies to refine boundary conditions so as to further reduce data-model disagreement. Furthermore, considering the similarities between the mid-Pliocene warmth and the forecasted warming discussed above, the insights gained from this investigation offer an independent constraint about where, for what reasons, and of what magnitude is warming to be expected in the future. We have designed our investigation into this topic around three key questions:

1. What are the impacts of the differences in boundary conditions on the mid-Pliocene warming (Sect. 4.2)

    2. What are the physical mechanisms through which the mid-Pliocene warms? (Sect. 4.3)

    3. How do the modifications to the boundary conditions themselves change the effectiveness of those physical mechanisms? (Sect. 4.3)

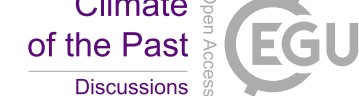

The second theme of this paper is the question of the climate system's sensitivity to various forcings and feedbacks. As mentioned earlier, if we were to reverse the late Cenozoic decrease in $pCO_2$ within a climate model then we should be able to reverse the climate changes and reproduce a mid-Pliocene-like climate. In practice, we do not need to perform a 3 Myr integration with continuously evolving forcings and boundary conditions to be able to reproduce the

mid-Pliocene; we simply need to perform a sufficiently accurate simulation which starts from the PI and is forced by mid-Pliocene boundary conditions. Provided that the simulation is allowed sufficient time to reach quasi-equilibrium, this accelerated warm approach to the mid-Pliocene would be a reasonable approximation to the full reversal of the warming.

   Several, long-term feedbacks communicate with the climate system through changes to the system's boundary
conditions. Unfortunately, at present we are limited by the technological capabilities of our climate models to effortlessly incorporate changing boundary conditions, and therefore our simulations are forced with stationary boundary conditions. Nevertheless, the impacts of the feedbacks will be implicitly communicated to the system through the boundary conditions and the system will have to adjust to these feedbacks in its evolution towards the quasi-equilibrium state. Naturally, the quality of this effort can only be as good as the quality of the simulated

mid-Pliocene; because, if we are not able to simulate a reasonable mid-Pliocene then what possible grounds could we have to argue that the inferences of any sensitivity parameters are grounded in some reality? In this regard we are well positioned, for, as mentioned earlier our mid-Pliocene simulation is in very good agreement with proxy reconstructions of the climate of that time (Chandan and Peltier, 2017).

   This paper is organized as follows: in Sect. 2 we first introduce our numerical model, and describe the setup and the
process of initialization of our numerical experiments. In Sect. 3 we provide detailed introduction to and theoretical formulation of the analytical methods extensively employed in the interpretation of our simulation outcomes. The results are discussed in Sect. 4 and the conclusions of this study in Sect. 5.

## 2   Numerical Model and Experimental Design

### 2.1   The UofT version of CCSM4

The Coupled Climate System Model version 4 (CCSM4; Gent et al. (2011)) is a numerical climate model that is comprised of four major sub-models – the atmosphere model, which is the Community Atmosphere Model version 4 (CAM4; Neale et al. (2013)), the ocean model, which is the Parallel Ocean Program, version 2 (POP2; Smith et al. (2010)), the land model, which is the Community Land Model version 4 (CLM4; Lawrence et al. (2012)), and the sea ice model, which is the Community Ice Code version 4 (CICE4; Hunke and Lipscomb (2008)). These four

components interact in a fully coupled manner with one another through a flux coupler without flux adjustment. The reader is referred to Chandan and Peltier (2017) for a summary of the improvements of these components over those that formed the basis of the previous CCSM3 model (Collins et al., 2006).





In Chandan and Peltier (2017) we introduced, and employed a configuration of CCSM4 which was modified slightly from its default configuration in order to remove the impacts of two ocean model parameterizations which we argued to be highly tuned to modern day conditions and therefore whose application to simulating palaeo-oceans is questionable (see also Griffiths and Peltier (2008, 2009); Peltier and Vettoretti (2014)). This configuration —
which we referred to as the 'University of Toronto version of CCSM4' (UofT-CCSM4) to distinguish it from its default configuration (NCAR CCSM4) — has both the overflow parameterization and the tidal-mixing scheme disabled and the vertical profile of diapycnal diffusivity fixed to that used in the ocean component (POP1) of the CCSM3 model. Additionally, in order to be able to compare our mid-Pliocene simulations to the control simulations without any ambiguity associated with these changes, we also ran our pre-industrial (PI) and modern day controls using the
same model configuration. The new simulations presented in this paper are also run using this University of Toronto configuration.

## 2.2  Numerical Experiments

The analyses presented in this paper involve several experiments which have been proposed for the PlioMIP2 program (see Haywood et al., 2016, Table 3). These simulations are referred to using the nomenclature first employed by
Lunt et al. (2012b) and subsequently adopted with modifications for PlioMIP2 by Haywood et al. (2016). In this notation simulations are referred to by the form $Ex^c$ where $c$ is the concentration of atmospheric $CO_2$ and $x$ represents boundary conditions that have been changed from the PI such that $x$ can be absent (for the case in which no boundary conditions have been modified) or it can be either or both 'o' for a change of orography and 'i' for a change of ice sheets (Table 1.) The PlioMIP2 simulations can be categorized into control simulations, mid-Pliocene
simulations, and sensitivity simulations.

### 2.2.1  Control Simulations

Two PlioMIP2 experiments: the Core simulation $E^{280}$ and the Tier 2 simulation $E^{400}$ constitute our set of control simulations. These simulations were introduced in Chandan and Peltier (2017) where they were referred to as $E^{280}{}_P$ and $E^{400}{}_P$ respectively. Both simulations are configured with modern-day topography, bathymetry, land-sea
mask (LSM), vegetation and ice sheets. The atmospheric $CO_2$ concentration in the former simulation is set to the PI value of 280 ppmv and as such we will refer to that simulation as the 'PI control,' whereas the latter simulation has a modern-day like $CO_2$ concentration of 400 ppmv and we will therefore refer to it as the 'modern control.' This simulation is 'modern' only insofar as the $CO_2$ concentration is concerned; it doesn't have modern concentrations of other trace greenhouse gases, or modern land-units such as urban areas or agricultural land-units.



### 2.2.2 Mid-Pliocene Simulations

In this paper by 'mid-Pliocene simulation' we mean a simulation that has full mid-Pliocene surface boundary conditions i.e. the E$oi$ case with mid-Pliocene orography, vegetation, LSM and ice sheets, and has an atmospheric $CO_2$ concentration in the range of most mid-Pliocene reconstructions (350 – 450 ppmv). Therefore, we do not refer

to the PlioMIP2 E$oi^{280}$ simulation which has mid-Pliocene surface boundary conditions but 280 ppmv atmospheric $pCO_2$ concentration as a mid-Pliocene simulation and instead refer to it as a 'sensitivity simulation' (next section) designed to explore the response of the climate to PI-like low atmospheric $pCO_2$ under mid-Pliocene surface boundary conditions. The boundary conditions we use are from the latest mid-Pliocene reconstruction from the PRISM group — PRISM4 (Dowsett et al., 2016), and which has been adopted as the mandatory boundary conditions set for the

PlioMIP2 program. We have simulated two mid-Pliocene simulations, the Core simulation E$oi^{400}$ and the Tier 1 simulation E$oi^{450}$, both of which were described in Chandan and Peltier (2017) wherein they were referred to as E$oi^{400}$P and E$oi^{450}$P respectively.

### 2.2.3 Sensitivity Simulations

We refer to those PlioMIP2 simulations which are neither entirely mid-Pliocene, nor entirely present-day (or PI) as

sensitivity simulations. These have been proposed to investigate the impact on the climate system from changes to each of the boundary conditions that differentiates the mid-Pliocene from the PI. There are three configurations that constitute this set of sensitivity experiments: namely E$i$, E$o$ and E$oi$. The E$o$ and E$i$ sensitivity experiments have two variants with 280 and 400 ppmv atmospheric $pCO_2$, while the E$oi$ sensitivity experiment has 280 ppmv atmospheric $pCO_2$ (as discussed above, we categorize the other E$oi$ variants with higher $CO_2$ concentrations as mid-Pliocene

simulations). We therefore report on five new PlioMIP2 simulations in this paper: E$i^{280}$, E$i^{400}$, E$o^{280}$, E$o^{400}$ and E$oi^{280}$.

Figure 1 shows the topography for each of these three configurations as-well-as the orographic anomaly of these configurations with respect to the PlioMIP2 modern orography. These anomalies are added to the local-modern orography to produce the atmospheric boundary conditions for each configuration in accordance with the anomaly

method that we have previously employed (Chandan and Peltier, 2017). In the E$i$ configuration only those regions covered by present day ice sheets are modified to yield the distribution of the mid-Pliocene ice sheets. This change is clearly seen in the orography anomaly for this configuration (Fig. 1d) in which large reductions in orography over most of Greenland, West Antarctica and the sub-glacial basins of East Antarctica are the result of the reduced mid-Pliocene ice cover. The mid-Pliocene Greenland ice sheet reconstruction in PlioMIP2 is based on the modeling

results from the PLISMIP project (Dolan et al., 2012). The configuration of the Antarctic ice sheets is provided by the PRISM3D (Hill et al., 2007; Dowsett et al., 2010) reconstruction that was also used in the previous phase of PlioMIP (Haywood et al., 2013). In the PRISM3D reconstruction there is some thickening of the ice sheet over the interior of East Antarctica compared to the present day.



For this configuration a complication arises over West Antarctica where the marine grounded ice sheets inextricably link the local ice sheet distribution with the local LSM. In this case replacing the present-day ice configuration with the mid-Pliocene configuration would also result in unwanted changes to the LSM in the region (such as the changes seen in the $Eoi$ configuration). Therefore, in order to maintain the LSM integrity in the ice sheet sensitivity

experiments, we have replaced all regions where there is ice in the present day but not in the mid-Pliocene with very low orography (Fig. 1a). Lastly, in this configuration all other boundary conditions, including vegetation and river routing in the land model, are kept identical to present day.

In the $Eo$ configuration an anomaly in the orography exists at all locations where there is mid-Pliocene land, except at the locations of present-day ice sheets (Fig. 1b,e). In addition, anomalies also exist over parts of West

Antarctica where the replacement of present-day marine grounded ice sheets by ocean has led to large negative values of the anomaly. The presence of marine grounded ice sheets in Antarctica again raise complications; in this case, changing the mid-Pliocene LSM would unavoidably alter the local ice distribution, whereas the aim of this configuration is to leave the ice sheets unchanged from the present day. Therefore the integrity of present day ice sheets cannot be entirely maintained for this configuration. In addition to the changes to the orography and the

LSM, this configuration also implements full mid-Pliocene bathymetry and consequently, the ocean model sees the same mid-Pliocene bathymetry as that which was implemented in the mid-Pliocene simulations of Chandan and Peltier (2017). The vegetation distribution and the river routing schemes are also changed to that in the mid-Pliocene simulations.

The methodology for implementing the $Ei$ and the $Eo$ boundary conditions into the UofT-CCSM4 model is identical

to that employed in Chandan and Peltier (2017) for the implementation of the full-Pliocene ($Eoi$) configuration and the reader is referred to that publication for the details of this process. The $Eoi$ configuration includes the complete set of changes made to all the surface boundary conditions. The topography, and the orographic anomaly with respect to PlioMIP2 modern conditions are shown in Fig. 1c,f.

### 2.2.4 Model Initialization

Simulating five sensitivity experiments in addition to the control and the mid-Pliocene experiments already reported in Chandan and Peltier (2017) to near statistical equilibrium within a coupled-climate model at $1°$ resolution requires the expenditure of a significant amount of computational resources. It was therefore necessary to carefully consider the process through which we initialized the five sensitivity experiments so as to ensure that the requirements for computational resources could be kept reasonable. Because the ocean component of the climate takes the longest

time to equilibrate, we decided to initialize the sensitivity experiments, wherever possible, from an intermediate state obtained during the evolution of one of the PlioMIP2 simulations of Chandan and Peltier (2017) whose ocean is expected to be close to the ocean in the sensitivity experiments. An intermediate state, as opposed to an equilibrated state was used to ensure that we do not incur overwhelming influence of the simulation from which the initial



conditions were derived while retaining the benefits of a faster spin-up. This initialization process is illustrated in Figure 2.

The two $Ei$ experiments were initialized from the two control simulations, as illustrated in Fig. 2. This is because it was hypothesized that the changes to the ice sheets would be limited in their ability to significantly modify the

state of the global ocean. On the other hand, the changes to the orography (and bathymetry) were expected to be more important as not only are these changes more pervasive, but also because the modified bathymetry would interact with the ocean circulation directly. For this reason, we expect that the ocean in the mid-Pliocene simulation would be close to the equilibrium state that the oceans would reach in the $Eo$ experiments. On the basis of the same arguments $Eoi^{280}$ could also have been branched from our 400 ppmv mid-Pliocene experiment. However, since

both the $Eoi$ mid-Pliocene experiments reported in Chandan and Peltier (2017) were initialized from modern day conditions (Levitus and Boyer, 1994), we decided to do the same for $Eoi^{280}$.

## 3   Theory

### 3.1   The Factorization Technique

There are four critical differences between the boundary conditions of the mid-Pliocene and the PI: changes to

orography, changes to ice sheets, change in the concentration of atmospheric $pCO_2$, and changes to ocean bathymetry. We will focus on the impacts of the first three of these differences. The impacts on the mid-Pliocene climate owing to the difference in bathymetry are expected to be important, particularly since a different bathymetry can alter the locations of deep water formation at high-latitudes, and modify the strength of the overturning circulation. However, we choose not to address this issue directly here not only because no sensitivity experiment for the ocean

bathymetry is specified in PlioMIP2, but also because in using the factorization methodology discussed below, the treatment of ocean bathymetry as a separate factor would require eight additional experiments. Other differences between the mid-Pliocene and the modern day, which are mostly confined to the land and include changes to the soil type, vegetation distribution and the distribution of wetlands and lakes, are expected to contribute much less to the mid-Pliocene warmth (Lunt et al., 2012b).

To understand the impact upon some observable of the climate due to a perturbation made to a specific boundary condition, one can simply difference the value of that observable between two simulations which differ from one another only with regards to that particular perturbation. For instance, if one wants to study the effect on the SAT caused by changes made to the elevation of a certain mountain range, then studying the SAT difference between two simulations (one with the revised elevations, and one without) would suffice. However, the situation is clearly more

complex when several boundary conditions are modified in concert and one wishes to discriminate the impacts on the observable from each of the individual boundary condition changes. This situation is commonly encountered in palaeoclimatology as the past usually differs from the control by way of more than one boundary condition, such as the case between the mid-Pliocene and the PI.



When the impacts of two or more changes are sought, the same method can be extended with the help of additional simulations. However, this linearized approach would not give any information regarding the nonlinear interactions between the modified boundary condition. Stein and Alpert (1993) proposed a factorization method to separate the impacts of more than one 'factors' and quantify the magnitude of the nonlinear interactions between the factors, also

referred to as their 'synergy'. This method has seen extensive application in palaeoclimatology, such as for studying the effects of ocean-vegetation interaction (Wohlfahrt et al., 2004) and the effects of soil schemes (Stärz et al., 2016) on the climate of the mid-Holocene; the impacts of boundary conditions during the last interglacial (Loutre et al., 2014), contributions to the warming during the last interglacial (Crucifix and Loutre, 2002), and other interglacials over the past 800,000 years (Yin and Berger, 2012); vegetation dynamics during the last glacial maximum (LGM;

Claussen et al., 2013; Jahn et al., 2005) and the impacts of soil schemes on the LGM climate (Stärz et al., 2016); the contributions to the mid-Pliocene warming (Lunt et al., 2012b) and the strengths of the East Asian monsoons during the mid-Pliocene (Zhang et al., 2015).

Lunt et al. (2012b) argue that the formulation of Stein and Alpert (1993) suffers from the shortcoming that it leads to a non-unique solution. To overcome this limitation they propose a re-formulation of this technique in which

the synergy terms are partitioned evenly between the contributions of each factor so as to yield a symmetric and unique factorization. The principle drawback of factorization methods is that they require $2^N$ simulations, where $N$ is the number of factors that are being studied. For our case where three boundary condition are changed, this method requires a total of eight simulations. Due to the power dependence, adding another category, such as ocean bathymetry, would require eight additional simulations, whereas two additional categories (five in total) would

require 24 additional simulations. Clearly, this approach is only feasible for small values of $N$.

In the factorization technique to be employed in this paper, we seek a decomposition of the form:

$$\Delta \xi = d\xi_{CO_2} + d\xi_{orog} + d\xi_{ice} \tag{1}$$

where, $\Delta \xi$ is the total anomaly of some observable of the climate, $\xi$, between the control simulation and some other simulation of primary interest (in our case these are respectively the PI and the mid-Pliocene simulations).

The quantities $d\xi_{CO_2}$, $d\xi_{orog}$ and $d\xi_{ice}$, herein referred to as the 'factorized components/terms', are respectively the contributions to the total anomaly due to changes made to the atmospheric $p\text{CO}_2$, orography and ice sheets. The mathematical expressions for these terms are (Lunt et al., 2012b; Haywood et al., 2016):

$$
\begin{aligned}
d\xi_{CO_2} &= \left( \text{E}^{400} + \text{Eo}^{400} + \text{Ei}^{400} + \text{Eoi}^{400} - \text{E}^{280} - \text{Eo}^{280} - \text{Ei}^{280} - \text{Eoi}^{280} \right)/4 \\
d\xi_{orog} &= \left( \text{Eo}^{280} + \text{Eo}^{400} + \text{Eoi}^{280} + \text{Eoi}^{400} - \text{E}^{280} - \text{E}^{400} - \text{Ei}^{280} - \text{Ei}^{400} \right)/4 \\
d\xi_{ice} &= \left( \text{Ei}^{280} + \text{Ei}^{400} + \text{Eoi}^{280} + \text{Eoi}^{400} - \text{E}^{280} - \text{E}^{400} - \text{Eo}^{280} - \text{Eo}^{400} \right)/4
\end{aligned}
\tag{2}
$$

where the right hand side involving our experiment IDs should be read to imply the manner in which the observable $\xi$ is to be combined from those experiments.



## 3.2 One-dimensional Energy Balance Model

Here we discuss the 1-D EBM that we will use to explore several mechanisms through which the climate system adjusts its energy balance, and determine to what extent each of them is responsibe for the mid-Pliocene warmth? These mechanisms affect the zonal-mean energy balance by changing the Earth's radiative properties and its meridional

heat transport. This approach was formulated by Heinemann et al. (2009) and first applied to understanding the Paleocene/Eocene climate. Subsequently, it has been applied to the Eocene (Lunt et al., 2012a; Loptson et al., 2014), the middle-Miocene (Krapp and Jungclaus, 2011), the mid-Pliocene (Hill et al., 2014) and the snowball Earth (Voigt et al., 2011; Liu et al., 2013, 2017). We begin with a review of the model as formulated by Heinemann et al. (2009) and show how the impacts of these mechanisms may be estimated using standard outputs from any GCM. Then we

show how we can extend the 1-D EBM to further understand how the individual changes to the boundary conditions themselves impact each of these mechanisms.

The 1-D EBM (henceforth, just EBM) is written as:

$$SW_{TOA}^{\downarrow}(\phi)[1 - \alpha(\phi)] - H(\phi) = \epsilon(\phi)\sigma\tau^4(\phi), \tag{3}$$

where, $\phi$ is the latitude, $\sigma$ is the Stefan-Boltzmann constant and $SW_{TOA}^{\downarrow}(\phi)$, $\alpha(\phi)$, $H(\phi)$, $\epsilon(\phi)$ and $\tau(\phi)$ are respec-

tively the zonal-mean downwelling shortwave flux at the top of the atmosphere (TOA), the zonal-mean planetary albedo, the divergence of the zonal-mean meridional heat transport, the zonal-mean planetary emissivity and the zonal-mean surface temperature. Henceforth, all the physical quantities are to be understood as zonal means and therefore for readibility we stop using the adjective 'zonal-mean' and drop the explicit dependence of these quantities on $\phi$. The planetary albedo and the planetary emissivity can be easily calculated from the radiative fluxes computed

within the atmospheric component of the GCM, namely as:

$$\alpha = \frac{SW_{TOA}^{\uparrow}}{SW_{TOA}^{\downarrow}} \quad \text{and} \quad \epsilon = \frac{LW_{TOA}^{\uparrow}}{LW_{surf}^{\uparrow}}, \tag{4}$$

where $SW_{TOA}^{\uparrow}$, $LW_{TOA}^{\uparrow}$ and $LW_{surf}^{\uparrow}$ are the shortwave upwelling flux at the TOA, longwave upwelling flux at the TOA and longwave upwelling flux at the surface of the Earth, respectively. The divergence of the meridional heat transport is similarly diagnosed as the sum of the net shortwave and the net longwave fluxes at the TOA:

$$H = SW_{TOA}^{net} + LW_{TOA}^{net}. \tag{5}$$

The surface temperature $\tau$ in the EBM is determined by the parameters $\alpha, \sigma$ and $H$, so that, by defining $\tau \equiv T(\alpha, \epsilon, H)$ the solution to the EBM as formulated in Eq. (3) becomes:

$$T(\alpha, \epsilon, H) = \left[\frac{1}{\epsilon\sigma}(SW_{TOA}^{\downarrow}(1 - \alpha) + H)\right]^{1/4}. \tag{6}$$





Now, the perturbations in surface temperature in response to a change in either $\alpha$, $\sigma$ or $H$ can be expressed as:

$$
\begin{aligned}
dT_{emiss} &= T(\alpha,\epsilon,H) - T(\alpha,\epsilon',H) \\
dT_{alb} &= T(\alpha,\epsilon,H) - T(\alpha',\epsilon,H) \\
dT_{H} &= T(\alpha,\epsilon,H) - T(\alpha,\epsilon,H'),
\end{aligned}
\tag{7}
$$

where the primes are used to designate modified values. If we assume that the changes in the parameters are small, then the total change in the temperature between the two simulations, $\Delta T = T(\alpha,\epsilon,H) - T(\alpha',\epsilon',H')$ can be approximated as:

$$
\Delta T \simeq dT_{emiss} + dT_{alb} + dT_{H}.
\tag{8}
$$

Equation (8) breaks down the total warming into contributions from three mechanisms: changes to the planetary

albedo, changes to the planetary emissivity and changes to the meridional heat transport. Since planetary albedo includes both the surface albedo and the cloud albedo we can further partition $dT_{alb}$ to split the effects of both by expressing it to first-order as:

$$
dT_{alb} = dT_{swc} + dT_{salb},
\tag{9}
$$

where $dT_{swc}$ is the temperature change induced due to changes in the shortwave forcing by the clouds and $dT_{salb}$ is

the temperature change induced due to changes to the surface albedo. The latter can be diagnosed from the GCM's clear-sky fluxes (denoted by subscript $cs$) as:

$$
dT_{salb} = T(\alpha_{cs},\epsilon_{cs},H_{cs}) - T(\alpha'_{cs},\epsilon_{cs},H_{cs}).
\tag{10}
$$

Similarly, the planetary emissivity includes cloud emissivity as-well-as the emissivity of the atmospheric greenhouse gases (including water vapour), and a change to either or both can contribute to a change in the planetary emissivity.

Thereby $dT_{emiss}$ can also be partitioned, to first-order, as:

$$
dT_{emiss} = dT_{gg} + dT_{lwc},
\tag{11}
$$

where, $dT_{gg}$ is the effect on temperature from a change in emissivity arising from a change in the concentration of atmospheric greenhouse gases and $dT_{lwc}$ is the effect on temperature due to a change in the longwave forcing by the clouds. Using GCM clear-sky fluxes, $dT_{gg}$ can be computed as

$$
dT_{gg} = T(\alpha_{cs},\epsilon_{cs},H_{cs}) - T(\alpha_{cs},\epsilon'_{cs},H_{cs}).
\tag{12}
$$

So far we have reviewed the EBM methodology as it was formulated in Heinemann et al. (2009) for application to the Eocene. Equations (3) and (6)–(12) allow us to quantify the contributions to the mid-Pliocene minus PI temperature anomaly from changes to the climate system's radiative parameters, concentration of atmospheric



greenhouse gases and the meridional heat flux. However, they do not help us understand which of the modifications made to the various boundary conditions has contributed to changes to the climate's physical regime. An important motivation behind the PlioMIP2 sensitivity simulations is to isolate the impact of the various changes to the boundary conditions on the climate and in particular the surface temperature. Quantifying how the boundary conditions changes themselves affect the mechanisms of warming is important to developing that understanding. We therefore need to be able to extend the EBM framework to satisfy this objective.

We achieve this by incorporating concepts from the factorization methodology to extend the Heinemann et al. (2009) formulation. We begin by using Eq. (1) to break each of the terms on the right hand side of Eq. (8) into contributions arising from changes to orography, $pCO_2$, and ice sheets as:

$$
\begin{aligned}
dT_{emiss} &= dT_{CO_2,emiss} + dT_{orog,emiss} + dT_{ice,emiss} \\
dT_{alb} &= dT_{CO_2,alb} + dT_{orog,alb} + dT_{ice,alb} \\
dT_H &= dT_{CO_2,H} + dT_{orog,H} + dT_{ice,H} .
\end{aligned}
\tag{13}
$$

We illustrate the methodology through which the nine quantities on the right sides of Eqs. (13) can be evaluated with the help of examples. Analogous to Eqs. (7) a quantity, for example, $dT_{CO_2,emiss}$, can be expressed as the temperature difference:

$$
dT_{CO_2,emiss} = T(\alpha, \epsilon, H) - T(\alpha, \epsilon + d\epsilon_{CO_2}, H),
\tag{14}
$$

where, $d\epsilon_{CO_2}$ is the perturbation to the planetary emissivity caused by a change to the $pCO_2$. This perturbation can be estimated by factorizing the PI to mid-Pliocene planetary emissivity change, $\Delta\epsilon$, using Eq. (1) as:

$$
\Delta\epsilon = d\epsilon_{CO_2} + d\epsilon_{orog} + d\epsilon_{ice},
\tag{15}
$$

where each term on the right hand side is evaluated using Eqs. (2) for the case $\xi = \epsilon$. Similarly, the quantity $dT_{orog,H}$ for example, can be written as

$$
dT_{orog,H} = T(\alpha, \epsilon, H) - T(\alpha, \epsilon, H + dH_{orog}),
\tag{16}
$$

where the change in the divergence of the meridional heat transport due to a change in orography, $dH_{orog}$, is obtained by factorizing $H$. Likewise, all the nine quantities in Eqs. (13) can be evaluated. Additionally, the factorization of the clear-sky components of $\alpha$, $\epsilon$ and $H$ can be used in conjunction with the method outlined earlier in this section, to disentagle the impact of clouds in each of the nine quantities.

## 4 Results

### 4.1 Model Evolution

The analyses presented in this paper are based on the climatologies computed over 30 year intervals. Table 2 lists the model years over which each simulation's climatology has been computed. These climatologies are computed





towards the end of considerably long numerical integrations in order to ensure that each of the simulations is in a state of quasi-equilibrium. We therefore begin the discussion of our results by first discussing the evolution and the quasi-equilibrium characteristics of the five sensitivity experiments that are new in this paper.

Figure 3 shows the evolution of the 2 m surface air temperatures (SAT) and the top of the atmosphere (TOA)
energy imbalances in the new simulations. The timeseries for the evolution of the SATs demonstrate that all models have reached the desired state of quasi-equilibrium. The TOA energy imbalance timeseries show that towards the end of their integrations, all models have either reached a steady state of energy imbalance ($Eo^{280}$ and $Ei^{280}$) or are progressing on a very slow declining trend. In all cases the climatological TOA energy imbalances are quite small (Table 2). Both these diagnostics strongly suggest that all models have reached quasi-equilibrium. Since the
planetary TOA energy imbalance is controlled by the oceans, as they are the largest reservoirs of heat in the climate system, it is instructive to examine the mean state of the oceans throughout the water column to further assess the reasonableness of quasi-equilibria. Figure 4 shows the evolution of the mean ocean temperatures at all depths as Hovmöller diagrams for each of the sensitivity simulations. We see that in $Ei^{280}$ the temperatures in the upper ocean (0–550 m) have nearly stabilized while the ocean below continues to warm at a very slow rate of $\sim 0.04\,^{\circ}\mathrm{C}\,\mathrm{century}^{-1}$
(see Supplementary Table S1). A very similar situation is observed for $Ei^{400}$ but with a slightly greater rate of warming of the middle (550–1850 m) and the deep oceans (below 1850 m) which is clearly visible in the Hovmöller diagram. The SST and the upper ocean temperatures have however stabilized and do not exhibit any trend. This simulation has settled to a TOA imbalance of $0.16\,\mathrm{Wm}^{-2}$, which even after almost 1,500 years of integration, remains stubbornly identical to the climatological TOA imbalance of $0.17\,\mathrm{Wm}^{-2}$ in the $E^{400}$ simulation of Chandan and Peltier (2017)
from which it was branched (see Sect. 2.2.3).

As mentioned previously, the oceans in the $Eo$ experiments were initialized from the same restart files produced at an intermediate time during the integration of the 400 ppmv $pCO_2$ mid-Pliocene experiment (Fig. 2). The ocean in this initial condition was previously evolving under the influence of a 400 ppmv $pCO_2$ atmosphere and is therefore much too warm for a 280 ppmv $pCO_2$ atmosphere. Thereupon, $Eo^{280}$ evolves by first eliminating all the excess heat
contained in its oceans, (Fig. 3b) from the model at the TOA to reach quasi-equilibrium after just 700 model years, and remains in quasi-equilibrium for the subsequent 200 years of integration. Some of the excess heat also permeates downward to warm the deep oceans. The TOA energy imbalance, at only $0.03\,\mathrm{Wm}^{-2}$, is the best among all our experiments and the oceans from the surface down to a depth of $\sim 1900$ m shows zero temperature trend. The $Eo^{400}$ simulation on the other hand does not show much variation in its TOA energy imbalance during the course of its
evolution. The absence of any large initial transient in the TOA energy imbalance suggests that the initial ocean state was a very close match to the $Eo^{400}$ case. The TOA energy imbalance starts at $\sim 0.15\,\mathrm{Wm}^{-2}$, then slowly increases to just over $0.20\,\mathrm{Wm}^{-2}$ during which the deep ocean warms up (Fig. 3d) and then the TOA imbalance begins a very gradual decrease before settling into the climatological value of $0.07\,\mathrm{Wm}^{-2}$. The SST and the upper ocean show a trend of $\sim 0.03\,^{\circ}\mathrm{C}\,\mathrm{century}^{-1}$ while the deepest ocean is warming at twice that rate.





Among the five new experiments in this paper, only $\mathrm{E}oi^{280}$ was started from modern day ocean temperatures (Levitus and Boyer, 1994) so as to maintain conformity to the $\mathrm{E}oi$ experiments in Chandan and Peltier (2017). Therefore while the $\mathrm{E}i$ and the $\mathrm{E}o$ experiments benefited from the spin-up already performed for their antecedent experiments, $\mathrm{E}oi^{280}$ had to undergo full spin-up. In hindsight, starting from modern conditions was a good choice

as it appears that the modern day initial state was fairly close to the state that the model wanted to evolve towards, at least in the upper oceans. The TOA energy intake during the first $\sim 1000$ years was invested almost exclusively in warming the deep oceans as seen in Fig. 3e. After this warming, the TOA energy imbalance stabilized and has remained essentially fixed.

## 4.2    Surface Warming: Impacts of Boundary Conditions

In order to understand the impacts of the changes to the orography, the ice sheets and the $p\mathrm{CO}_2$ on the mean annual surface air temperature (MASAT), we begin by first reviewing the attributes of the MASAT anomalies obtained by means of simple differencing between the sensitivity experiments and the control experiments. The MASAT anomalies of the $\mathrm{E}i$ and the $\mathrm{E}o$ sensitivity experiments with respect to control experiments are shown in Fig. 5. In discussing these anomalies, our intention is to focus upon the impacts of changes to the surface boundary conditions (orography

and ice sheets) and therefore, the control experiments for the computation of these anomalies are chosen such that both the control and the experiment from which we are differencing the control, have the same $p\mathrm{CO}_2$. This ensures that the anomalies of interest are not overwhelmed by the anomaly from the difference in $p\mathrm{CO}_2$. The effect of changing the $p\mathrm{CO}_2$ is discussed subsequently.

Two MASAT anomalies that stem from the changes to the ice sheets: $\mathrm{E}i^{280} - \mathrm{E}^{280}$ and $\mathrm{E}i^{400} - \mathrm{E}^{400}$ are shown

in Fig. 5a–b. Both anomalies bear close resemblance to one another in the southern hemisphere (SH), indicating that changing the distribution of the ice sheets over the Antarctic continent (Sect. 2.2.3) yields the same outcome regardless of the background value of $p\mathrm{CO}_2$. The strong warming seen in the Australian and the French sectors of Antarctica is due to the local reduction in orography resulting from the removal of some of the marine grounded ice within the Aurora and the Wilkes sub-glacial basins and the accompanying decrease in ice sheet elevation in the

surrounding regions (see orography anomalies in Fig. 1d). The cooling over the interior of Antarctica is also related to changes to the orography; in this case, a thicker East Antarctic ice sheet in the PRISM4 reconstruction. However, it is seen that the anomalies in the northern hemisphere (NH) do depend on the $p\mathrm{CO}_2$ background. An extensive region of $> 5\,^{\circ}\mathrm{C}$ warming south of Greenland and extending from the Labrador Sea through the GIN Seas and into the Barents Sea, that is present in the anomaly of the $\mathrm{E}i^{280}$ experiment, is absent from the anomaly of the $\mathrm{E}i^{400}$

experiment. The anomaly over Greenland itself, nevertheless remains identical between the two experiments and is orographic in origin as a result of the removal of much of the Greenland ice sheet.

Two MASAT anomalies that stem from the changes to the orography: $\mathrm{E}o^{280} - \mathrm{E}^{280}$ and $\mathrm{E}o^{400} - \mathrm{E}^{400}$ are shown in Fig. 5c–d. In contrast with the ice sheet induced anomalies, these anomalies are larger in magnitude, spatially much more extensive and almost entirely positive. In the SH the largest anomalies are found clustered in coastal regions



of Antarctica where present-day marine grounded sections of the ice sheets have been removed to make way for an open ocean during the mid-Pliocene (see orography anomalies in Fig. 1e). This includes all of East Antarctica and small coastal pockets in West Antarctica, such as those in the location of the present-day Amery Ice Shelf and in the Wilkes Basin and the Aurora Basin. In these locations the MASAT anomalies result from both (i) the reduction in

orography and (ii) the change in the surface boundary from ice to water. In the NH the warming over the Queen Elizabeth Archipelago and the Arctic coast of Siberia is directly associated with the increased fraction of land and the changes to local vegetation. Once again we see that there is a broad region of extensive warming around Greenland that is present in the anomaly of the 280 ppmv experiment but absent, or at-least greatly diminished in the anomaly of the 400 ppmv experiment.

In addition to the two ice sheet related anomalies and the two orography related anomalies that are shown in Fig. 5, there are still other combinations of PlioMIP2 simulations that could also have been used to assess the impacts of changes to these two BCs (Sect. 3.1). For example, two additional differences, $\mathrm{E}oi^{280} - \mathrm{E}o^{280}$ and $\mathrm{E}oi^{400} - \mathrm{E}o^{400}$, also in principle capture the impact of the same changes made to the ice sheets. Similarly, there are a total of four combinations that explore the impact of changes to the orography. These additional anomalies are not included in

Fig. 5. This is because we wanted to focus upon just those differences that would be the most natural choices for an investigator wanting to test for the effects of changing the boundary conditions. Furthermore, by contrasting only two such differences we are able to satisfactorily make our point regarding the dependence of the results upon the background state. In the next section we will revisit the impacts of these boundary conditions using the factorization technique which employs our complete set of experiments.

In discussing the effects of changing $p\mathrm{CO}_2$ we depart briefly from the preceding minimalist argument and show all four possible combinations (Fig. 6). The control simulations here are the 280 ppmv $p\mathrm{CO}_2$ experiments. It is readily apparent that the background, which in the case of these anomalies is characterized by the ice sheets and the orography, again yields considerable influence on the anomaly. The broad, elevated warming around Greenland, seen in the difference $\mathrm{E}^{400} - \mathrm{E}^{280}$, and which is related to the local reduction in sea ice, is absent in the other differences.

This is because the extensive region of thin, fragile sea ice that is present in $\mathrm{E}^{280}$ and whose melting in the $\mathrm{E}^{400}$ experiment leads to this warming, is absent in the $\mathrm{E}i$, $\mathrm{E}o$ and $\mathrm{E}oi$ cases (because of the warming introduced by these modifications) and is no longer available to be thawed by the warming from the increased $p\mathrm{CO}_2$. Curiously, the $\geq 5\,^{\circ}\mathrm{C}$ warming over the Arctic present in the difference $\mathrm{E}^{400} - \mathrm{E}^{280}$, and in the difference between the $\mathrm{E}i$ simulations is absent from the other two differences. Essentially, this is saying that the impact of changing $p\mathrm{CO}_2$ is less in the full

$\mathrm{E}oi$ case than it is when changing $p\mathrm{CO}_2$ separately in $\mathrm{E}i$ and $\mathrm{E}o$ and adding them together.

The discussions thus far highlight the dependence of the response of the climate system to a perturbation in its boundary conditions, on the stationary background state of the system. Both columns in Fig. 5 are supposed to represent the impacts of the same changes to the boundary conditions, but the inferred impacts differ significantly between them. This is the motivation for application of the factorization methodology previously described, and to

which we now turn.



### 4.2.1 Insights from factorization

The total MASAT anomaly between the mid-Pliocene and the PI, $\Delta T$, and its decomposition using Eqs. (2) into contributions originating from changes to $pCO_2$, $(dT_{CO_2})$, orography, $(dT_{orog})$, and ice sheets, $(dT_{ice})$, are shown in Fig. 7. The zonal means of these anomalies are shown in Fig. 8 and their global means are listed in Table 3. The difference $\Delta T$ is the same difference previously discussed in Chandan and Peltier (2017). It is readily seen that the factorization has exposed distinct features between the anomalies of the sensitivity simulations shown in Figs. 5 and 6 which provides confidence in the methodology we are applying.

The warming from the 120 ppmv increase in $pCO_2$ between the PI and the mid-Pliocene is nearly uniform in the zonal direction (Fig. 7b) and increases in magnitude towards the poles where $dT_{CO_2}$ reaches as high as $5\,^{\circ}\mathrm{C}$ (Fig. 8). This polar amplification is reminiscent of the pattern of temperature increase that the climate system is experiencing at present. Away from the poles, $dT_{CO_2}$ is nearly uniform with a warming $\sim 1\,^{\circ}\mathrm{C}$. The two lobes of locally enhanced warming on either side of the Antarctic peninsula are due to the enhancement of the warming from the reduction of sea ice. The global means in Table 3 show that $dT_{CO_2}$ is the largest contributor to the mid-Pliocene warming. This conclusion was also reached by Lunt et al. (2012b) using the much older PRISM2 palaeoenvironmental reconstruction (Dowsett et al., 1999).

The orographic contribution to the mid-Pliocene warming (Fig. 7c) is largely confined to the coastlines of Antarctica and to high-latitudes in the NH where there have been significant changes to the orography and the LSM. In fact, the largest $dT_{orog}$ values exist exactly over those regions where the LSM has been changed. This makes sense as the change from ice/permafrost to (warmer) ocean represents a drastic change to the lower boundary conditions of the atmosphere. Surrounding these are expansive regions of reduced, but still relatively high, temperature anomalies over the oceans which are due to changes to the sea ice concentration (the anomaly is seen to closely follow the contours of sea ice reduction). The elevated warming over the Tibetan Plateau present in the total anomaly, $\Delta T$, is seen to be picked up by the factorization process as an almost exclusively orography related change. In the zonal mean, $dT_{orog}$ is seen to have an amplitude as high as $dT_{CO_2}$. In the mid-latitudes of the SH the zonal mean of $dT_{orog}$ reaches a global minimum which is due to the fact that there is very little land in those latitudes and therefore very little change in orography.

The factorization has consigned $dT_{ice}$ almost entirely to the high-latitudes. This is seen both in the spatial pattern of the anomaly and in the zonal mean. In the latter, it is seen that $dT_{ice}$ has almost no influence throughout the tropics and the mid-latitudes. This is again a positive outcome, because we would be rightfully suspicious of the factorization if it had assigned any effect on the temperature anomaly in regions far removed from the locations of land ice removal.

These discussions show that the results from the factorization of the MASAT are physically comprehensible, which is certainly a requirement for us to be confident in the outcome of the factorization. A final check on the validity of these results is the assessment of how faithfully the total temperature anomaly and its factorized components satisfy



Eq. (1)? This is assessed with the help of the residual $\Delta T - (dT_{CO_2} + dT_{orog} + dT_{ice})$ which is shown in Supplementary Fig. S1. The residual shows that the factorization has done a remarkably good job throughout the globe. A ribbon of large valued residuals that extends from the Labrador Sea to the Barents Sea is the only notable exception. A possible reason for why factorization has performed poorly in this region is presented in the observation that the ribbon

closely follows the margins of sea ice. As elaborated further in Sect. 4.5 below, the complex thermodynamics of sea ice at its margins makes it difficult for the factorization method to disentangle the effects of the various changes to boundary conditions on the sea ice concentration. Since the sea ice is involved in thermal exchange with the air in its vicinity, the complications owing to the thermodynamics of sea ice are also communicated to the surface air, leading to the failure of the SAT factorization in those regions. This interaction with the sea ice should, however, also lead

to the presence of large residuals off the coast of Antarctica. But this is not found to be the case. It is likely that this asymmetry between the two hemispheres is due to the additional impacts of northward heat transport and deep water formation.

     Supplementary Figs. S2–S5 show the factorization of the SAT anomalies for each of the seasons. These figures along with the global means of the factorized component for each season (Table 3) show that the warming from

the elevated $pCO_2$ remains remarkably steady across all seasons. The warming from changes to the ice sheets is also seen to be relatively steady. The orographic contribution to the warming on the other hand is characterized by extremely large amplitude differences between the seasons, such that, its impact during JJA in twice that during DJF. It is the variation in this component that is seen to reduce the total warming in winter and which leads to the increase in seasonality (compared to the PI) that was previously reported in Chandan and Peltier (2017). The global

mean residuals of the seasonal factorizations are given in Table 3 and these are also found to be very small.

## 4.3   Mechanisms of Warming

We next focus on understanding the mechanisms that lead to the observed warming in our 400 ppmv $pCO_2$ mid-Pliocene simulation compared to the PI. Since the orbital parameters and the solar constant are the same between the mid-Pliocene and the PI simulations, the TOA shortwave flux, and its latitudinal distribution is also identical

between the two. Therefore the warming has to do with how much of the incident energy is retained by the planet and how it is transported in the meridional direction to balance energy deficits.

     In Fig. 9 we show the zonal mean MASAT anomaly between the PI and the 400 ppmv $pCO_2$ mid-Pliocene experiment, and its dissociation using the EBM methodology discussed in Sect. 3.2, into contributions arising from changes to the Earth's radiative properties and the MHT. The curve labeled 'GCM' is the MASAT anomaly obtained

directly from the UofT-CCSM4 model and is identical to the curve identified as $\Delta T$ in Fig. 8. The dashed-gray line labeled 'EBM' is the zonal mean MASAT anomaly predicted using the EBM solution given in Eq. (6). This line is difficult to see in the figure because it virtually overlaps with the GCM anomaly. The impressive agreement between the EBM and the GCM results is further reinforced through their meridional averages, which are $3.82\,^\circ\text{C}$ and $3.80\,^\circ\text{C}$ for the EBM and the GCM respectively.




The other curves on Fig. 9 show the EBM inferred contributions to the MASAT anomaly from changes to cloud albedo (Eq. 9), surface albedo (Eq. 10), cloud emissivity (Eq. 11), emissivity from greenhouse gases (Eq. 12) and the MHT (Eq. 7). The warming from each of these mechanisms should add up to the EBM inferred temperature anomaly if our methodology is soundly formulated. The dashed-black curve in Fig. 9 is the sum of these quantities

and it is indeed nearly indistinguishable from the EBM (and the GCM) deduced profiles of temperature anomaly. The meridional averages (planetary averages) of these mechanisms, given in Table 5, sum to $3.87\,^{\circ}$C which is only marginally greater than either the EBM or the GCM planetary average.

The dominant mechanism of warming during the mid-Pliocene is found to be changes to the planetary emissivity. The reduction in emissivity impedes the mid-Pliocene atmosphere's ability to radiate energy to space and thereby

allows it to retain a greater proportion of the energy introduced by the incident shortwave radiation. Planetary emissivity induced warming, $dT_{emiss}$, is found to contribute up to $2.43\,^{\circ}$C, or roughly $63\,\%$ of the total PI to mid-Pliocene temperature anomaly.

The planetary emissivity reduction is almost entirely driven by the reduction in the atmosphere's emissivity owing to the increase in the concentration of the greenhouse gases ($dT_{gg}$): $CO_2$ and water vapour. The impact from changes

to cloud emissivity ($dT_{lwc}$) also leads to a net warming, but it is negligible in comparison to the greenhouse effect. However, this is not a 'textbook warming' resulting from a higher *imposed* $p$CO$_2$ and the concomitant change to the atmospheric water vapour content (which is free to adjust according to the Clausius-Clapeyron relationship), as it is seen that this effect, $dT_{CO_2,gg}$, on it's own would only contribute $1.25\,^{\circ}$C, or roughly $54\,\%$ of the total warming that we have inferred in our analysis to be due to increased greenhouse gases. The other $46\,\%$ of the greenhouse

effect comes from the increase in atmospheric water vapour as a result of the changes to the orography and the ice sheets. These two changes to the boundary conditions contribute $0.79\,^{\circ}$C and $0.26\,^{\circ}$C to the greenhouse effect, respectively. Water vapour is therefore an extremely potent greenhouse gas during the mid-Pliocene. Supplementary Fig. S6 contrasts the zonal mean of the vertically-integrated total precipitable water content between the PI and the mid-Pliocene. The total precipitable water in the mid-Pliocene atmosphere is $1.48 \times 10^{16}$ kg which is an increase

of $\sim 22\%$ over the PI amount of $1.21 \times 10^{16}$ kg. In this regard, the mid-Pliocene greenhouse effect is a textbook greenhouse warming enhanced by the long timescale feedbacks from changes to orography and ice sheets.

Warming owing to changes to the planetary albedo, $dT_{alb}$, (shown in Fig. 10a) contributes $1.47\,^{\circ}$C to the mid-Pliocene warmth. $dT_{alb}$ results from the net effect of cooling driven by changes to the cloud albedo ($dT_{swc}$, $-0.65\,^{\circ}$C) and warming driven by changes to the surface albedo ($dT_{salb}$, $2.12\,^{\circ}$C). Cloud albedo changes are seen to cool the

planet everywhere except in the tropics and the mid-latitudes of the SH (Fig. 10c). The cooling is most pronounced at high-latitudes where temperature anomalies as low as $-10\,^{\circ}$C are observed. Despite the presence of these large anomalies at high-latitudes, the meridionally integrated $dT_{swc}$ only comes out to a modest $-0.65\,^{\circ}$C. This is because the high-latitudes occupy a very small fraction of the Earth's surface area and the global average is thus influenced more by the modest magnitude anomalies in the tropics and the mid-latitudes that together account for most of the





surface area. In these regions, the see-saw pattern in $dT_{swc}$ between the two hemispheres leads to cancellations that further reduces the net impact of changes to the cloud albedo.

Supplementary Fig. S7b for the zonal mean anomaly of the surface albedo shows that throughout the range of latitudes, the mid-Pliocene surface albedo is either less than or equal to the PI surface albedo. Consequently,

surface albedo's contribution to the MASAT anomaly is either positive or identically zero (Fig. 9). Table 5 shows that globally, surface albedo change is the second most dominant (after the greenhouse effect) of the five mechanisms of warming. Regionally, its magnitude peaks at $60°$–$70°$ S and north of $80°$ N latitudes. It makes perfect sense for the largest reduction in surface albedo (and the greatest temperature increase) to occur in these latitudes since the direct changes of the removal of land ice, and the accompanying changes in the reduction in sea ice and snow fall on

land have drastically reduced the surface albedo in these regions.

The largest contributor to surface albedo change (Supplementary Fig. S7b) and correspondingly the largest contributor to surface albedo induced warming (Fig. 10b) is orography. Besides the obvious impacts near the poles, orography has also been implicated by the factorization process to be the sole cause for the modest surface albedo induced warming at $15°$ N and $30°$ N. This warming at $15°$ N is due to the albedo reduction from the vegetation

changes that are made in conjunction with the orography changes. Specifically, it is related to the increase in grassland and dry shrubland at the southern margins of the Sahara desert which leads to a decrease in surface albedo (Supplementary Fig. S8). The warming at $30°$ N is related to the orographic changes over the Tibetan Plateau (Supplementary Fig. S8) and the associated changes in ground snow cover. It is encouraging to see that these orography related changes are correctly identified by the factorization process. The impacts of ice sheets and

$pCO_2$ on surface albedo are confined to the polar regions, as there are no mechanisms (in the absence of a dynamic vegetation model) for either of them to affect the surface albedo away from the poles ($CO_2$ can also be an influence through changes to the snowfall rate over the tops of mountain ranges, but this is likely to be a small effect compared to the influence of dynamic vegetation).

The MASAT anomalies at high-latitudes are seen to result from a delicate balance between individual contributions

with very large magnitudes and opposing signs. In particular, the partial cancellation of the two effects with the largest amplitudes — surface albedo changes and cloud albedo changes, is very important in determining the MASAT anomaly. Since clouds are heavily parametrized in climate models, there is considerable uncertainty within these models with regards to the effects of clouds. In this case, what this uncertainty means is that any change to the cloud component would clearly impact the effect of planetary albedo, and its partition into the effects coming from

surface albedo changes and cloud albedo changes. The practical implication of this is that our ability to model the mid-Pliocene climate accurately depends in large part on the ability of our climate models to properly simulate clouds.

The redistribution of heat in the meridional direction cannot affect the total energy balance of the planet, and therefore cannot lead to net cooling or warming, as it simply moves heat from areas of local excess to areas of local

deficit. Accordingly, the planetary average of the heat transport should be theoretically zero. Table 5 shows that



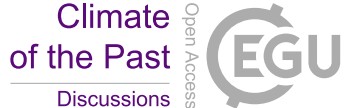

in our analysis the meridionally integrated $dT_H$ ($-0.03\,^\circ\mathrm{C}$) does come out sufficiently close to zero. Although heat transport does not affect the net planetary energy balance, it does affect the local energy balance. Fig. 9 shows that the local impact on temperature from a change to the MHT is non-negligible only in the high-latitudes of both hemispheres. In these regions the EBM analysis shows that there is a cooling from a reduction in the MHT during

the mid-Pliocene.

## 4.4  A mid-Pliocene Perspective on Climate Sensitivity

Climate sensitivity has been a cornerstone concept in climate science for decades owing to its popularity as a means of quantifying the response of the climate to anthropogenic greenhouse gases. However, in the most general form, it is simply defined as the change in the globally averaged MASAT in response to an imposed radiative forcing. Several

specific forms of climate sensitivity are possible within this general definition subject to which processes are treated as forcings and which are treated as internal feedbacks. Much confusion has been created in the literature in recent years due to the introduction of several different forms of climate sensitivity together with inadequate differentiation between the forcings and the feedbacks (PALAEOSENS Project Members, 2012). As Previdi et al. (2013) note in a recent review of the subject: "It is important to distinguish between these forms [of climate sensitivity] in order to

avoid confusion and to reconcile results from different studies that employ alternative sensitivity definitions."

Two measures of climate sensitivity appear in the climate science literature more often than others. The first is the Equilibrium Climate Sensitivity (ECS), more commonly known as the Charney Sensitivity (CS), after Jule Gregory Charney who was chair of the 1979 National Research Council 'study group on carbon dioxide and climate' (National Research Council, 1979) which is widely recognized as the first modern assessment of the impact of greenhouse

gases on the warming of the lower atmosphere. The CS is defined as the change in surface temperature, per $Wm^{-2}$ of applied radiative forcing, once all the fast feedback processes such as changes to water vapour, sea ice, aerosols and clouds have come to equilibrium. Since these processes equilibrate very quickly, with even the slowest among these, sea ice, taking only a few decades to reach equilibrium, the typical timescale over which CS is considered to be applicable is a hundred years. For this reason CS has become important in discussions of the end-of-century climate

particularly within the context of the work that has been coordinated by the IPCC.

The second measure of climate sensitivity that is widely employed, especially in the palaeoclimate community, is the Earth System Sensitivity (ESS) which extends the CS by also including internal feedbacks from slow (long timescale) processes such as changes to land-based ice sheets, vegetation, ocean circulation, and the carbon cycle, and is therefore important to addressing the question of the state of the climate over tens-of-thousands of years.

However, in practice, only some of these slow feedbacks are included in calculations for the ESS, and others are either omitted or included as forcings. To make matters more complicated, proxy-based estimates of ESS are also affected by orographic changes (as the orography in the past was different from today) which should be ideally considered as a separate forcing, but owing to the difficulty in quantifying the magnitude of forcing from the orographic changes, the orographic effect is unfortunately considered to be an internal feedback in response to an external forcing. A



further difficulty regarding the orographic changes, as we discuss below, is that some orographic changes are in response to the melting of polar ice sheets and the subsequent adjustment of the surface of the solid earth, and these may in fact be reasonably considered to constitute slow internal feedbacks as opposed to externally applied forcing.

We will therefore seek herein to infer a hierarchy of climate sensitivities of increasing complexity in order to explore
the response of the climate over a very large range of timescales and in response to various choices of forcings and feedbacks. The sensitivities that we calculate provide insight not only on how the climate responds to a given forcing over a short timescale, but also on intermediate and very-long timescales. We begin with a discussion of the CS which is relevant on the shortest timescales ($\sim 100$ years) and which has therefore become important to discussions of the climate at the end of the 21$^{\text{st}}$ century. Then we assess the sensitivity on an intermediate-timescale on which
the feedback from ice sheets starts to become important. On the longest timescale we consider the ESS for which we expand upon the work previously presented in Chandan and Peltier (2017).

Before proceeding with the discussion of these sensitivities, it is important to make a comment regarding the notation that we will be using in the remainder of this paper. PALAEOSENS Project Members (2012) (henceforth, simply PP) have proposed a nomenclature with the aim of building a common vernacular among researchers to
unambiguously discuss climate sensitivities and to avoid the confusion that Previdi et al. (2013) have cautioned us about. We will therefore employ PP's notation and we begin with a brief overview of the fundamentals of this notation. The climate sensitivity, $S$, in its general form is expressed as:

$$S = \frac{\Delta T}{\Delta R} \,, \tag{17}$$

where $\Delta R$ is some imposed radiative perturbation and $\Delta T$ is the ensuing globally averaged change in MASAT.
Furthermore, for some abstract process $P$, we label the radiative perturbation from that process as $\Delta R_{[P]}$ and the temperature response due to that perturbation and from all associated fast feedbacks as $\Delta T_{[P]}$. Finally, the 'specific sensitivity' (a term introduced by PP) of the climate system to the process $P$ is defined as

$$S_{[P]} = \frac{\Delta T_{[P]}}{\Delta R_{[P]}} \,. \tag{18}$$

We now begin with the discussion of the shortest timescale sensitivity, CS. Since the only radiative forcing is that
from the difference in the $p\text{CO}_2$ between the mid-Pliocene and the present day, $\Delta R$ in Eq. (17) is just the specific forcing $\Delta R_{[CO_2]}$. The change in temperature in response to this forcing, $\Delta T_{[CO_2]}$, is just that due to both the imposed radiative forcing and feedbacks from all fast timescale processes. Accordingly, CS, (which PP denote as $S^a$ where the superscript stands for 'actuo', meaning present-day) is given as:

$$S^a = \frac{\Delta T_{[CO_2]}}{\Delta R_{[CO_2]}} \,. \tag{19}$$

It is thus seen that CS is just the specific sensitivity to the change to $p\text{CO}_2$ i.e. $S^a = S_{[CO_2]}$. The radiative forcing is easily calculated (Myhre et al., 1998) as

$$\Delta R_{[CO_2]} = 5.35 \times \ln(400/280) = 1.91 \, \text{Wm}^{-2} \,. \tag{20}$$





In contrast, it is not at all straightforward to compute $\Delta T_{[CO_2]}$. To be sure, given our objective of contributing a mid-Pliocene perspective on CS, $\Delta T_{[CO_2]}$ must be derived in some way from our set of simulations that incorporates the effect of the mid-Pliocene boundary conditions. This is one reason why we cannot use for $\Delta T_{[CO_2]}$ the temperature difference between the control simulations $E^{400}$ and $E^{280}$ as neither of these simulations communicates any

information regarding the unique mid-Pliocene surface boundary conditions. Furthermore, a general argument as to why we cannot simply use the temperature difference between any two of our simulations, differing only in the $CO_2$ concentration (e.g. $Eoi^{400}$ and $Eoi^{280}$), is that the results above show that there is no reason to prefer any one such difference over any of the other differences. This is where the factorization methodology becomes useful because it tries to circumvent the subtlety surrounding the dependence of the response of a system on the background, to tease

out the specific response from changing $CO_2$.

Factorization of the MASAT anomaly comes to our rescue and provides a way (Eq. 2) to estimate $\Delta T_{[CO_2]}$, and whose value was previously calculated to be $1.67\,°C$ (Table 3). With this we infer the CS of the UofT-CCSM4 model to be $0.88\,°KW^{-1}m^2$, or equivalently $3.25\,°K$ per doubling of $pCO_2$. This is remarkably close to the CS estimate of $3.2\,°K$ per doubling of $pCO_2$ that has been reported by Bitz et al. (2012) for the closely related NCAR-CCSM4 model.

However, there is an important difference between their study and ours; their result is a modern-day inference of the CS as their simulations are performed against a background defined by modern-day orography, LSM, ice sheet distribution, bathymetry and vegetation. This is different from our case since the temperature difference we use is the average (Eq. 2) of all the possible ways in which the temperature difference between a 280 ppmv and 400 ppmv atmosphere can be calculated with differing background states.

Charney Sensitivity is a reasonable measure of the climate system's sensitivity only on a timescale of $\sim 100$ years because it does not incorporate the effects of the slower feedback processes. Among the various slow feedback processes that operate in the climate system, one which holds particular relevance for the future, is the feedback from large-scale wasting of polar ice sheets. This process operates on timescales of a few-hundred to half-thousand years, which is not fast enough to qualify for the purposes of CS, but is still, considerably faster than the slow feedback

processes such as changes to ocean circulation and the carbon-cycle, and glacial isostatic adjustment (GIA) induced changes to orography (for recent discussions of the global process of isostatic adjustment see Peltier et al. (2015) and Roy and Peltier (2017)). State-of-the-art numerical modelling of the Antarctic ice sheet, forced with atmospheric boundary conditions derived from regional climate model simulations using projected emission scenarios suggest that the entire West Antarctic Ice Sheet could collapse in just 250 years time (Pollard and DeConto, 2016). The

feedback from land ice is therefore relevant to the understanding of our climate a few hundred years into the future, and indispensable to discussions of lasting climate change mitigation strategies. We can quantify the effect of this feedback with a new intermediate-timescale sensitivity which is constructed by extending CS to include the land ice feedback (Fig. 11b). We denote this new sensitivity as $S^i$ and write it as:

$$S^i = \frac{\Delta T_{[CO_2]} + \Delta T_{[ice]}}{\Delta R_{[CO_2]}}, \tag{21}$$



where, the total temperature change is now expressed as the sum of the temperature change due to the higher $pCO_2$, $\Delta T_{[CO_2]}$, and the temperature change as a result of the feedback from the melting of the ice sheets, $\Delta T_{[ice]}$. The factorization of the MASAT anomaly gives $\Delta T_{[ice]} = 0.47\,^{\circ}\mathrm{C}$ (Table 3) and we therefore infer that $S^i = 1.12\,^{\circ}\mathrm{KW}^{-1}\mathrm{m}^2$ or, equivalently, $4.16\,^{\circ}\mathrm{K}$ per doubling of $pCO_2$. This is $\sim 30\,\%$ higher than CS.

5    On the longest timescales the potential of warming in the climate system is quantified by its ESS. Chandan and Peltier (2017) arrived at a mid-Pliocene inference for the ESS of $\sim 7\,^{\circ}\mathrm{C}$ per doubling of $CO_2$ (or $\sim 2\,\mathrm{KW}^{-1}\mathrm{m}^2$). Their calculation assumed that the only forcing to the mid-Pliocene climate was from the change in the concentration of the atmospheric $CO_2$, and that the responses from changes to the ice sheets and orography could be treated as internal feedbacks operating over very long timescales. The treatment of the ice sheet response as an internal feedback is a reasonable assumption since land ice extent and volume respond to the direct radiative perturbation introduced by the greenhouse gases. However, as we will discuss below, for the case of orography this assumption is only partially accurate.

Almost all of the changes to the LSM during the mid-Pliocene occurred due to the reduction in land ice volume and the ensuing increase in global sea level, which can be considered as a slow internal feedback to a higher $pCO_2$. In addition, the component of the mid-Pliocene orography which is due to the isostatic rebound of the surface of the solid earth (orographic change when the surface is above water, bathymetric change when the surface is below water) following the removal of the ice sheet loads can also be considered as an internal response. Because both of these processes are controlled by the Glacial Isostatic Adjustment process (GIA) we refer to these two components of orography collectively as the GIA component of orography. Figure 11d illustrates the feedbacks from the interaction between land ice and the GIA component of orography included in the ESS definition of Chandan and Peltier (2017). Mathematically, this definition is expressed as:

$$ESS = \frac{\Delta T_{[CO_2]} + \Delta T_{[ice]} + \Delta T_{orog}}{\Delta R_{[CO_2]}} = \frac{\Delta T}{\Delta R_{[CO_2]}} \ . \tag{22}$$

But, two additional component of the mid-Pliocene orography (i) that due to mantle convection induced deformation on the surface, usually referred to as dynamic topography and (ii) the filling over northern Canada of the erosional features created by numerous Pleistocene glaciations, are unrelated to the forcing from $CO_2$ and should actually be treated as external forcings. However, in the calculations of Chandan and Peltier (2017) these were also treated as internal feedbacks since there is no means available at present to isolate them from the GIA component of orographic forcing. To the extent that the contribution from these components is not disproportionately large compared to the contribution from the GIA components, the estimate from Chandan and Peltier (2017) is a good representation of the ESS. Furthermore, this choice of the definition of ESS allowed Chandan and Peltier (2017) a straightforward comparison to observationally derived ESS estimates (Pagani et al., 2010) and to the estimates from the original PlioMIP program (Haywood et al., 2013).

However, if the assumption of the dominance of the GIA components of the orography anomaly over the non-GIA components is incorrect, then the inference by Chandan and Peltier (2017) could be an overestimation. In this case,

we need to consider orography as a forcing rather than as an internal feedback. A different sensitivity measure (Fig. 11c) is then necessary, which is the specific sensitivity to changes in $p\mathrm{CO_2}$ and orography (denoted $S^p$ by PP):

$$S^p \equiv S_{[CO_2,OR]} = \frac{\Delta T}{\Delta R_{[CO_2]} + \Delta R_{[OR]}} \,, \tag{23}$$

where, $\Delta R_{[OR]}$ is the orographic forcing and $\Delta T$ is the total temperature change between the PI and the mid-
Pliocene, which is $3.80\,^\circ\mathrm{C}$ (Chandan and Peltier, 2017). The $p\mathrm{CO_2}$ forcing $\Delta R_{[CO_2]}$ is again inferred using Eq. (20). To compute $\Delta R_{[OR]}$, following PP, we assume that the efficacy (Hansen et al., 2005), $e$, of orographic forcing is the same as the efficacy of $p\mathrm{CO_2}$ forcing, i.e. we assume $\Delta T_{[OR]} = e\,\Delta R_{[OR]}$ and $\Delta T_{[CO_2]} = e\,\Delta R_{[CO_2]}$. By eliminating $e$ from these two equations and substituting the result in Eq. (23) we can convert the problem of inferring the orographic radiative forcing into one of inferring the orographically induced temperature change:

$$S_{[CO_2,OR]} = \frac{\Delta T}{\Delta R_{[CO_2]}\left(1 + \frac{\Delta T_{[OR]}}{\Delta T_{[CO_2]}}\right)} \,. \tag{24}$$

Factorization of the MASAT anomaly (Eq. 2) yields $\Delta T_{[OR]} = 1.54\,^\circ\mathrm{C}$, from which we get $S_{[CO_2,OR]} = 1.04\,^\circ\mathrm{KW^{-1}m^2}$ or, equivalently, $3.84\,^\circ\mathrm{K}$ per doubling of $p\mathrm{CO_2}$. A few points need to be made regarding this calculation: firstly, the efficacy of the orographic forcing is almost certainly not the same as that from $p\mathrm{CO_2}$ (Hansen et al., 2005). Therefore, a more general expression for the forcing in the denominator of Eq. (24) would be

$$\Delta R_{[CO_2]}\left(1 + f \times \frac{\Delta T_{[OR]}}{\Delta T_{[CO_2]}}\right) \,, \tag{25}$$

where, $f$ is the ratio of the orography efficacy to the $p\mathrm{CO_2}$ efficacy. Therefore, the inference of $S_{[CO_2,OR]}$ is subject to the uncertainty on the magnitude of $f$. For example, if $f = 0.9$, i.e. if the orography induced radiative imbalance were only $90\,\%$ as effective as $p\mathrm{CO_2}$ induced radiative imbalance of the same magnitude, then $S_{[CO_2,OR]} = 1.09\,^\circ\mathrm{KW^{-1}m^2}$ or, equivalently, $4.03\,^\circ\mathrm{K}$ per doubling of $p\mathrm{CO_2}$. On the other hand, if $f = 1.10$, then $S_{[CO_2,OR]} = 0.99\,^\circ\mathrm{KW^{-1}m^2}$
or, equivalently, $3.66\,^\circ\mathrm{K}$ per doubling of $p\mathrm{CO_2}$. Indeed, the value of $f$ could be well beyond this $\pm10\,\%$ bound, for instance, recently Modak et al. (2016) studied the efficacy of the solar forcing and found it to be $\sim 80\,\%$ relative to the $\mathrm{CO_2}$ forcing in the NCAR CAM5 model, which is a successor to the CAM4 atmospheric model in the presently employed UofT-CCSM4 coupled-climate model.

Secondly, regardless of the efficacy of the orographic forcing (i.e. regardless of $f$), the radiative forcing in $S_{[CO_2,OR]}$
is really just the forcing $\Delta R_{[CO_2]}$ scaled by a factor, which in our case is $\sim 1.922$, and therefore this total radiative forcing can be solved for the atmospheric $p\mathrm{CO_2}$ that would on its own result in the same magnitude of forcing and which comes out to 555 ppmv, or nearly double the PI concentration. It is well known from experiments with GCMs in which $p\mathrm{CO_2}$ is varied that the efficacies of various feedback processes in the climate, and thereby the climate sensitivity itself depends on the magnitude of forcing (Hansen et al., 2005; Colman and McAvaney, 2009) and
generally climate sensitivity reduces with increased forcing. It is partly for this reason, and partly because of the uncertainty in $f$ that $S_{[CO_2,OR]}$ is less than the ESS.





To this point we have considered two forcings, $pCO_2$ and orography, and have either ignored land ice altogether, or treated it as an internal feedback. We now consider the magnitude of climate sensitivity if land ice is also treated as a forcing (Fig. 11e). This case implies a scenario in which the reduced polar ice cover is not a response to the elevated warming from a higher $pCO_2$ and a different orography, but rather that the reduced ice state co-evolved,

independently of these two factors. It is known from studies of ice-rafted debris in the North Atlantic (Jansen et al., 2000; Kleiven et al., 2002) that Greenland used to exist in a nearly ice-free state for a long time until the inception of the present-day ice sheet near the end of the mid-Pliocene. The precise causes for the inception of the Greenland ice sheet (and by corollary the causes for its absence) are not yet fully understood (Contoux et al., 2015). The West Antarctic Ice Sheet also likely did not exist for quite some time prior to the PlioMIP2 interval of marine isotope stage

KM5c (De Schepper et al., 2014). Although the uncertainties surrounding the histories of these ice sheets during the mid-Pliocene is large (De Schepper et al., 2014), we nevertheless extend the possibility of ice sheets being a forcing rather than an internal feedback the benefit of the doubt. Therefore, we consider the specific sensitivity $S_{[CO_2, OR, LI]}$:

$$S_{[CO_2, OR, LI]} = \frac{\Delta T}{\Delta R_{[CO_2]} + \Delta R_{[OR]} + \Delta R_{[LI]}} \, , \qquad (26)$$

where $\Delta R_{[LI]}$ is the forcing from land ice and can be obtained by appealing to the same efficacy argument that was used to calculate $\Delta R_{[OR]}$ earlier. We get $S_{[CO_2, OR, LI]} = 0.9 \,^\circ\mathrm{K W}^{-1}\mathrm{m}^2$ or, equivalently, $3.35 \,^\circ\mathrm{K}$ per doubling of $pCO_2$.

### 4.5 Sea Ice

We include a brief discussion of sea ice in our results because it is a major determinant of the planetary albedo,

reduction of which was found to be the second largest contributor to mid-Pliocene warming. The climatological sea ice concentration anomalies of the five sensitivity experiments and the mid-Pliocene experiment, with respect to control experiments are shown in Fig. 12. The anomalies are shown on a form of Lambert Azimuthal Equal Area projection so as to be able to show both poles in one figure. The analysis of these figures suggests that the orography changes induce a greater reduction in sea ice than the changes to the ice sheets.

We again turn to factorization to see if we can better understand the contributions of each of the boundary conditions changes. Supplementary Fig. S9 shows the outcome of applying the factorization methodology to the sea ice concentration anomaly between the mid-Pliocene and the PI. It is seen that the increase in $pCO_2$ and the changes made to the orography together account for most of the loss of sea ice. The largest orography contributed reductions to sea ice are confined to the Hudson Bay, the Queen Elizabeth Archipelago and the coastlines of the Arctic Ocean

because in the factorization process we have explicitly ascribed all the anomaly at any location where the LSM has changed to orography. However, unlike the MASAT, the residuals from the factorization of sea ice anomaly are very large (Supplementary Fig. S10). This greatly diminishes any prospects of gaining meaningful insights into sea ice changes using the factorization methodology. The large residuals are confined to the margins of the sea ice where



the sea ice is thinnest, least concentrated and most prone to melting. The warming induced from all three boundary condition changes will compete to melt the ice. The thermodynamics of sea ice is a notoriously nonlinear process with many factors influencing the melting and does not simply scale with the magnitude of warming. Along with the fact that there is only so much ice to melt, disentangling the effects from the three processes understandably 5  becomes challenging for the factorization process.

The annual means of the sea ice area and volume in each of our simulations are provided in Table 2. The Arctic sea ice is found to be much more susceptible than the Antarctic sea ice to the climatic changes induced as a result of making a change to any of the boundary conditions. The percentage reduction in volume is always much greater than the percentage reduction in sea ice area. This points to thinning of the sea ice in response to the perturbations. As a 10  result of the thinning from the changes, the average thicknesses of the NH and the SH sea ice drop from $\sim 4.10$ m and $\sim 1.48$ m respectively in our PI control to $\sim 0.75$ m and $\sim 0.88$ m respectively in the 400 ppmv $p\mathrm{CO_2}$ mid-Pliocene experiment.

A careful analysis of the data in Table 2 also reveals a very peculiar dependence of the sea ice response to changes to surface boundary conditions, on the $p\mathrm{CO_2}$ of the system. In any given experimental configuration of the climate 15  model, it is found that the NH sea ice responds more strongly to the changes when the system itself is in the low $p\mathrm{CO_2}$ state than when it is in the high $p\mathrm{CO_2}$ state, i.e. for any given response the relative change is greater in the low $p\mathrm{CO_2}$ case. The opposite situation is found for the SH sea ice which exhibits a greater response to the boundary conditions when in the high $p\mathrm{CO_2}$ state than in the low $p\mathrm{CO_2}$ state.

## 4.6  Meridional Heat Transport

20  The impacts of the changes to the boundary conditions within PlioMIP2, upon the planetary meridional heat transport (MHT) are relevant not only for understanding the mid-Pliocene puzzle, but they are also imperative to continuing efforts to understand the influence on this immensely important component of the climate system under contemporary global warming. It was recently shown in Chandan and Peltier (2017) that with the PlioMIP2 boundary conditions there is a small-but-not-insignificant reduction in the MHT compared to both the PI and the modern-day climates 25  and in both the atmospheric and the oceanic components of the transport. Here, our objective is to understand how each of the individual changes to the boundary conditions have affected the MHT. In what follows, we adhere to the format used earlier for the discussion of the SAT; first we consider MHT changes through selected comparisons, then we factorize the MHT anomaly and assess how reasonable our conclusions derived from the comparisons were.

Figure 13a–b compare the atmospheric, oceanic and total MHTs in the $\mathrm{E}i^{280}$ and the $\mathrm{E}i^{400}$ simulations to the 30  $\mathrm{E}^{280}$ and the $\mathrm{E}^{400}$ controls, respectively. It is readily evident, that modifications to the polar ice sheets have had a fairly inconspicuous effect on the total MHT and the oceanic heat transport (OHT). Some differences are observable, upon careful examination, but the differences are significantly smaller than the magnitudes of the heat transports (i.e. the relative deviations are small). It is also clear that there is very little dependence on the background state characterized by the atmospheric $p\mathrm{CO_2}$ except in the mid-latitudes in the NH. The OHTs can be further decomposed,





into transports in the Atlantic and the Indo-Pacific basins. These decompositions for the E$i$ experiments are shown in Fig. 14a–b. Overall, the modifications to the ice sheets have resulted in an increase in OHT in the Atlantic basin, in the low $p\text{CO}_2$ case and a decrease in the high $p\text{CO}_2$ case. At $30°$ N for example, the Atlantic in E$i^{280}$ transports $\sim 6\%$ more heat than in E$^{280}$ whereas the Atlantic in E$i^{400}$ transports $\sim 5\%$ less heat than in E$^{400}$. A more stark difference

is seen around $50°$ N where the Atlantic OHT in E$i^{280}$ is $\sim 17\%$ larger than E$^{280}$ while that in E$i^{400}$ is $\sim 13\%$ less than control.

The MHTs in the E$o$ experiments are likewise compared to their equivalent $p\text{CO}_2$ control experiments in Figs. 13d–e and 14d–e. In the NH, the orography changes lead to a robust decrease in the total MHT and the magnitude of the decrease is significantly larger than any decrease in the MHT from changes to the ice sheets. For instance,

compared to the PI, the largest decrease in the total MHT in E$o^{280}$ is $\sim 5\%$ while it is $< 2\%$ in the case of E$i^{280}$. This decline in the northward total MHT is seen to be driven almost entirely by the reduction in the northward AHT. In contrast, the effect of orography on the total MHT and the AHT in the SH is very small. The dependence of the MHT on the background $p\text{CO}_2$ is seen to be small in the NH and almost non-existent in the SH. The OHT in the Atlantic basin appears to be the only transport noticeably affected by the background $p\text{CO}_2$ (Fig. 14d–e) wherein we see that

analogous to the E$i$ case, in the mid-latitudes of the NH, there is an increase in the OHT compared to control in the low $p\text{CO}_2$ state and a decrease in the high $p\text{CO}_2$ state. Also it should be noted that unlike the E$i$ case, orography changes are seen to lead to a large reduction in the northward OHT in the Indo-Pacific basin.

In Figs. 13g–h and 14g–h the MHTs for the 280 ppmv and 400 ppmv E$oi$ experiments are compared to their equivalent $p\text{CO}_2$ control experiments. These E$oi$ heat transports look nearly identical to the corresponding E$o$

experiments. This suggests that the ice sheet effects and the orography effects are combining nearly linearly, and thereby enabling the larger magnitude effects from the changes to the orography to dominate the much smaller impacts from changes to the ice sheets.

The impacts from changing $p\text{CO}_2$ are presented for three configurations of the surface boundary conditions in the rightmost columns of Figs. 13 and 14. It is seen that regardless of the surface boundary conditions of the experiments,

the impacts of changing $p\text{CO}_2$ are nearly identical in all cases. Three important observations can be made; firstly, changing $p\text{CO}_2$ has no impact whatsoever on the AHT in either hemisphere. Secondly, in addition to having no impact on the AHT in the NH, there is also no effect on the total MHT or the OHT in the NH. And, lastly, the only impact of changing $p\text{CO}_2$ appears to be a small increase in the southward total MHT driven by the increase in OHT in the oceans of the SH.

### 4.6.1 Factorization results

The factorizations for the MHT, AHT and OHT anomalies (mid-Pliocene minus PI) are shown in Fig. 15. The first feature to notice is the asymmetry between the two hemispheres in the total MHT anomaly: during the mid-Pliocene there is a net reduction in the heat transport in the NH and a net increase in the heat transport in the SH. In the mid-latitudes of the NH, the net decrease is largely the consequence of the decrease in the AHT, whereas in the



tropics and the sub-tropics where the OHT is strongest, it is the reduction in the OHT that drives the decline in the MHT. OHT is also largely responsible for the increase in the total MHT throughout the SH mid-latitudes.

The AHT anomalies are largest in the mid-latitudes, where AHT is the dominant mode of heat transport. Orography is seen to be the dominant driver of OHT anomalies, while the impacts from ice sheets and atmospheric $pCO_2$ are considerably smaller. The AHT anomaly is entirely negative in the NH, and mostly positive in the SH. The factorization performs very well at all latitudes except for between $40°$–$60°$ N where the sum of all factorized components (dashed black line in Fig. 15b) underestimates the total anomaly.

In the mid-Pliocene climate the OHT changes sign at $\sim 5°$ S latitude (Fig. 14); north of this latitude the heat transport is northward, and it is southward, south of this latitude. The OHT anomalies appear to be anti-symmetric about this latitude such that the northward transport is inhibited while the southward transport is energized. The absolute magnitudes of the changes are nearly the same on either side. The factorization shows that the reduction in the northward transport is dominated by the orographic component whose impact outside of the range of this OHT anomaly, is negligible. In contrast, the positive OHT anomaly in the SH is dominated by the $pCO_2$ response, which also, is much smaller outside of the range of this SH anomaly.

# 5   Discussion and Conclusion

In this paper our focus has been on two critical issues, namely, understanding the origins of the mid-Pliocene warmth that has been recently simulated by Chandan and Peltier (2017) using the latest inference of the mid-Pliocene boundary conditions, and arriving at a mid-Pliocene inference regarding the sensitivity of the climate system on a range of timescales. We have organized our discussion in this section around these two topics.

## 5.1   Origins of Mid-Pliocene Warmth

Regarding the origin of the mid-Pliocene warmth, we have designed our investigation so as to address three key questions:

1. What are the impacts of the differences in boundary conditions on the mid-Pliocene warming?

2. What are the physical mechanisms through which the mid-Pliocene warms?

3. How do the modifications to the boundary conditions themselves change the effectiveness of those physical mechanisms?

### 5.1.1   What are the impacts of the differences in boundary conditions on the mid-Pliocene warming?

With our suite of PlioMIP2 simulations, we have been able to perform the very first nonlinear factorization of the MASAT anomaly between the mid-Pliocene simulation forced with the most recent PRISM4 boundary conditions and the PI. A total of eight simulations were needed to employ the technique of (Stein and Alpert, 1993; Lunt

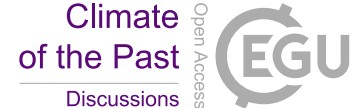

et al., 2012b) for three categories of modifications: orography, ice sheets and atmospheric $p\mathrm{CO}_2$. We find that the factorization approach performs remarkably well in disentangling the effect of each of the three changes on the total warming. The residuals from the factorization are negligible in magnitude over $> 95\,\%$ of the surface area of the Earth.

The higher $p\mathrm{CO}_2$ in the mid-Pliocene atmosphere (relative to the PI) is found to be the primary cause of warming, contributing $\sim 45\,\%$ to the total MASAT anomaly. The contribution from orography change is a very-close second accounting for $\sim 42\,\%$ of the anomaly. The remaining $13\,\%$ of the temperature difference is associated with the reduction in polar ice sheets. In contrast, using the much older PRISM2 boundary conditions for the mid-Pliocene along with the UK Met Office coupled ocean-atmosphere GCM HadCM3, version 4.5 (Gordon et al., 2000), Lunt

et al. (2012b) found the contributions from $\mathrm{CO}_2$, orography and ice sheets to be $48\,\%$, $31\,\%$ and $21\,\%$ respectively. This indicates that, save for the inter-model differences, the large changes to the orography between the newest PRISM4 reconstruction and the older PRISM2 reconstruction have had a significant impact on the warming. Since these orography changes are concentrated at high-latitudes, the impacts of these changes at a regional scale are even higher than those at the global scale. The differences in the ice sheet contribution is also due to the differences in

the polar ice sheets between PRISM2 and PRISM4.

    From the factorization of the seasonal SAT anomalies it is found that the contributions from atmospheric $p\mathrm{CO}_2$ and ice sheets are relatively stable through all seasons. In contrast, the contribution of orography is found to change by $100\,\%$ between DFJ and JJA. The reduction in the orographic warming during DJF is found to be the reason for the increased seasonality during the mid-Pliocene that was previously identified and discussed in Chandan and Peltier

20 (2017).

    In this paper we have extensively applied the factorization technique as formulated in Lunt et al. (2012b), and in doing so we have come to recognize some of its limitations. The performance is not identical across the range of climate observables. While the method is extremely adept at breaking the SAT anomaly into various components, for the case of sea ice concentration, factorization is seen to do a poor job as evidenced by the large residuals that it

produces. Similarly, the method performs very well in breaking down the changes to albedo and emissivity, changes to the total MHT and the OHT, but, it comes up short in factorizing the AHT. It is therefore very important that practitioners of the method consider the residuals so as to be sure that their conclusions are sound.

### 5.1.2   What are the physical mechanisms through which the mid-Pliocene warms?

To investigate the physical mechanisms through which the mid-Pliocene could warm compared to the PI, we have

adopted a widely employed one-dimensional EBM formulation (Heinemann et al., 2009; Hill et al., 2014; Lunt et al., 2012a). Through this we have been able to disaggregate the zonal mean temperature anomaly into contributions from changes to cloud albedo and surface albedo (together, planetary albedo), cloud emissivity and greenhouse gas emissivity (together, planetary emissivity) and changes to the meridional heat transport.





The principal mechanism of warming, responsible for $63\,\%$ or almost two-third of the total warming, is found to be the reduction in the planetary emissivity. This is mostly due to the reduction in the atmosphere's emissivity from increase in the concentration of the atmospheric greenhouse gases: $CO_2$ and water vapour. The effect of cloud emissivity is found to be negligible.

Changes to the planetary albedo accounts for the remaining one-third ($37\,\%$) of the total warming and it is found to result from a balance between the cooling effects of the clouds due to an increase in cloud albedo, and warming due to a decrease in the surface albedo. Both these effects attain their largest values at high-latitudes where the delicate balance between these opposing effects is pertinent to determining the magnitude of total warming. It is therefore clear that the ability of our climate models to characterize the impacts of clouds will be a large determiner of our ability to simulate mid-Pliocene warmth at high-latitudes.

In general, our findings regarding the dominant mechanisms of warming as well as the spatial features of the various mechanisms of warming, are consistent with the outcomes of the previous phase of PlioMIP as reported in Hill et al. (2014); however, the amplitudes of the mechanisms are now greatly amplified in the presence of the revised boundary conditions. Finally, we note that the 2/3–1/3 partition of the contribution to warming between decrease in planetary emissivity and decrease in planetary albedo was also found by Heinemann et al. (2009) for the case of another hothouse in the recent Earth's past – the late Paleocene/early Eocene.

### 5.1.3  How do the modifications to the boundary conditions themselves change the effectiveness of those physical mechanisms?

We have formulated a simple extension to the EBM model with the theoretical formulations of the factorization method to enable us to further evaluate the influence of each of the boundary condition differences on the effectiveness of the mechanisms of warming mentioned above. It is found that changes to $pCO_2$, orography and ice sheets respectively contribute $54\,\%$, $34\,\%$ and $12\,\%$ to the warming from increased greenhouse gases – $CO_2$ and water vapour. While the $pCO_2$ is prescribed within our simulations, the water vapour content is free to adjust to the atmosphere's carrying capacity according to the Clausius-Clapeyron relationship and is found to be a significant contributor to the warming. In fact, the total precipitable water in the mid-Pliocene is $\sim 22\,\%$ higher than the PI. While some of the increased water vapour is the result of the feedback from $CO_2$ induced warming, a significant fraction of water vapour originates as feedback due to orography and ice sheet changes.

Planetary albedo change – the second to leading mechanism of warming, is found to be dominated by changes to orography which contributes $54\,\%$ to the albedo change induced warming. Orography changes are also the highest amplitude changes and although these dominate at the high-latitudes of both hemispheres there is also notable warming in the mid-latitudes of the NH owing to changes to the orography of the Tibetan plateau and changes to vegetation in sub-Saharan Africa. The higher $pCO_2$ is found to contribute $33\,\%$ to the warming, followed by ice sheet contribution at $13\,\%$. The impacts of $CO_2$ and ice sheet changes on the surface albedo induced warming are found to be confined to the high-latitudes.



Changes to the MHT do not change the planetary energy balance, but they do affect the local energy balance. The large scale pattern of MHT change is characterized by a persistent reduction in MHT in the NH and an increase in the MHT in the mid-latitudes of the SH. We find that the modifications to the orography almost entirely explain the MHT reduction in the NH, whereas the increase in MHT in the SH is primarily due to the elevated $pCO_2$ and supported in small part by the increase in MHT due to orography. The orography driven reduction in the NH MHT is the result of a reduction in the OHT in the tropics and a reduction in the AHT in the mid-latitudes. The impact of increased $pCO_2$ is only notable in the SH tropics and mid-latitudes, where the increased $pCO_2$ is found to lead to an increase in the OHT. Its impact on the AHT is found to be entirely negligible.

## 5.2 Mid-Pliocene inference of climate sensitivity

We have attempted to infer sensitivities of the climate system on various timescales through our suite of simulations. Our mid-Pliocene derived inference of the CS (applicable on a timescales of $\sim 100$ years) of the UofT-CCSM4 model is $S^a = 3.25\,°\text{K}$ per doubling of $pCO_2$ (Table 4). This is essentially indistinguishable from the estimate $S^a = 3.2\,°\text{K}$ per doubling of $pCO_2$ that has been reported by Bitz et al. (2012) for the closely related NCAR-CCSM4 model. The accord between these two inferences of CS using very different lines of analysis, and despite the dependence of feedback mechanisms on the temperature (Colman and McAvaney, 2009), shows that the CCSM4 CS of $3.2\,°\text{K}$ per doubling of $pCO_2$ is a very robust result. The difference between the UofT-CCSM4 and the NCAR-CCSM4 model with regards to parameterizations in the ocean model (Chandan and Peltier, 2017) obviously has no consequence for the short-timescale response of the climate as the ocean responds on a much longer timescale.

Following the initial, fast timescale response of the climate to an applied radiative perturbation (which is characterized by the CS), the long-term evolution of the climate is governed by the slow feedbacks that start 'coming online' at various timescales. On the timescale of a few hundred years after the initial radiative forcing, which we refer to here as intermediate timescale, the albedo driven positive feedback from the melting of the polar ice sheets (Pollard and DeConto, 2016) becomes important. We infer the climate system's sensitivity including this feedback, $S^i$, to be $4.16\,°\text{K}$ per doubling of $pCO_2$, which is $30\,\%$ larger than the estimate for the CS. On a still longer timescale, the surface of the solid earth begins to respond to the removal of the ice load, and starts interacting with the climate through changes to the orography and the bathymetry. The climate system's long-term sensitivity, and which includes this GIA response, is characterized by its ESS which we have previously inferred in Chandan and Peltier (2017) to be $\sim 7\,°\text{C}$ per doubling of $pCO_2$.

Although the CS is the focus of much study, we wish to note the particularly large inference for $S^i$, and which one should be cognizant of while planning long-term climate mitigation strategies. The significance of $S^i$ is best illustrated by comparing the potential warming implied by this parameter to predicted levels of warming at year 2300 CE obtained through modelling efforts employing the Extended Concentration Pathway (ECP; Meinshausen et al. (2011)) scenarios. These are future emission scenarios developed as simple extensions of the Representative Concentration Pathway (RCP; van Vuuren et al. (2011)) scenarios for years 2101–2300 CE for the purposes of

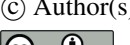



assessing the response of the climate to anthropogenic emissions beyond the end of this century (Collins et al., 2013). Among the several extension scenarios, ECP4.5, in which the radiative forcing stabilizes at $4.5\,\mathrm{Wm^{-2}}$ after year 2150 CE (although the forcing has already begun fluctuating about this value at least 100 years earlier) is of particular interest to us since in this scenario the $p\mathrm{CO_2}$ stabilizes to a value roughly double ($\sim 560$ ppmv) that during the

PI. Although it is abundantly clear by now that under no situation will the future emissions follow this optimistic scenario, this is nevertheless a convenient case for illustrating the impact of feedbacks. The radiative forcing in this scenario is a little over the $3.7\,\mathrm{Wm^{-2}}$ that we would expect from a doubling of $p\mathrm{CO_2}$ due to the presence of other greenhouse gases.

Intercomparison work performed for the IPCC AR5 (Collins et al., 2013; Zickfeld et al., 2013) using Earth System

Models of Intermediate Complexity (EMICs) has shown that under the ECP4.5 scenario we are committed to a warming of $3.11 \pm 0.60\,^\circ$C by 2300 CE (compared to the 1850–1900 reference level). Even longer integrations, out to year 3000 CE, performed with EMICs under the assumption that radiative forcing remains stationary at year 2300 CE values suggests that the warming remains approximately the same. In contrast, our inference of $S^i$ would suggest that even if radiative forcings stabilized today at a $p\mathrm{CO_2}$ level which is nearly identical to that prescribed in our

mid-Pliocene experiment, and $\sim 160$ ppmv less that that in ECP4.5 scenario, then the warming by 2300 CE could still be $\sim 2.1\,^\circ$C greater than the PI. Alternatively, in the case of emissions stabilizing to $\sim 560$ ppmv in year 2150 CE, as in the ECP4.5 scenario, our inference of $S^i$ suggests that we would be firmly committed to a warming of $> 4\,^\circ$C. This implies that the EMIC results, which do not take into account the intermediate timescale feedbacks, could be underestimating the potential for warming on this timescale by $\sim 1\,^\circ$C.

Similarly, the potential for long-term future warming suggested by our high-inference of the ESS can be contextualized by comparing to results of even longer EMIC simulations. The utility of such a comparison is significantly less than that presented above for $S^i$, owing to the extremely unconstrained task of forecasting an emission scenario for the timescales over which ESS is meaningful. Zickfeld et al. (2013) considered an even longer extension of the ECP4.5 scenario in which the 2300 CE radiative forcing is kept stationary until 3000 CE after which the $p\mathrm{CO_2}$ is

allowed to evolve freely. They found that even out to year 4000 CE the EMICs do not predict any increase in surface temperature beyond the 2300 CE estimate. Our mid-Pliocene informed estimate for the ESS would suggest, however, that there is potential for a greater magnitude of warming 2000 years into the future than suggested by EMICs.

*Acknowledgements.* Computations were performed on the TCS supercomputer at the SciNet HPC Consortium. SciNet is funded by: the Canada Foundation for Innovation under the auspices of Compute Canada; the Government of Ontario; Ontario Research

Fund - Research Excellence; and the University of Toronto. DC is very grateful to The Centre for Global Change Science, University of Toronto, which has funded multiple trips to conferences and workshops related to the work described in the paper. The research of WR Peltier at the University of Toronto is funded by NSERC Discovery Grant A9627.



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





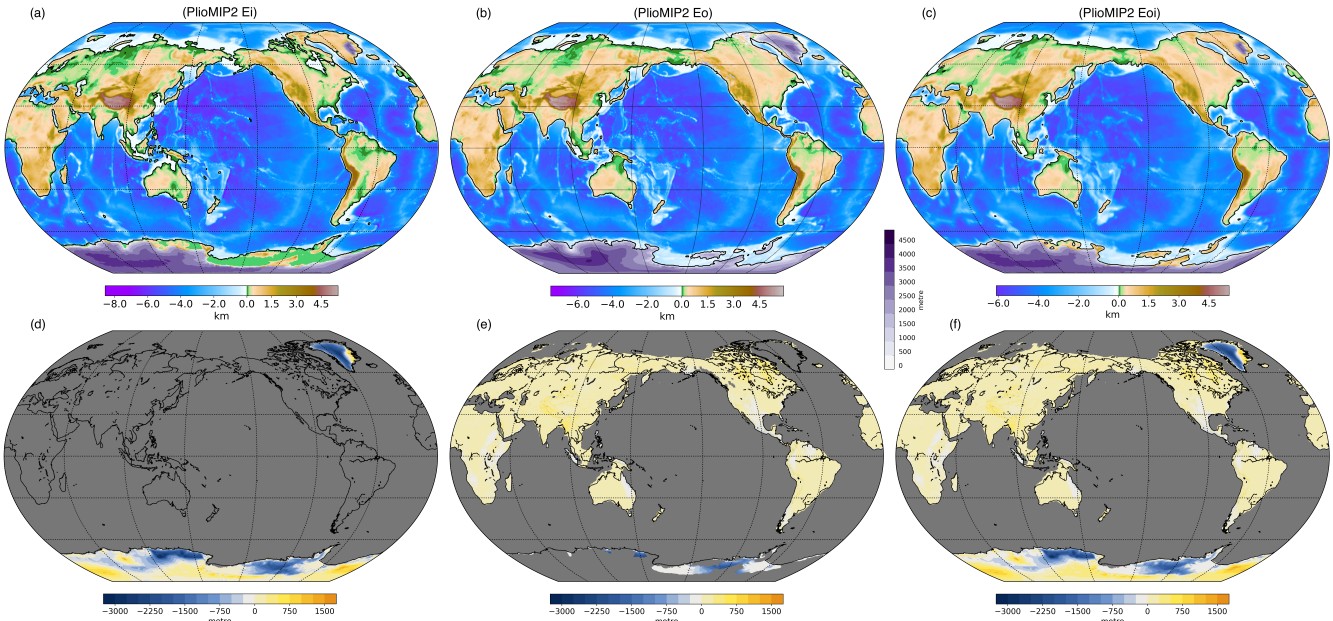

**Figure 1.** The topography for the E$i$, E$o$ and E$oi$ experiment configurations (top row) and the orographic anomaly of those configurations from PlioMIP2 modern (bottom row). The land-based ice sheets in the top row figures are displayed on the purple colored scale that is shown separately.





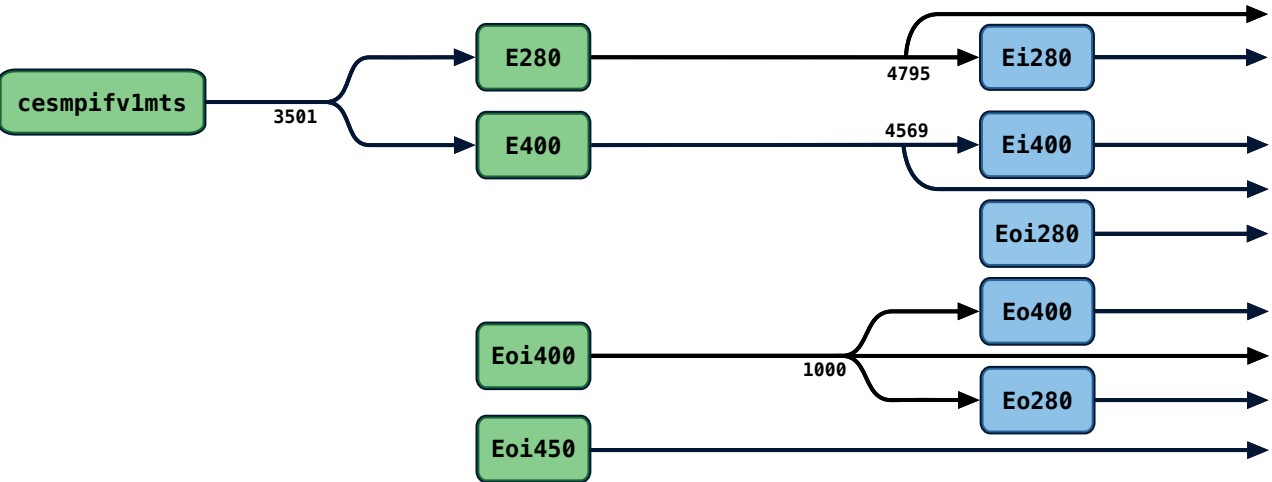

**Figure 2.** Relationship among our PlioMIP2 simulations. The simulations colored green are previously introduced in Chandan and Peltier (2017) while the blue colored simulations are new to this paper. The numbers at the locations of branching indicate the model year from which one or more new simulations were branched. All simulations without an ancestor have their ocean components initialized from modern day conditions (Levitus and Boyer, 1994). The simulation cesmpifv1mts is a non-PlioMIP2 pre-existing PI control simulation.





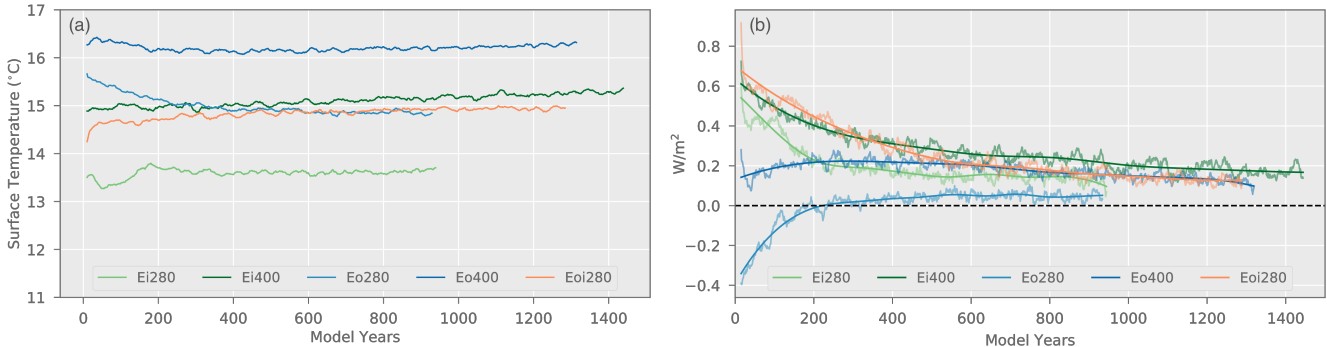

**Figure 3.** Evolution of the (a) 2 m surface air temperature and (b) the top of the atmosphere energy imbalance for the sensitivity experiments.



**Figure 4.** Hovmöller diagrams for the evolution of ocean temperatures as a function of depth in the five sensitivity simulations.





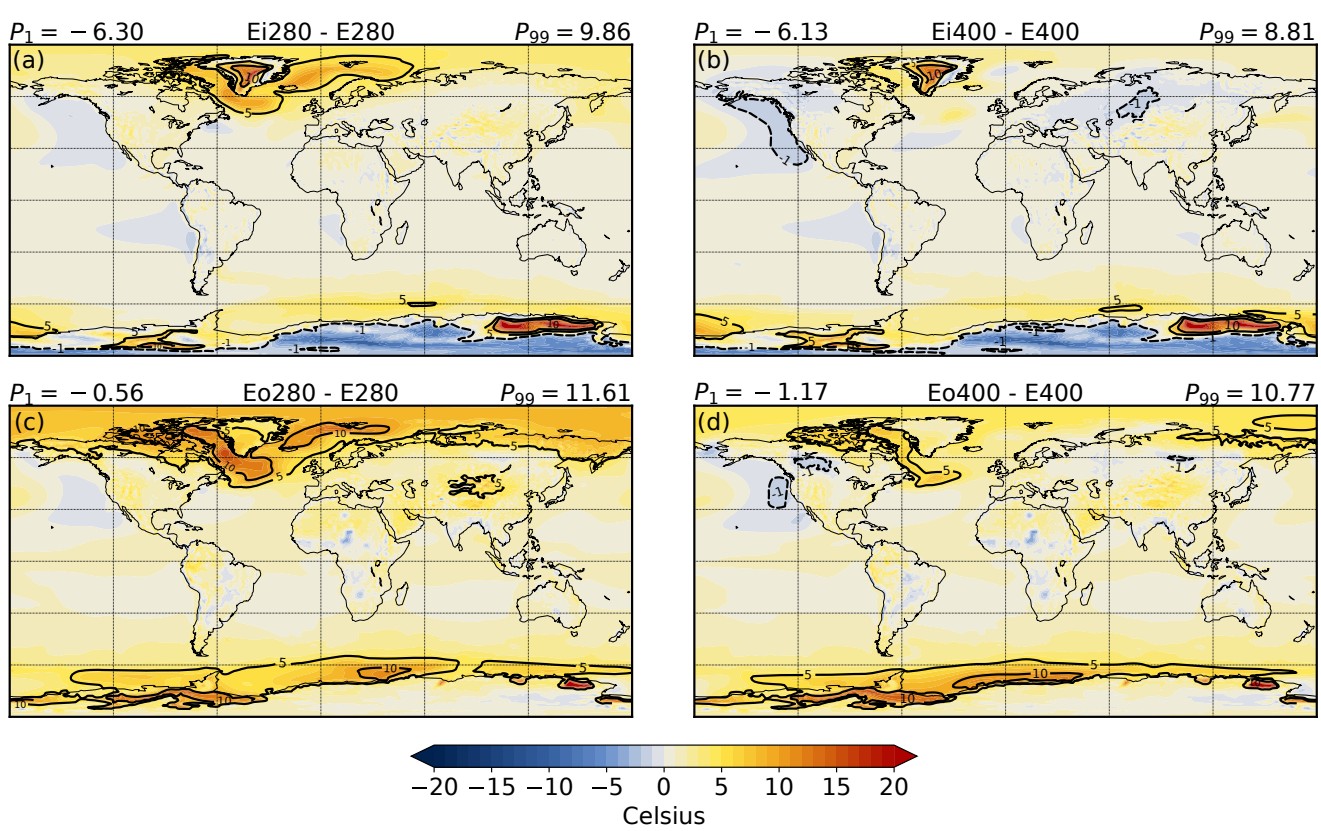

**Figure 5.** Climatological MASAT anomalies of the E$i$ and the E$o$ sensitivity experiments with respect to controls. The control experiment is chosen such that the $pCO_2$ is identical between the sensitivity experiment and the control. The left column shows the anomalies for the 280 ppmv $pCO_2$ experiments and the right column shows the anomalies for the 400 ppmv $pCO_2$ experiments. The 1$^{st}$ and the 99$^{th}$ percentiles of the anomalies are shown above each sub-figure. The climatology years are noted in Table 2.





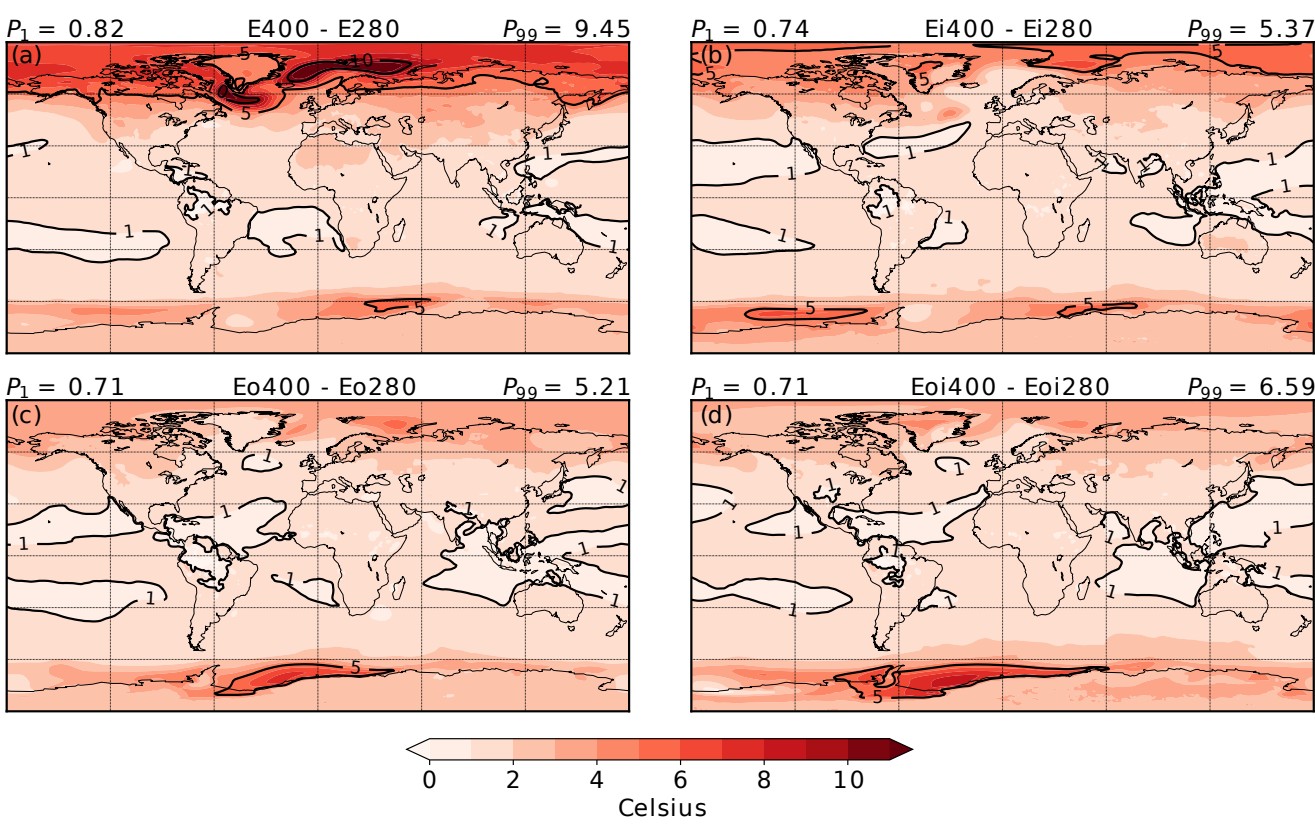

**Figure 6.** Climatological MASAT anomalies from a change to $p$CO$_2$ shown for all four combinations through which it can be deduced within our set of simulations. The 1st and the 99th percentiles of the anomalies are shown above each sub-figure. The climatology years are noted in Table 2.





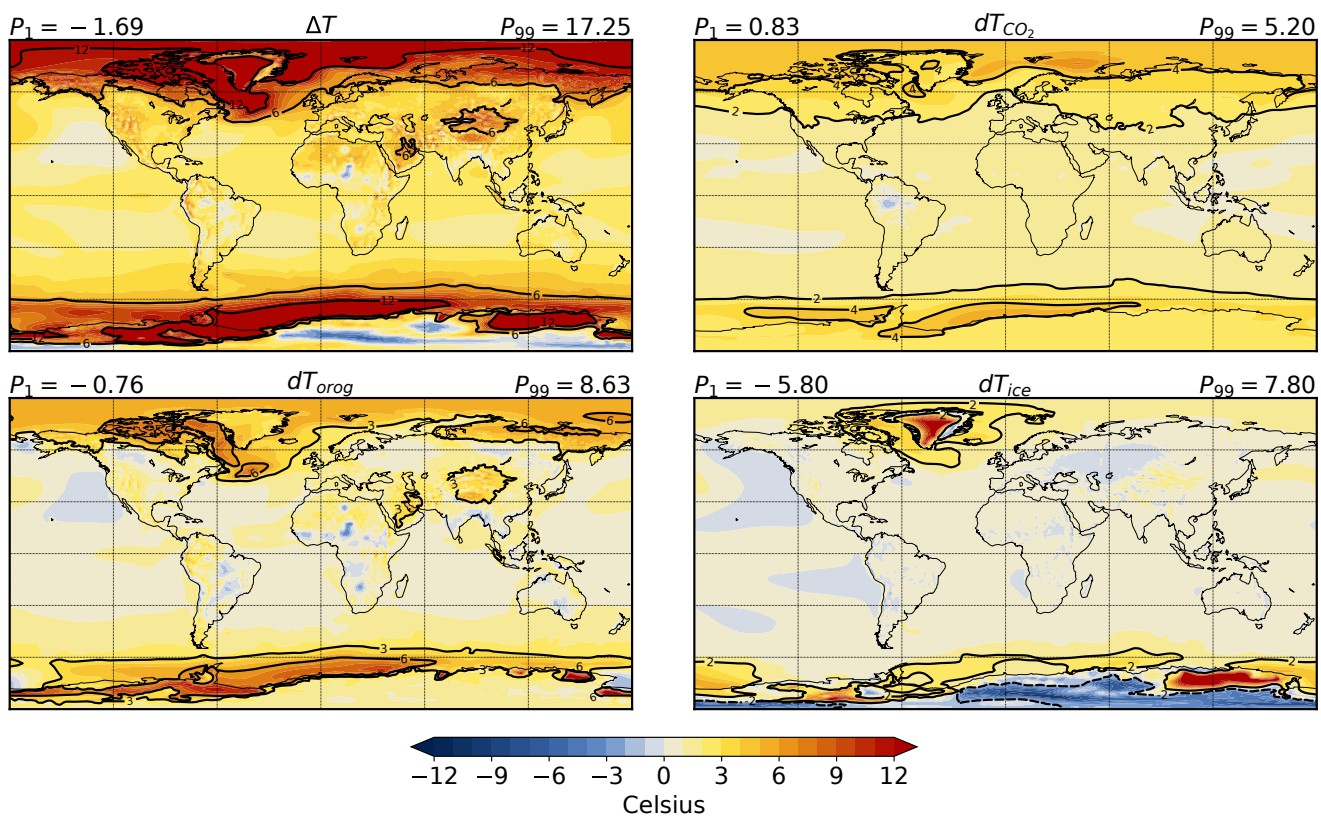

**Figure 7.** MASAT anomaly between the mid-Pliocene and the PI ($\Delta T$) and its factorization into contributions from change in atmospheric $pCO_2$ ($dT_{CO_2}$), change in orography ($dT_{orog}$) and change in ice sheets ($dT_{ice}$) using Eqs. (2). The 1st and the 99th percentiles of the quantities plotted are shown above each sub-figure.





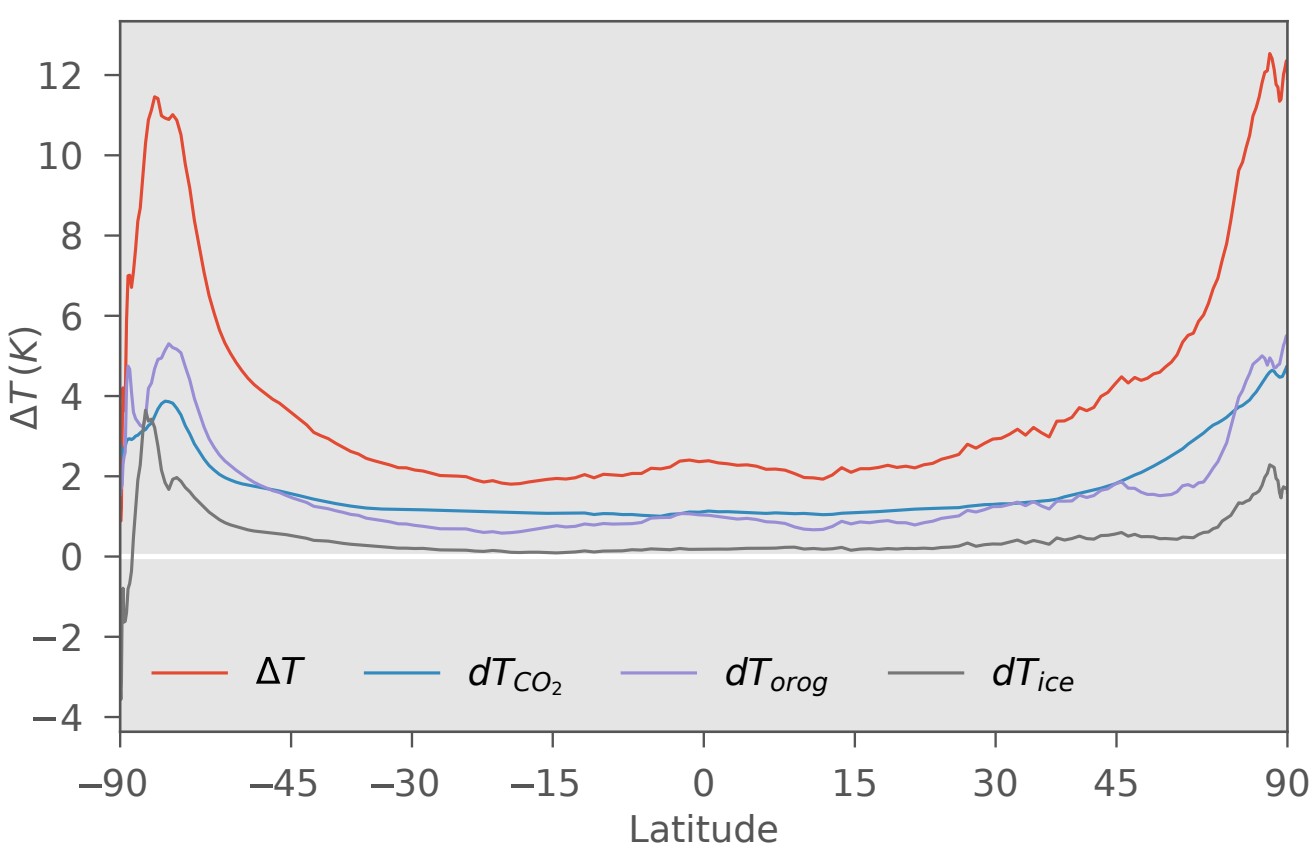

**Figure 8.** Zonal mean of MASAT anomalies shown in Figure 7. The abscissa is linear in the sine of latitude.



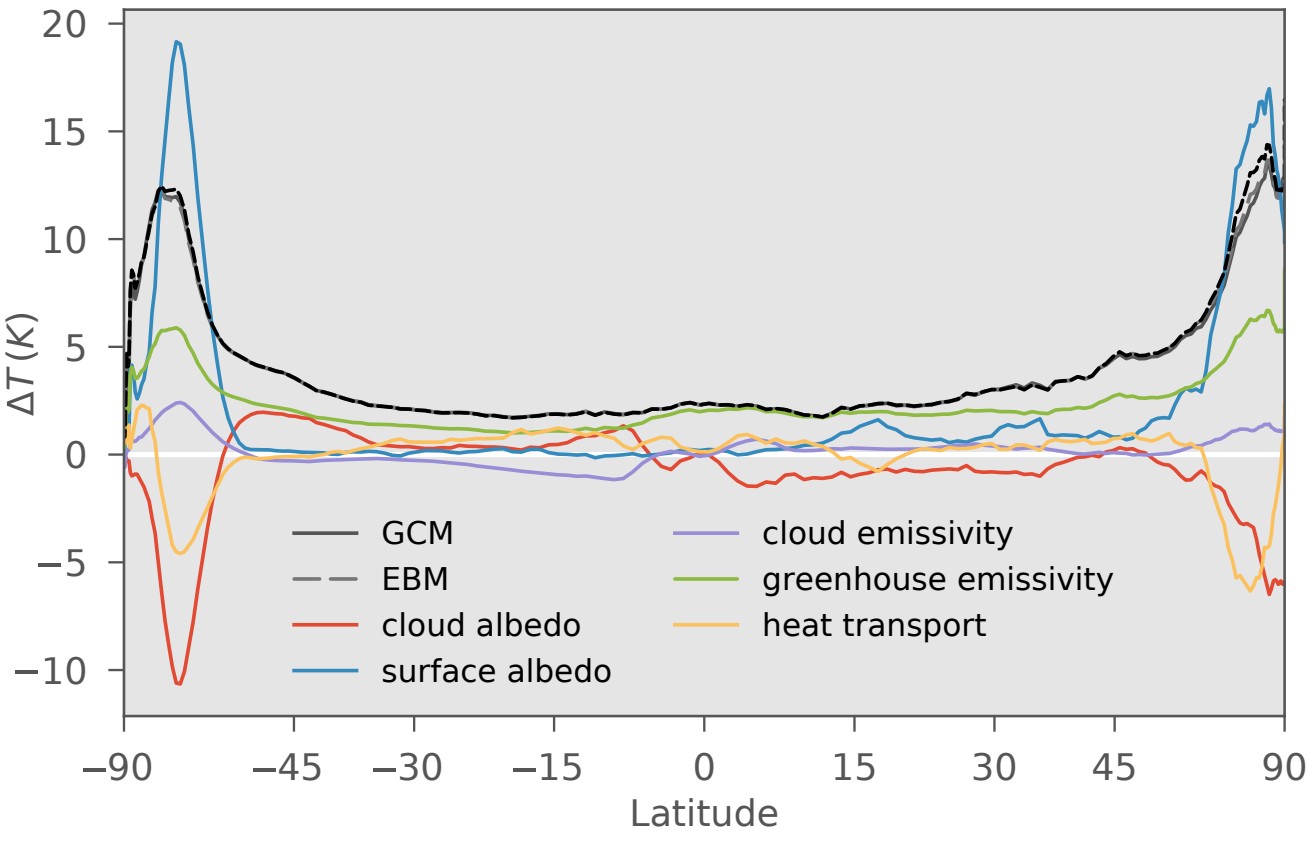

**Figure 9.** Factorization of the zonal mean mid-Pliocene minus PI temperature anomaly computed using the 1-D EBM (dashed gray) into contributions arising from changes to cloud albedo (red), cloud emissivity (purple), surface albedo (blue), greenhouse gas concentrations (including water vapor; green) and the total meridional heat transport (yellow). The solid gray line is the mid-Pliocene minus PI temperature anomaly computed directly from the GCM output. The dashed black line is the sum of the aforementioned individual contributions and is seen to almost perfectly overlap the total anomaly computed from both the EBM and the GCM. The abscissa is linear in the sine of latitude.





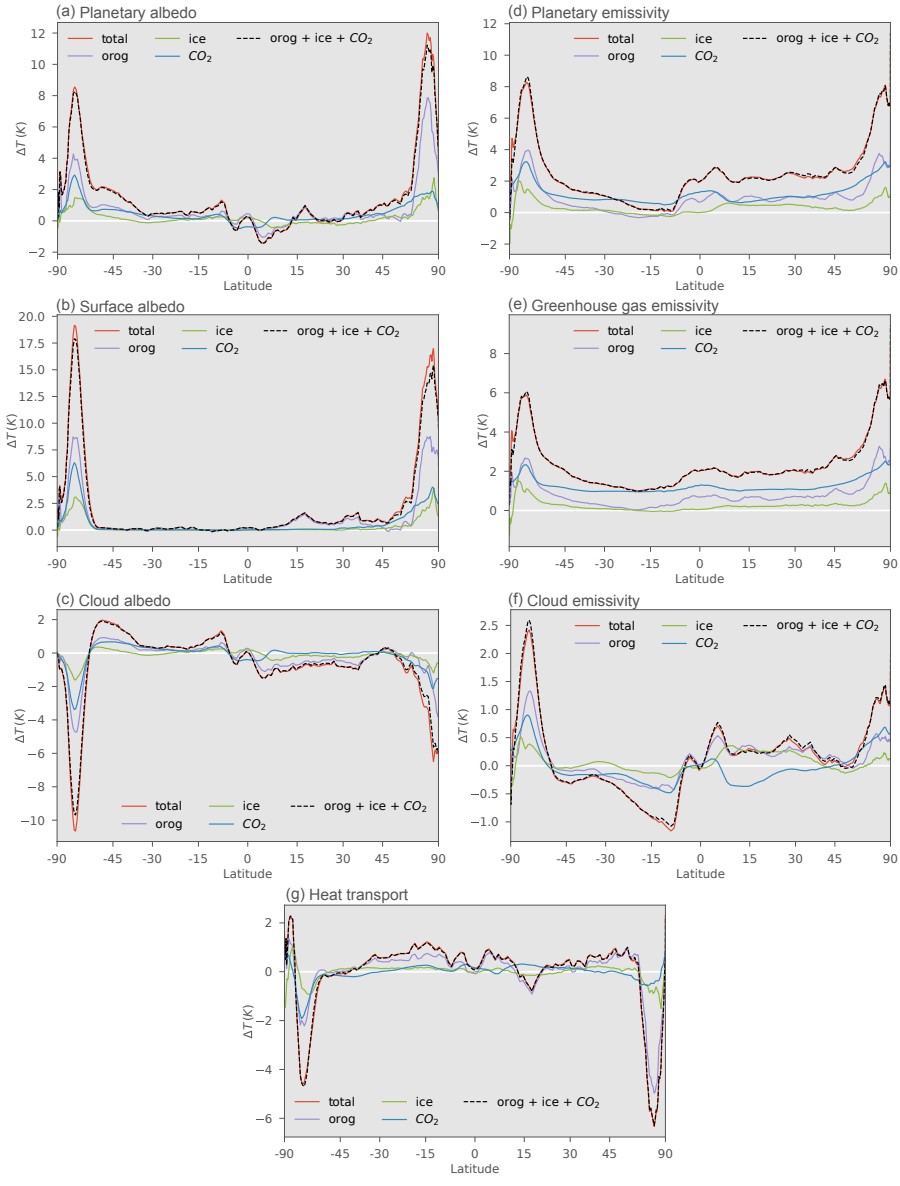

**Figure 10.** Factorization of the PI to mid-Pliocene change in zonal mean temperature (red curve) due to change in (a) planetary albedo, (b) clear-sky or surface albedo, (c) cloud albedo, (d) planetary emissivity, (e) clear-sky emissivity, or emissivity change due to change in greenhouse gas concentrations including water vapor, and (f) cloud emissivity and (g) meridional heat transport , into contributions arising from changes made to the following boundary conditions: orography (purple), ice sheets (green) and atmospheric $p\mathrm{CO_2}$ (blue). The abscissa is linear in the sine of latitude.



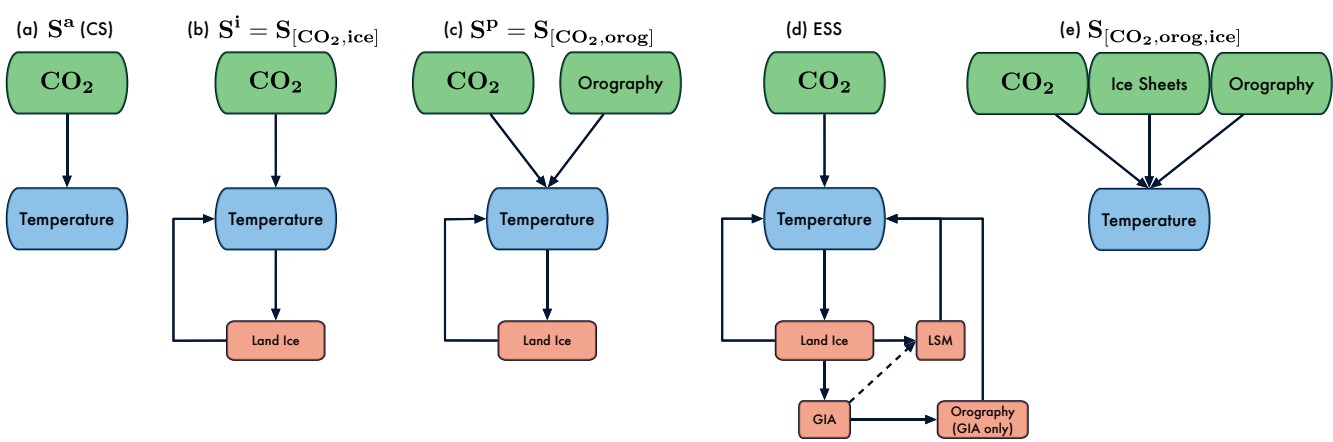

**Figure 11.** Schematic representation of various measures of climate sensitivity. See main-text for details.





**Figure 12.** Climatological sea ice concentration anomalies of the five sensitivity experiments and the 400 ppmv mid-Pliocene experiment with respect to controls. The control experiment is chosen such that the $pCO_2$ is identical between the experiment and the control. The top row shows the anomalies for 280 ppmv $pCO_2$ experiments and the bottom row shows the anomalies for 400 ppmv $pCO_2$ experiments. The climatology years are noted in Table 2.





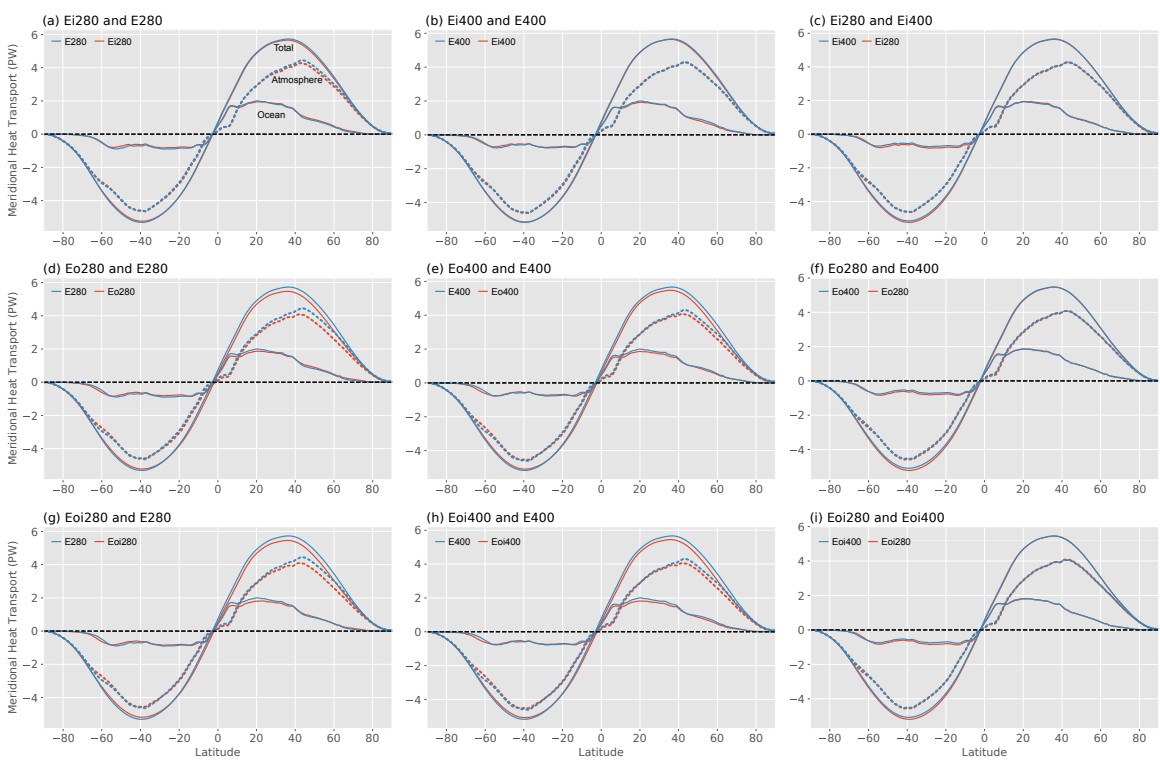

**Figure 13.** Various comparisons of the climatological total meridional heat transport as well as the individual transports by the atmosphere and the ocean. The climatology years are noted in Table 2.



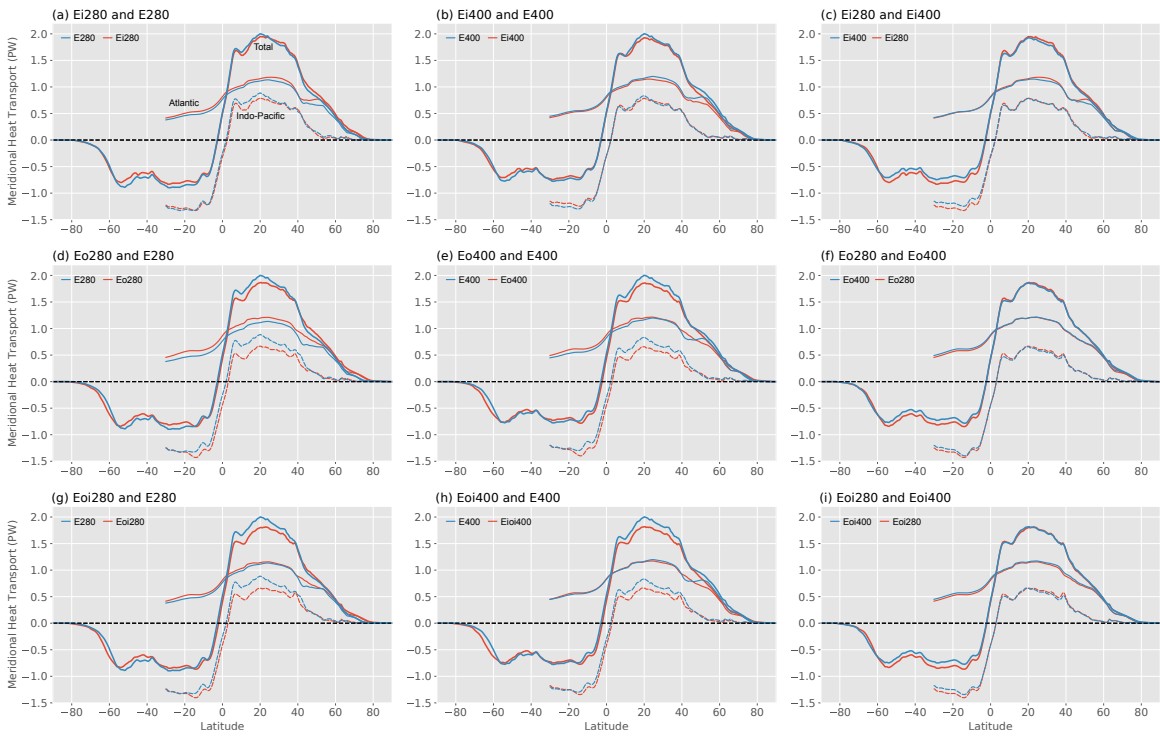

**Figure 14.** Various comparisons of the climatological total meridional heat transport by the ocean as well as the individual transports in the Atlantic and the Indo-Pacific basins. The total oceanic heat transport curves are the same curves that are labeled 'Ocean' and shown in Fig. 13. The climatology years are noted in Table 2.





**Figure 15.** Meridional heat transport anomalies (mid-Pliocene minus PI) and their factorization. (a) total meridional heat transport, (b) atmospheric heat transport, (c) oceanic heat transport. The dashed black curve in each figure is the sum of the factorized components.

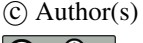



**Table 1.** Boundary conditions for the control, Pliocene and sensitivity experiments. The orbital parameters are the solar constant $S_0$, eccentricity $e$, obliquity $\epsilon$ and the longitude of perihelion $\tilde{\omega}$.

| Expt. | Topography[†] | Land[†] | Ice sheet | $CO_2$ ppmv | $CH_4$ ppbv | $N_2O$ ppbv | CFCs ppbv | $O_3$ | $S_0$ $Wm^{-2}$ | $e$ | $\epsilon$ | $\tilde{\omega}$ |
|---|---|---|---|---|---|---|---|---|---|---|---|---|
| $E^{280}$ | Modern | Modern | Modern | 280 | 760 | 270 | 0 | Modern | 1,365 | 0.016724 | 23.446° | 102.04° |
| $E^{400}$ | Modern | Modern | Modern | 400 | 760 | 270 | 0 | Modern | 1,365 | 0.016724 | 23.446° | 102.04° |
| $Eoi^{280}$ | Pliocene | Pliocene | Pliocene | 280 | 760 | 270 | 0 | Modern | 1,365 | 0.016724 | 23.446° | 102.04° |
| $Eoi^{400}$ | Pliocene | Pliocene | Pliocene | 400 | 760 | 270 | 0 | Modern | 1,365 | 0.016724 | 23.446° | 102.04° |
| $Eo^{280}$ | Pliocene | Pliocene | Modern | 280 | 760 | 270 | 0 | Modern | 1,365 | 0.016724 | 23.446° | 102.04° |
| $Eo^{400}$ | Pliocene | Pliocene | Modern | 400 | 760 | 270 | 0 | Modern | 1,365 | 0.016724 | 23.446° | 102.04° |
| $Ei^{280}$ | Modern | Modern | Pliocene | 280 | 760 | 270 | 0 | Modern | 1,365 | 0.016724 | 23.446° | 102.04° |
| $Ei^{400}$ | Modern | Modern | Pliocene | 400 | 760 | 270 | 0 | Modern | 1,365 | 0.016724 | 23.446° | 102.04° |

†. Includes orography, bathymetry and LSM
†. Includes vegetation, soil and wetland

**Table 2.** Climatology based diagnostics for the models considered in this paper.

| Expt. | Length (yr) | Climatology (yr) | TOA $(Wm^{-2})$ | AMOC (Sv) | MASAT (°C) | SST (°C) | Sea ice area ($\times 10^6 km^2$) NH | SH | Sea ice volume ($\times 10^3 km^3$) NH | SH |
|---|---|---|---|---|---|---|---|---|---|---|
| $E^{280\,†}$ | 5,200 | 5,131 – 5,160 | 0.11 | 21.5 | 13.0 | 17.9 | 13.6 | 16.2 | 55.4 | 24.0 |
| $E^{400\,†}$ | 2,010 | 1,931 – 1,960 | 0.17 | 24.2 | 15.0 | 19.1 | 9.6 | 11.9 | 16.2 | 14.2 |
| $Eoi^{280\,†}$ | 1,305 | 1,271 – 1,300 | 0.10 | 22.2 | 15.3 | 19.3 | 6.2 | 10.6 | 5.9 | 13.9 |
| $Eoi^{400\,†}$ | 2,820 | 2,691 – 2,720 | 0.11 | 23.4 | 16.8 | 20.5 | 5.1 | 6.0 | 3.8 | 5.3 |
| $Eoi^{450}$ | 2,780 | 2,611 – 2,640 | 0.10 | 23.7 | 17.3 | 20.9 | 4.6 | 5.3 | 3.0 | 4.2 |
| $Eo^{280}$ | 950 | 921 – 950 | 0.03 | 23.3 | 15.1 | 19.3 | 6.4 | 11.3 | 6.9 | 14.5 |
| $Eo^{400}$ | 1,335 | 1,301 – 1,330 | 0.07 | 23.3 | 16.6 | 20.4 | 5.1 | 7.4 | 4.0 | 7.2 |
| $Ei^{280}$ | 960 | 931 – 960 | 0.06 | 23.7 | 14.0 | 18.5 | 11.0 | 12.5 | 29.6 | 15.1 |
| $Ei^{400}$ | 1,460 | 1,431 – 1,460 | 0.16 | 23.5 | 15.7 | 19.6 | 8.9 | 8.1 | 12.4 | 7.7 |

†. From Chandan and Peltier (2017).




**Table 3.** Annual and seasonal global mean SAT anomaly between the mid-Pliocene and the PI, and the contributions to these anomalies from changes to the boundary conditions. The residual is the quantity $\Delta T - (dT_{CO_2} + dT_{orog} + dT_{ice})$ and is seen to be very small in all cases. All values are in units of $^\circ$C

|  | Annual | DJF | JJA | SON | MAM |
|---|---|---|---|---|---|
| $\Delta T$ | 3.78 | 3.18 | 4.32 | 3.72 | 3.89 |
| $dT_{CO_2}$ | 1.67 | 1.69 | 1.62 | 1.66 | 1.71 |
| $dT_{orog}$ | 1.54 | 1.00 | 2.06 | 1.48 | 1.60 |
| $dT_{ice}$ | 0.47 | 0.36 | 0.55 | 0.48 | 0.48 |
| residual | 0.10 | 0.13 | 0.09 | 0.10 | 0.10 |

**Table 4.** Various measure of climate sensitivity calculated from our experiments.

| Sensitivity | Magnitude |
|---|---|
| CS ($S^a$) | $0.88\,^\circ$KW$^{-1}$m$^2$ ($3.25\,^\circ$K per doubling $p$CO$_2$) |
| $S^i$ | $1.12\,^\circ$KW$^{-1}$m$^2$ ($4.16\,^\circ$K per doubling $p$CO$_2$) |
| $S_{[CO_2,OR]}$ ($S^p$) | $1.04\,^\circ$KW$^{-1}$m$^2$ ($3.84\,^\circ$K per doubling $p$CO$_2$) |
| $S_{[CO_2,OR,LI]}$ | $0.90\,^\circ$KW$^{-1}$m$^2$ ($3.35\,^\circ$K per doubling $p$CO$_2$) |
| ESS | $2\,^\circ$KW$^{-1}$m$^2$ ($7\,^\circ$K per doubling $p$CO$_2$) |

**Table 5.** Estimates of the contributions to the mid-Pliocene minus PI temperature difference from various mechanisms. All values are in units of $^\circ$C. The last row in the column indicating the total contribution of a mechanism is calculated from the decomposition shown in Fig. 9 and therefore may differ slightly from the sum of the contribution from ice sheets, orography and CO$_2$.

|  | EBM | | | | | | | | GCM |
|---|---|---|---|---|---|---|---|---|---|
|  | Planetary Albedo | | | Planetary Emissivity | | | Heat transport ($dT_H$) | Total |  |
|  | Cloud ($dT_{swc}$) | Surface ($dT_{salb}$) | Total ($dT_{alb}$) | Cloud ($dT_{lwc}$) | Greenhouse ($dT_{gg}$) | Total ($dT_{emiss}$) |  |  |  |
| Ice sheets | -0.13 | 0.32 | 0.19 | 0.07 | 0.26 | 0.32 | -0.01 | 0.50 | 0.47 |
| Orography | -0.31 | 1.06 | 0.75 | 0.12 | 0.79 | 0.91 | 0 | 1.66 | 1.54 |
| CO$_2$ | -0.14 | 0.59 | 0.46 | -0.04 | 1.25 | 1.21 | -0.03 | 1.64 | 1.67 |
| Total[†] | -0.65 | 2.12 | 1.47 | 0.13 | 2.30 | 2.43 | -0.03 | 3.80 | 3.80 |

†. The total row is not the sum of the rows above it