# Peer review of "On the mechanisms of warming the mid-Pliocene and the inference of a hierarchy of climate sensitivities with relevance to the understanding of climate futures"

_Climate of the Past, 2018_

## Referee Comment (RC1) · Anonymous Referee #1 · 6 Apr 2018

General Comments

The authors present the results of a modeling study evaluating the mechanisms that may have sustained Pliocene warmth and how these mechanism affect climate sensitivity on several timescales. Their effort to diagnose the mechanisms supporting the polar amplified warming within their simulation is impressively comprehensive. The factorization technique based on eight sensitivity simulations is used to diagnose the relative contribution of changes in orography, ice sheets and CO2, and a 1-D energy balance model based analysis is used to diagnose the specific mechanism

(albedo/emissivity/heat transport). The authors find that CO2 and the water vapor feed-back are responsible for around 2 thirds of their Pliocene warming and albedo changes the remaining third. The majority of the warming occurs polewards of 50N&S and sub-sequently the discussion focuses on the mechanisms behind this. The paper is clearly written particularly given its length and the breath of analysis employed.

There are however several aspects of Pliocene climate and areas of uncertainty the deserve mention and/or more discussion in the manuscript:

Specific Comments

Pg. 2 last paragraph and the subsequent climate sensitivity assessment - That aerosols and their indirect effects are assumed to have been similar to modern while in fact the proxy data currently provides little constraint on this. Some mention should be made of this caveat.

Pg. 3 third paragraph & Pg 4 second paragraph & Fig. 9 - While the mid-Pliocene simulation "reasonably accurately capture features of the proxy-inferred enhanced warming in the high-latitudes during the mid-Pliocene." it does not appear to be capturing the structure and amplitude tropical to subtropical warming as is the case for most models and also the zonal SST gradient along the equator e.g. see Fig. 1 in Brierley et al. 2015. What is amazing in Fig. 9 is just how uniform the zonal mean warming is equatorward of 45N&S and we know from the data this is not the case e.g. Dowsett et al. 2013 Figs 2 & 3 Fig OR Fig. 3b in Haywood et al. 2016. Some mention should be made of this shortcoming and the fact that the result are perhaps more relevant to high latitude Pliocene climate than tropical and subtropical Pliocene climate.

Pg. 18 lines 29-30 - This weak cloud forcing outside of the high latitudes is the reason why there is no weakening of the meridional SST gradient between the mid-latitudes and the deep tropics and the zonal SST gradient along the equator - e.g. see Fedorov et al. 2015, Burls and Fedorov 2014

Pg. 19 lines 5-7 - This result is consistent with Feng et al. (2017).

Pg. 19 lines 30-32 - Another noteworthy reference that highlights the potential importance of mixed-phase clouds is Sagoo and Storelvmo (2017)

Technical Comments

Pg. 7 end of paragraph two - Does this mean that the Eo experiments represents not only the effects associated with changing to Pliocene orography but also bathymetry? In which case changes in ocean-gateways? This needs to be clarified.

On a related note some discussion is needed of the recent paper by Otto-Bliesner et al., 2017 pointing to changes in Pliocene gateways as a mechanism supporting Arctic warmth during the Pliocene. Could this help explain some of the hemispheric asymmetry discussed in paragraph 1 on page 17?

Pg. 26 lines 17 - 18 - Any ideas of the mechanisms behind why the situation is the opposite with SH sea ice displaying a greater response under higher $CO_2$? This is an interesting result.

The second last sentence on page 30 is unclear.

References mentioned

Brierley, C., Burls, N., Ravelo, C., & Fedorov, A. (2015). Pliocene warmth and gradients. Nature Geoscience, 8(6), 419–420. http://doi.org/10.1038/ngeo2444

Haywood, A. M., Dowsett, H. J., & Dolan, A. M. (2016). Integrating geological archives and climate models for the mid-Pliocene warm period. Nature Communications, 7, 10646. http://doi.org/10.1038/ncomms10646

Dowsett, H. J., Foley, K. M., Stoll, D. K., Chandler, M. A., Sohl, L. E., Bentsen, M., et al. (2013). Sea Surface Temperature of the mid-Piacenzian Ocean: A Data-Model Comparison. Scientific Reports, 3, 2013–2018. http://doi.org/10.1038/srep02013

Dowsett, H. J., Robinson, M. M., Haywood, A. M., Hill, D. J., Dolan, A. M., Stoll, D. K., et al. (2012). Assessing confidence in Pliocene sea surface temperatures to evaluate predictive models. Nature Climate Change, 2(5), 365–371. http://doi.org/10.1038/nclimate1455

Otto-Bliesner, B. L., Jahn, A., Feng, R., Brady, E. C., Hu, A., & Löfverström, M. (2017). Amplified North Atlantic warming in the late Pliocene by changes in Arctic gateways, 44(2), 957–964. http://doi.org/10.1002/2016GL071805

Fedorov, A. V., Burls, N. J., Lawrence, K. T., & Peterson, L. C. (2015). Tightly linked zonal and meridional sea surface temperature gradients over the past five million years. Nature Geoscience, 8(12), 975–980. http://doi.org/10.1038/ngeo2577

Burls, N. J., & Fedorov, A. V. (2014). Simulating Pliocene warmth and a permanent El Niño‐like state: The role of cloud albedo. Paleoceanography, 29(10), 893–910. http://doi.org/10.1002/2014PA002644

Feng, R., Otto-Bliesner, B. L., Fletcher, T. L., Tabor, C. R., Ballantyne, A. P., & Brady, E. C. (2017). Amplified Late Pliocene terrestrial warmth in northern high latitudes from greater radiative forcing and closed Arctic Ocean gateways. Earth and Planetary Science Letters, 466, 129–138. http://doi.org/10.1016/j.epsl.2017.03.006

Sagoo, N., & Storelvmo, T. (2017). Testing the Sensitivity of Past Climates to the Indirect Effects of Dust. Geophysical Research Letters, 44(11), 5807–5817. http://doi.org/10.1002/2017GL072584

---

## Referee Comment (RC2) · Anonymous Referee #2 · 9 Apr 2018

Major comments

Although this paper presents valid results, I must say that the paper left me at times very confused. I find the paper verbose, unnecessarily long while at the same time unprecise and messy (in particular in the introduction). As a consequence the potentially interesting results do not strike out. The contribution of Pliocene climate models to estimations of long term climate sensitivity have already been explored by Lunt et al. and Haywood et al. The authors here investigate these concepts with different boundary conditions, in particular a closed Bering Strait, and add new concepts of climate sen-

sitivity over different time scales. Additionnally they explore the mechanisms leading to a warming in the UofT model in response to PRISM4 boundary conditions. These mechanisms are similar to previous studies on this topic, which makes sense.

I recommend a thorough rewriting with a focus on being concise and precise, removing all repetitions and unnecessary considerations. In particular, the introduction needs a complete re-writing in order to give the reader some perspective on what has already been done in previous studies on the mechanisms of warming in climate models (e.g. Hill et al Clim Past 2014) and the estimations of climate sensitivity/earth system sensitivity (e.g. Lunt et al., Nature Geoscience 2009 ; Haywood et al., Clim Past 2013). Then you can eventually say in which ways and with which method your study is going to fill in the gaps. The discussion as well needs rewriting, to avoid repetitions and put their results in better perspective compared to previous studies. Also, describe the results in the results section, not in the discussion (e.g. page 31, lines 1-8)

Remove the part around reversing climate in the introduction, which is very confusing and unnecessary.

About the fact that ice sheet is considered in a part of the results as a factor which has "evolved independently" from CO2 and topography changes... Well, this is completely unconceivable... The precise cause of the absence of Greenland ice sheet before the late Miocene is due to low topography, with several phases of uplift during the late Miocene and Pliocene (Solgaard et al., PPP 2013) after which some ice starts to appear, also seen in IRD records (Jansen et al 2000, Kleiven et al 2002) Then an important inception of Greenland ice sheet occurs due to decline in pCO2 at the end of the Pliocene (Lunt et al., Nature 2008), which probably took several tens of millenia according to variations in insolation (Contoux et al., 2015). Nothing mysterious there... Please remove that part, or rephrase accordingly.

The great absence in this paper is the changes to the land surface. If I understand correctly, you have imposed Pliocene vegetation and Pliocene lakes in your model, for

all experiments which includes "orography" changes. Although it is clearly visible that the PRISM4 great African lakes (MegaChad, Okavango) have an impact, locally, on air temperature (figure 5c, 5d, 7c), you never mention this in the text. Vegetation is never mentioned either, it is not even clear if it is imposed or coupled. Please say a few words about the impact of these boundary conditions as far as you can, because you can't separate them from topography changes if I understand well. This last point should also be mentioned ; your analysis of orography changes includes in fact vegetation and lakes (maybe you mentioned it and I missed it) ; so if you want to state that the changes are due to topography only, you first have to demonstrate that the changes due to vegetation&lakes are of second order.

Specific comments

1/ Abstract, 1rst sentence : the authors state that the Pliocene climate has "supported the same magnitude of global warmth as that which has been projected for the climate at the end of the 21rst century when pCO2". This sentence gives the impression that the Pliocene was a period during which temperatures have risen, which is not true, as the authors explain later in the introduction. Please rephrase accordingly, stating something like "during which temperatures were $2°$ to $3°C$ higher than the preindustrial, which is of the same magnitude as the warmth expected for the end of the 21st century" 2nd sentence : "these mechanisms explore changes to the ..." ? This formulation is not understandable. You can explore the mechanisms of change ; but can the mechanisms explore the changes ? Did you mean "these mechanisms explain changes" ? In which case, explain what mechanisms you're talking about, otherwise it's really mysterious.

2/ Page 2, line 1 : "the mid-Pliocene warm period ($\sim$3.3 Mya)". The mid-Pliocene warm period is not centered around 3.3 Ma which is precisely the date of the MIS M2 glaciation. The mid-Pliocene warm period has been defined in PlioMIP1 and 2 as lasting from roughly 3.3 to 3 Ma. The specific "time slice" of PlioMIP2 is the warm MIS KM5c interval at 3.205 Ma, not the glacial MIS M2 at 3.3 Ma.

3/ Page 2, line 24 : "the onset of Pleistocene glacial-interglacial cycles ∼1Ma". The onset of glacial-interglacial cycles is the beginning of the Pleistocene at 2.6 Ma. The transition from 40-kyr glacial-interglacial cycles to 100-kyr glacial interglacial cycles occurs at roughly 1 Ma.

4/ Page 2, last paragraph : from then on the introduction was difficult to understand, feeling more like the authors are listening to themselves, while lacking to explain in brief words the different concepts of climate sensitivity (among others). I have rarely seen an introduction with so few references. I have read about 10 times this last paragraph on page 2 and beginning of page 3 until line 10, but it can't seem to make sense to me given that the climate system, and in particular ice sheets have hysteresis trajectories and multiple equilibria (see eg Abe-Ouchi et al Nature 2013).

5/Page 3, line 33 : "what are the mechanisms through which the Pliocene warms ?" You cannot say that because the Pliocene doesn't warm. Replace by : "What are the mechanisms through which the UofT model warms in response the mid-Pliocene boundary conditions ?" (replace also accordingly in the discussion or, better, avoid the repetition of the questions...)

6/Page 4, first paragraph : this can be fixed in one sentence : mid-Pliocene climate is considered to be in equilibrium with its forcings. Second paragraph : please remove. This paragraph is unnecessary, it just repeats that your Pliocene simulation fits with the data, but you have already said that page 3, line 21.

7/ Page 4, model description ; The resolution of the model (atmosphere and ocean) should be mentioned in the model description. Is vegetation imposed or fully coupled ? Although you say here that all components are fully coupled, in your previous paper your state that the dynamical vegetation as turned off. Please precise, this is very important. Has the model been validated for the preindustrial ? There should be some reference for the validation of the UofT version of the model for the modern climates (and eventually other climates ? LGM, Mid-Holocene...)

[Figure]

8/ Page 7, lines 25 to 30 : these two very long sentences can be removed, or replaced by "to save computer resource" in the next sentence.

9/Page 9, lines 18-19 "due to power dependence... would require eight additional simulations, .... 24 additional simulations". You already said that page 8 line 21, and earlier by stating you need $2^N$ simulations.

10/Page 15, lines 10-19 : this is unnecessary to my opinion.

11/Page 18, line 6 "sum to 3.87°C" I wonder where you take that number from, as I have not managed to calculate this from numbers in Table 5. Lines 19-20 "the other 46% comes from the increase in atmospheric water vapour as a result of changes in topography and ice sheet" This is the result of your EBM and is certainly true. However, I really wonder how changes in topography and ice sheet can lead to an increase in atmospheric water vapor, except via the fact that they cause a warming and thus lead to more evaporation ? Am I right ? Can you explain in the discussion please ? Also, the changes in sea level (included in topography) and ice sheet were ultimately caused by CO2 increase, so even if water vapor effect is important following ice sheet and topography changes, some of these changes were intially caused by CO2 increase. I think it is important to be very precise here because a lot of people read our papers even outside the scientific community.

12/ Page 23, line 31. Here the authors cite Pagani et al 2010 as a reference for "observationally-derived ESS estimates". Pagani et al do not have "observations" of Pliocene temperatures, they have proxies. Please replace by "proxy-derived ESS estimates" or another formulation

Good luck.

---

## Author Comment (AC1) · 4 May 2018

**Response to Reviewer 1**

We are very grateful to the reviewer for the time he/she has taken to review our manuscript. The comments and suggestions have enabled us to further improve the clarity of the manuscript in such a way as to make the work more accessible to our readers.

> Pg. 2 last paragraph and the subsequent climate sensitivity assessment — that aerosols and their indirect effects are assumed to have been similar to modern while in fact the proxy data currently provides little constraint on this. Some mention should be made of this caveat.

We thank the reviewer for this comment and have added an appropriate remark regarding aerosols in the revised version of the manuscript.

> Pg. 3 third paragraph & Pg 4 second paragraph & Fig. 9 — While the mid-Pliocene simulation "reasonably accurately capture features of the proxy-inferred enhanced warming in the high-latitudes during the mid-Pliocene." it does not appear to be capturing the structure and amplitude tropical to subtropical warming as is the case for most models and also the zonal SST gradient along the equator e.g. see Fig. 1 in *Brierley et al.* [2015]. What is amazing in Fig. 9 is just how uniform the zonal mean warming is equatorward of 45N&S and we know from the data this is not the case e.g. *Dowsett et al.* [2013] Figs 2 & 3 Fig OR Fig. 3b in *Haywood et al.* [2016]. Some mention should be made of this shortcoming and the fact that the result are perhaps more relevant to high latitude Pliocene climate than tropical and subtropical Pliocene climate.

We thank the reviewer for this very pertinent comment. In our previous paper *Chandan and Peltier* [2017], when discussing data-model comparison we should have specifically noted the shortcoming that our simulated results have in common with results from the initial phase of the PlioMIP program in that the model is unable to simulate the proxy-inferred negative SST anomalies at tropical sites characterized by strong equatorial divergence, and at Mediterranean and Atlantic coastal sites. This remaining issue was regrettably overlooked. Although our current paper does not deal with data-model comparison, we have inserted a brief comment in the introduction of the revised manuscript regarding this shortcoming of our results.

> Pg. 18 lines 29–30 — This weak cloud forcing outside of the high latitudes is the reason why there is no weakening of the meridional SST gradient between the mid-latitudes and the deep tropics and the zonal SST gradient along the equator, e.g. see *Fedorov et al.* [2015], *Burls and Fedorov* [2014]

We have added a comment in the revised manuscript regarding the findings of *Fedorov et al.* [2015], *Burls and Fedorov* [2014] which show that the meridional cloud albedo exerts a direct control on the zonal SST gradient, and which could be applicable to the mid-Pliocene, with the

caveat that the question of the existence of a reduced zonal SST gradient along the equator during the mid-Pliocene is currently debatable. This importance of cloud albedo has also been discussed in some detail in the analyses of *Yang, Peltier and Hu* [2016] in the context of an effort to determine the dependence of the zonal tropic SST gradient upon the atmospheric $CO_2$ concentration.

> Pg. 19 lines 5–7 — This result is consistent with *Feng et al.* [2017].

We thank the reviewer for bringing this reference to our attention and have included it in the revised version of the manuscript.

> Pg. 19 lines 30–32 — Another noteworthy reference that highlights the potential importance of mixed-phase clouds is *Sagoo and Storelvmo* [2017]

We also thank the reviewer for bringing this reference to our attention, and have also included reference to it in the revised version of the manuscript.

> Pg. 7 end of paragraph two — Does this mean that the E*o* experiments represents not only the effects associated with changing to Pliocene orography but also bathymetry? In which case changes in ocean-gateways? This needs to be clarified.
>
> On a related note some discussion is needed of the recent paper by *Otto-Bliesner et al.* [2017] pointing to changes in Pliocene gateways as a mechanism supporting Arctic warmth during the Pliocene. Could this help explain some of the hemispheric asymmetry discussed in paragraph 1 on page 17?

1. Yes, the E*o* experiments include all non-ice sheet "physical" changes to the planet's surface. We do mention in the second paragraph on page 7 of the original version of the manuscript that there are also changes to bathymetry, vegetation and river routing in this configuration. What we have done in the revised manuscript is to add a further comment concerning this in section 2.2.2 where the notation used to refer to our experiments is first introduced.

2. *Otto-Bliesner et al.* [2017] do not discuss the impact of Pliocene gateways on Arctic warmth; rather they discuss the impact on warming in the North Atlantic region. Although there would be other impacts in the high latitudes of the NH from the closure of the gateways, those impacts are not explicitly presented in their paper. We have discussed their findings in our previous paper *Chandan and Peltier* [2017] when explicitly performing data-model comparisons, specifically, we've noted that the significantly improved data-model comparison that we have obtained for the high latitudes could in fact be partly due to the closure of the Arctic oceanic gateways. In this manuscript however, we have not included this reference because there is no discussion in this paper to which their findings are directly applicable.

    With regards to whether the asymmetry could be understood through the results of *Otto-Bliesner et al.* [2017], it is important to be reminded that the asymmetry being discussed

is in the surface air temperature residual $\Delta T - (dT_{CO_2} + dT_{orog} + dT_{ice})$. Since this is a residual it is therefore difficult to say which boundary condition(s), or which higher-order interaction between boundary conditions is responsible for the asymmetry (specifically, for the excess positive temperatures in the North Atlantic). It is more natural to expect that the origin for this could be in the asymmetry in the physical meridional transport of heat and in the deep water formation process (which is what we have done in the original manuscript).

> Pg. 26 lines 17–18 — Any ideas of the mechanisms behind why the situation is the opposite with SH sea ice displaying a greater response under higher $CO_2$? This is an interesting result.

We also think that this is interesting. At this point, however, and in the absence of the further analyses that will be required to provide a definitive explanation, we prefer not to speculate.

> The second last sentence on page 30 is unclear.

The second last sentence on page 30 currently reads "The higher $pCO_2$ is found to contribute 33% to the warming, followed by ice sheet contribution at 13%". Since the paragraph in which this sentence appears is summarizing the warming induced by planetary albedo change, and the contributions of the boundary condition changes to this warming, we have revised this sentence slightly to make it clearer as: "The contributions from higher $pCO_2$ and ice sheet changes to the planetary albedo change induced warming are found to be 33% and 13% respectively."

**References**

Brierley, C.M., Burls, N.J., Ravelo, A.C., Fedorov, A.V., 2015. Pliocene warmth and gradients. Nature Geosci 8, 419–420. doi:10.1038/ngeo2444

Burls, N.J., Fedorov, A.V., 2014. Simulating Pliocene warmth and a permanent El Niño-like state: The role of cloud albedo 1–18. doi:10.1002/(ISSN)1944-9186

Chandan, D., Peltier, W.R., 2017. Regional and global climate for the mid-Pliocene using the University of Toronto version of CCSM4 and PlioMIP2 boundary conditions. Clim. Past 13, 919–942. doi:10.5194/cp-13-919-2017

Dowsett, H.J., Foley, K.M., Stoll, D.K., Chandler, M.A., Sohl, L.E., Bentsen, M., Otto-Bliesner, B.L., Bragg, F.J., Chan, W.-L., Contoux, C., Dolan, A.M., Haywood, A.M., Jonas, J.A., Jost, A., Kamae, Y., Lohmann, G., Lunt, D.J., Nisancioglu, K.H., Ramstein, G., Abe-Ouchi, A., Riesselman, C.R., Robinson, M.M., Rosenbloom, N.A., Salzmann, U., Stepanek, C., Strother, S.L., Ueda, H., Yan, Q., Zhang, Z., 2013. Sea Surface Temperature of the mid-Piacenzian Ocean: A Data-Model Comparison. Sci. Rep. 3. doi:10.1038/srep02013

Dowsett, H.J., Robinson, M.M., Haywood, A.M., Hill, D.J., Dolan, A.M., Stoll, D.K., Chan, W.-L., Abe-Ouchi, A., Chandler, M.A., Rosenbloom, N.A., Otto-Bliesner, B.L., Bragg, F.J., Lunt, D.J., Foley, K.M., Riesselman, C.R., 2012. Assessing confidence in Pliocene sea surface temperatures to evaluate predictive models. Nature Climate Change 2, 365–371. doi:10.1038/nclimate1455

Fedorov, A.V., Burls, N.J., Lawrence, K.T., Peterson, L.C., 2015. Tightly linked zonal and meridional sea surface temperature gradients over the past five million years. Nature Geosci 8, 975–980. doi:10.1038/ngeo2577

Feng, R., Otto-Bliesner, B.L., Fletcher, T.L., Tabor, C.R., Ballantyne, A.P, Brady, E.C., 2017. Amplified Late Pliocene terrestrial warmth in northern high latitudes from greater radiative forcing and closed Arctic Ocean gateways. Earth and Planetary Science Letters 466, 129–138. doi:10.1016/j.epsl.2017.03.006

Haywood, A.M., Dowsett, H.J., Dolan, A.M., 2016. Integrating geological archives and climate models for the mid-Pliocene warm period. Nature Communications 6, 1–14. doi:10.1038/ncomms10646

Otto-Bliesner, B.L., Jahn, A., Feng, R., Brady, E.C., Hu, A., Löfverström, M., 2017. Amplified North Atlantic warming in the late Pliocene by changes in Arctic gateways. Geophys. Res. Lett. 44, 957–964. doi:10.1002/2016GL071805

Sagoo, N., and Storelvmo, T., 2017. Testing the Sensitivity of Past Climates to the Indirect Effects of Dust. Geophys. Res. Lett., 44(11), 5807–5817. http://doi.org/10.1002/2017GL072584

Yang, J., Peltier, W.R., Hu, Y., 2016. Monotonic decrease of the zonal SST gradient of the equatorial Pacific as a function of $CO_2$ concentration in CCSM3 and CCSM4Monotonic decrease of the zonal SST gradient of the equatorial Pacific as a function of $CO_2$ concentration in CCSM3 and CCSM4. J. Geophys. Res. Atmos. 1–17. doi:10.1002/(ISSN)2169-8996

---

## Author Comment (AC2) · 4 May 2018

**Response to Reviewer 2**

We are very grateful to the reviewer for the time he/she has taken to review our manuscript. The comments and suggestions have enabled us to further improve the clarity of the manuscript in a way that will make our work more accessible to our readers. We have categorized the reviewer's comments and our responses to those comments into three sections below.

**Regarding Clarity**

> Although this paper presents valid results, I must say that the paper left me at times very confused. I find the paper verbose, unnecessarily long while at the same time un-precise and messy (in particular in the introduction). As a consequence the potentially interesting results do not strike out.
>
> I recommend a thorough rewriting with a focus on being concise and precise, removing all repetitions and unnecessary considerations.

The reviewer has commented that there are "repetitions" and "unnecessary considerations" in our manuscript. However, aside from a very small number of specific and useful remarks (which we've addressed below) no specific instances of repetition or unnecessary considerations have been noted, in the absence of which it is impossible for us to appropriately respond. For this reason, the reviewer's suggestion of a "through rewriting with a focus on being concise and precise" is not particularly helpful. In contrast, the first reviewer has characterized our effort as "impressively comprehensive" and also explicitly stated that "the paper is clearly written particularly given its length and the breadth of analysis employed".

If the reviewer were able to provide us with specific instances in which our written text includes repetitions and unnecessary considerations we would be pleased to address them. As it stands and given the direct contradiction of this reviewer's general comment concerning "style" by the first reviewer we are at a loss as to how to respond. Could this be an issue that has arisen because of differences of mother tongue? The important thing is that the science presented in our paper is relevant, correct, and timely and it would seem that on these matters the reviewer agrees with us.

**Regarding the Introduction**

> In particular, the introduction needs a complete re-writing in order to give the reader some perspective on what has already been done in previous studies on the mechanisms of warming in climate models (e.g. Hill et al Clim Past 2014) and the estimations of climate sensitivity/earth system sensitivity (e.g. Lunt et al., Nature Geoscience 2009 ; Haywood et al., Clim Past 2013). Then you can eventually say in which ways and with which method your study is going to fill in the gaps. The discussion as well needs rewriting, to avoid

repetitions and put their results in better perspective compared to previous studies. Also, describe the results in the results section, not in the discussion (e.g. page 31, lines 1–8)

We begin addressing the first of the reviewer's comments regarding the introduction by noting that **the introduction (or any section of a scientific paper for that matter) is not meant to be a straight-jacket**. The diversity in the researches that are conducted and in the scientific questions addressed naturally leads to diverse methods of presentation. The comments regarding our introduction regrettably suggests that the reviewer subscribes to a very myopic and rigid take on scientific writing. We proceed to address specific arguments made in the comment above:

1. The "perspective" that the reviewer is asking for is indeed provided, in fact in considerable detail, in the manuscript. Section 3.1 devotes four large paragraphs to discussing the factorization technique, how this method is different from other methods of analysis, the origins of the method, its employment in a diverse range of palaeoclimate studies (no less than eight references), and the modified formulation of the method being used in the PlioMIP2 program.

   The methodology used to understand the mechanisms of warming is connected to seven diverse studies at the beginning of Section 3.2 and a detailed formulation as well as our extension to the standard methodology is presented.

   The climate sensitivity section 4.4 also begins by introducing the subject matter along with various nuances in four lengthy paragraphs, and is only then followed by a comprehensive discussion of the results of our analysis.

   The references that the reviewer cites were all included in our original manuscript.

2. The reviewer has again alluded to these mysterious "repetitions" without any specifics. The "Discussion and Conclusions" section, to which the reviewer is specifically referring here, is neatly divided into comments related to the two themes of our paper (mechanisms of warming and climate sensitivity) and the three important questions that we have posed in the introduction (page 3 original manuscript). It confounds us, really, how the reviewer could take exception to such a natural and logical means of organizing this section.

3. The reviewer's comment that the discussion should be presented with the results makes no sense at all. The results are several pages long (14 pages to be sure) and contain several details of the analysis. Whereas the discussion and conclusion section provides a succinct summary of the key findings, as is standard practice in scientific writing.

Remove the part around reversing climate in the introduction, which is very confusing and unnecessary.

Page 2, last paragraph : from then on the introduction was difficult to understand, feeling more like the authors are listening to themselves, while lacking to explain in brief words the different concepts of climate sensitivity (among others). I have rarely seen an introduction with so few references. I have read about 10 times this last paragraph on page 2 and beginning of page 3 until line 10, but it can't seem to make sense to me given that the climate system, and in particular ice sheets have hysteresis trajectories and multiple

> equilibria (see eg Abe-Ouchi et al Nature 2013).

1. The sections of the introduction that the reviewer is referring to attempt to explicitly state and argue some of the considerations that are implicitly taken for granted when past warm climates are used to study the future. In fact, what we are addressing here are actually the sort of climate evolution trajectories that the reviewer him/herself seems to be focused upon. We are discussing the reasonableness of transferring a mid-Pliocene based understanding of a past warm period to the inference of the future state of the Earth's climate, given that the system has evolved from a warmer (than mid-Pliocene) to the relatively cooler mid-Pliocene state, whereas the future warmer climate will have evolved from the cooler present-day climate. We are therefore drawing attention to the fact that climate conditions at any particular time will depend upon the initial conditions from which the state of interest has evolved.

2. Concerning the comment that "I have rarely seen an introduction with so few references": It is worth repeating what we have mentioned in the previous remark that **the introduction (or any section of a scientific paper for that matter) is not meant to be a straight-jacket!** Our paper is very well referenced (again see previous comment), comprising of 75 total references connecting to literature in topics as varied as RCP scenarios and future EMICs simulations, IPCC analysis, climate sensitivity, ice sheet modelling, glacial isostatic adjustment, palaeoclimate modelling studies for the Miocene, LGM, Pliocene, Eocene, interglacials, Pliocene climate, proxy inferences of characteristics of the Pliocene climate and several modelling based analysis of the mid-Pliocene over the past decade.

3. All concepts employed are carefully defined in clearly labeled sections in our manuscript: "The Factorization Technique", "One-dimensional Energy Balance Model", "Mechanisms of Warming" and "A mid-Pliocene Perspective on Climate Sensitivity" to name a few. In addition, the introduction concludes with an outline of how the paper is organized.

> Page 4, first paragraph : this can be fixed in one sentence : mid-Pliocene climate is considered to be in equilibrium with its forcings. Second paragraph : please remove. This paragraph is unnecessary, it just repeats that your Pliocene simulation fits with the data, but you have already said that page 3, line 21.

Regarding the first paragraph: We don't agree that there is anything that requires fixing. We are simply being clear and explicit about the implicit assumptions that are most often involved in palaeo climate modelling. This can hardly be considered a flaw, especially since the reviewer him/herself argues later on that "it is important to be very precise because a lot of people read our papers even outside the scientific community".

Regarding the second paragraph: No, it doesn't "just repeats that [our] Pliocene simulation fits with the data"! In fact, this paragraph is not even about our simulation, instead the paragraph presents an argument concerning the manner in which mid-Pliocene (or for that matter, just about all palaeoclimate) reconstructions are typically conducted by starting from a pre-industrial/modern state and from it seeking an equilibrium climate that is supposed to have existed previously under different boundary conditions, a procedure that explicitly neglects the

dependence upon the initial conditions from which the state of interest has evolved. We first argue that how one ought to be performing such climate reconstructions is by incorporating changing boundary conditions. We then go on to argue that at present such an ideal procedure is ruled out as a possibility because of the inadequacy of computational resources, and we conclude therefore that the conventional way of performing palaeoclimate simulations which is to seek climate states that are in equilibrium with the boundary conditions is appropriate to the target time. We then make the very important point that because we are settling for a less-than-ideal methodology for performing the palaeo-climate reconstructions (and because we are not attempting to take into account the (perhaps sensitive) dependence of the climate of the epoch of interest upon the initial conditions from which it has actually evolved), the quality of any analysis conducted on the basis of these assumptions can only be ascertained on the basis of the quality of the fit to data based climate inferences. This is a particularly important comment for mid-Pliocene based inferences because this geologic time-period is considered a 'natural analogue' for the future and therefore part of the motivation for studying the mid-Pliocene is to transfer the understanding gained to our expectations concerning the future of our climate. It is only here, having raised this objective argument, that we assert that "we are well positioned" to perform our analysis because our mid-Pliocene simulations reported in *Chandan and Peltier* [2017] are in "very good agreement with proxy reconstructions of the climate of that time", not withstanding the issues that continue to remain in the equatorial regions between approximately 45° north and south latitudes.

**Other comments**

> The contribution of Pliocene climate models to estimations of long term climate sensitivity have already been explored by Lunt et al. and Haywood et al. The authors here investigate these concepts with different boundary conditions, in particular a closed Bering Strait, and add new concepts of climate sensitivity over different time scales. Additionally they explore the mechanisms leading to a warming in the UofT model in response to PRISM4 boundary conditions. These mechanisms are similar to previous studies on this topic, which makes sense.

We have analyzed the mechanisms involved in sustaining the mid-Pliocene warmth within our model using the most up-to-date reconstruction for the mid-Pliocene, namely the PRISM4 reconstruction. We build upon and continue the work begun by *Lunt et al.* [2012] with the PRISM2 boundary conditions and revisited by *Hill et al.* [2014] using the PRISM3 boundary conditions. The mechanisms considered in our study are the same as those considered in these previous studies (and in several palaeoclimate studies as extensively reviewed in our paper) as the theoretical formulation is the same. The difference is that we are using a climate model (the University of Toronto version of CCSM4) that has actually delivered a high quality fit to the inferred high latitude constraints when forced with the most recent reconstruction of the mid-Pliocene.

In addition, and for the first time, we explicitly break down the contributions of these mechanisms further into the influences from changes to individual boundary conditions. This has never been done before and is clearly important to the community (i.e. PlioMIP) effort to

iteratively refine the Pliocene boundary conditions. Our extension of the conventional EBM approach helps with this effort by allowing for the assessment of how the boundary condition changes impact the warming through different distinct mechanisms.

Finally, the reviewer's comment that we investigate several concepts "with different boundary conditions, in particular a closed Bering Strait" is a gross over-simplification of the differences between the PRISM3 and the PRISM4 mid-Pliocene boundary conditions. The reviewer is referred to *Dowsett et al.* [2016] for a through review of the PRISM4 reconstruction and the many ways in which it differs from the previous version.

> About the fact that ice sheet is considered in a part of the results as a factor which has "evolved independently" from $CO_2$ and topography changes... Well, this is completely unconceivable... The precise cause of the absence of Greenland ice sheet before the late Miocene is due to low topography, with several phases of uplift during the late Miocene and Pliocene (Solgaard et al., PPP 2013) after which some ice starts to appear, also seen in IRD records (Jansen et al 2000, Kleiven et al 2002) Then an important inception of Greenland ice sheet occurs due to decline in $p$CO$_2$ at the end of the Pliocene (Lunt et al., Nature 2008), which probably took several tens of millenia according to variations in insolation (Contoux et al., 2015). Nothing mysterious there... Please remove that part, or rephrase accordingly.

The reviewer's comments concern the discussion of $S_{[CO_2, OR, LI]}$ which is the last parameter we discuss in the climate sensitivity section. For the purposes of this sensitivity parameter the ice sheet configuration is considered to act as a forcing rather than as an internal feedback (which was the case for several other climate sensitivity parameters that we had considered to this point in the manuscript). The parameter $S_{[CO_2, OR, LI]}$ was previously employed by *PALAEOSENS Project Members* [2012] in their supplementary materials, but they did not comment upon its significance, leaving us to speculate if it was considered only as a curiosity. We however, think that there is a precise significance to this and which applies not only to the mid-Pliocene but also to the very distant future of our climate. We first explain the reasoning for considering ice sheets as forcings before commenting on the parameter's significance.

In the case in which the GIS and the WAIS are absent, the positive ice-albedo feedback involving these ice sheets would clearly not be operational (only once the ice sheets have begun forming does this positive feedback become active). However, in the absence of these ice sheets, local warming may be sustained through the presence of low albedo on the Earth's surface and a low orography. This warming in-turn makes the inception of these ice sheets difficult. In this manner, the ice sheet configuration acts as a forcing. This consideration is of particular significance for the GIS as it has been shown on the basis of a two-dimensional continent and ocean resolving energy balance analysis [*North et al.*, 1983, *Hyde et al.*, 1990] that within specific ranges of parameters such as insolation and $CO_2$, Greenland is ideally suited for the inception of an ice sheet based solely on its location and the difference in heat capacity between its landmass and the surrounding ocean. Consequently, as the global drawdown of $CO_2$ continued during the mid-Pliocene and $CO_2$ level started to become favourable for the inception of the GIS, the pre-existing forcing that existed in the absence of the GIS would have acted as a barrier that the system would first need to overcome. Similarly, for Antarctica *DeConto et al.* [2008] show that the $CO_2$ threshold for glaciation is $\sim 750$ ppmv, but in spite of the mid-Pliocene $CO_2$ level remaining much less than this threshold value there was only

intermittent glaciation over West Antarctica and large sub-glacial basins of East Antarctica. It could again be argued that the absence of the ice sheets in these parts of Antarctica and the ensuing warmth they helped sustain, made it difficult for the ice sheets to develop.

Therefore, during the mid-Pliocene before the inception of the GIS, the WAIS, and extension of the EAIS into the sub-glacial basins of East Antarctica (and assuming relative stability of the interior sections of the EAIS) the response of the climate system to changes in forcing would likely have been characterized by $S_{[CO_2, OR, LI]}$. Similarly, if the future radiative forcing could be maintained at a level comparable to the present day (equivalent to the mid-Pliocene), then in the very distant future, long after the feedback processes that influence the ESS have stabilized, the sensitivity of the climate system would be characterized by $S_{[CO_2, OR, LI]}$. Therefore considering the hierarchy of climate sensitivities applicable to the future, the picture that emerges is as follows: on the short timescale the sensitivity is given by $S^a = 3.25$ °$K$ per doubling of $CO_2$, this increases to $S^i = 4.16$ °$K$ per doubling of $CO_2$ on an intermediate timescale and then further increases to $ESS = 7.0$ °$K$ per doubling of $CO_2$ on long timescales. Finally, once the feedbacks that influence the ESS have stabilized, the sensitivity of the system then drops to $S_{[CO_2, OR, LI]} = 3.35$ °$K$ per doubling of $CO_2$.

We have incorporated the discussion contained in the last two paragraphs into the revised version of the manuscript to further clarify the significance of $S_{[CO_2, OR, LI]}$.

With regards to the reviewer's comments, we note first that any sensitivity parameter only applies to a specific range of Earth-system conditions, which is the logical basis for our exploration of a 'hierarchy' of climate sensitivities. $S_{[CO_2, OR, LI]}$ is concerned with the time period before the onset of large scale (and ultimately permanent) glaciations over Greenland and West Antarctic, at the time when the system was 'primed' for the formation of these ice sheets (such as having the topography and the $CO_2$ level) but in which the initiation of ice sheets would involve overcoming the inertia provided by the absence of the ice sheets. This parameter is therefore not applicable to the several million years long interval from the late Miocene to the late Pliocene and is not influenced by the events that might have taken place during that interval. Further to the reviewer's comments, we do not intend to suggest that $CO_2$ did not have any impact on the inception of the GIS and the WAIS. We have clarified these points in the revised version of the manuscript.

The reviewer has suggested that the "precise cause" of the absence of the GIS was low orography and cites *Solgaard et al.* [2013] as reference. There are several reasons why we are still very far from knowing the "precise cause" for the absence of the GIS in the late Pliocene. Firstly, the hypothesis that Miocene tectonic uplifts of Greenland were required to initiate the inception of the GIS are derived on the basis of a rather sparse data set, and therefore it can not be entirely ruled out that no such proposed uplifts took place. Secondly, to reconstruct the topography that existed before the two hypothesized Miocene episodes of uplifts, for use in an ice sheet model, *Solgaard et al.* [2013] employ uniform step-wise corrections to the isostatically adjusted Greenland topography, but they don't adequately comment upon the reliability of these corrections and, by extension, upon the errors in their supposed pre-uplift topography. Since Greenland's isostatically adjusted topography after the removal of present day ice sheet is fairly high ($\sim 800$ m in the interior and $\sim 2-3$ km along the eastern mountain range) depending on the accuracy of these corrections, one can wonder whether any supposed uplifts were needed at all for the inception of the GIS, or if the topography prior to these episodes of uplifts was still sufficiently high that the formation of the GIS was inevitable once $CO_2$ levels declined

sufficiently. Furthermore, the energy balance analyses mentioned earlier and which does not in any way consider elevation, can readily show that Greenland is capable of supporting an ice sheet in the NH as it is the only sizable landmass in the NH surrounded by the snow line during both July and January. Because of this one can ask just how significant is Greenland's elevation to the formation of an ice sheet? Do we really need very high orography, and how much is high enough? Then there is the issue of their using questionable boundary conditions (besides topography) in their simulations, including the usage of mid-Pliocene SSTs to drive their AGCM coupled ice sheet model for the case of late ($\sim$ 10 mya) Miocene. Lastly, an obvious point is that theirs is just a single study testing for a specific hypothesis among several hypotheses that have been proposed regarding the absence/initiation of the GIS. For their findings to be considered to provide the "precise cause" one would require a degree of robustness that can only be provided on the basis of some consensus emerging from several diverse models and a systematic ruling out of other hypotheses.

Page 9, lines 18–19 "due to power dependence… would require eight additional simulations, …. 24 additional simulations". You already said that page 8 line 21, and earlier by stating you need $2^N$ simulations.

This is hardly a significant critique of our work. We don't see anything negative in investing a few words to clarify a sub-point in the context of an argument.

The great absence in this paper is the changes to the land surface. If I understand correctly, you have imposed Pliocene vegetation and Pliocene lakes in your model, for all experiments which includes "orography" changes. Although it is clearly visible that the PRISM4 great African lakes (MegaChad, Okavango) have an impact, locally, on air temperature (figure 5c, 5d, 7c), you never mention this in the text. Vegetation is never mentioned either, it is not even clear if it is imposed or coupled. Please say a few words about the impact of these boundary conditions as far as you can, because you can't separate them from topography changes if I understand well. This last point should also be mentioned ; your analysis of orography changes includes in fact vegetation and lakes (maybe you mentioned it and I missed it) ; so if you want to state that the changes are due to topography only, you first have to demonstrate that the changes due to vegetation & lakes are of second order.

1. Yes, the simulations involving orography changes (E$o$ and E$oi$) include changes to the vegetation, soil and lakes as well as bathymetry. This is mentioned in Table 1 and in the main text (page 7, second paragraph) of the original manuscript. We have now expanded the comment in the main text slightly in the revised manuscript so as to make this clearer.

2. We do not want to state that the changes are due to topography only. The "o" which stands for orography in the notation used for our experiments is simply a notation (as devised by *Lunt et al.* [2012] and adopted for the PlioMIP2 program *Haywood et al.* [2016]) but the orography changes include bathymetry and land surface changes. We have revised the manuscript at the location where the nomenclature is introduced to make it clear that the "o" case includes these various other changes and that in the rest of the document

orography related changes are to be read as changes due to the collectivity of changes made to orography, bathymetry and land surface.

3. It would be very useful to point out the local impact of the African mega lakes in our manuscript. We have added appropriate remarks in the revised manuscript in the section where the surface air temperatures are discussed.

4. The reasons why land surface has not been investigated separately in this paper are because (i) the PlioMIP2 experiment protocol does not specify experiments exploring land surface as a separate factor, and (ii) because (hopefully the reviewer won't chastise us for bringing this up again!) the requirement of $2^N$ simulations for the application of the factorization method would necessitate 8 additional experiments if land surface was included as a separate factor. The omission of land surface as a separate factor is reasonable as ....

5. ... *Lunt et al.* [2012] show that at least using the PRISM2 reconstruction, the contribution from land surface changes to the mid-Pliocene warmth is considerably smaller than the contributions from changes to orography, ice sheets or $CO_2$. This was explicitly discussed in the original version of the manuscript.
* * *
Page 4, model description ; The resolution of the model (atmosphere and ocean) should be mentioned in the model description. Is vegetation imposed or fully coupled ? Although you say here that all components are fully coupled, in your previous paper your state that the dynamical vegetation as turned off. Please precise, this is very important. Has the model been validated for the preindustrial ? There should be some reference for the validation of the UofT version of the model for the modern climates (and eventually other climates ? LGM, Mid-Holocene...)
* * *
1. Mentioning the resolution of the model would be certainly useful and we have updated the manuscript accordingly.

2. We have updated the manuscript to make it clear that the dynamic vegetation option has been turned off.

3. The model's performance has been compared to HadISST pre-industrial SST and sea ice in *Chandan and Peltier* [2017]. We have not published any equilibrium LGM or mid-Holocene results with our model as these efforts are ongoing, but the model has been applied to the study of the Dansgaard-Oeshger oscillations in *Peltier and Vettoretti* [2014], *Vettoretti and Peltier* [2015, 2016, 2018].
* * *
Page 7, lines 25 to 30: these two very long sentences can be removed, or replaced by "to save computer resource" in the next sentence.
* * *
We prefer to argue more explicitly regarding the constraints imposed upon the analyses we have performed and the manner in which we have tried to circumvent these constraints.

> Page 15, lines 10-19 : this is unnecessary to my opinion.

We are clarifying here for the readers that there are several ways of differencing the set of PlioMIP2 simulations to obtain the impacts of boundary conditions. This is an important point in our opinion.

> Page 18, line 6 "sum to 3.87°C" I wonder where you take that number from, as I have not managed to calculate this from numbers in Table 5.

We thank the referee for looking carefully at the numbers in Table 5. Not only has this prompted us to clarify the provenance of this particular number in our text but it has also allowed us to find a typographical error and fix it. Firstly, the error was that the first 3.80 in the last row of Table 5 should have been 3.87 (now fixed). Secondly, this number is the sum $dT_{swc} + dT_{salb} + dT_{lwc} + dT_{gg} + dT_H$ i.e. the sum of the warming from each of the five mechanisms computed using the EBM model, and which can be obtained by adding the values under the appropriate columns in Table 5. To make this clear, we have added the above expression as a footnote in Table 5.

> Lines 19–20 "the other 46% comes from the increase in atmospheric water vapour as a result of changes in topography and ice sheet" This is the result of your EBM and is certainly true. However, I really wonder how changes in topography and ice sheet can lead to an increase in atmospheric water vapor, except via the fact that they cause a warming and thus lead to more evaporation? Am I right? Can you explain in the discussion please? Also, the changes in sea level (included in topography) and ice sheet were ultimately caused by $CO_2$ increase, so even if water vapor effect is important following ice sheet and topography changes, some of these changes were intially caused by $CO_2$ increase. I think it is important to be very precise here because a lot of people read our papers even outside the scientific community.

1. Yes, that is correct. We have modified the manuscript to mention that the orography and the ice sheet contributions to increased water vapour are mediated through the temperature changes they introduce and the Clausius-Clapeyron relationship.

2. Regarding the second comment, the understanding that the topography and ice sheet changes are ultimately caused by $CO_2$ changes is central to our manuscript and arises in several parts involving discussions of our climate sensitivity inferences. Motivated by the reviewer's comments, what we have now done is to add a comment to this effect also in the paragraph to which the reviewer is referring.

> Page 23, line 31. Here the authors cite *Pagani et al.* [2010] as a reference for "observationally-derived ESS estimates". *Pagani et al.* [2010] do not have "observations" of Pliocene temperatures, they have proxies. Please replace by "proxy-derived ESS estimates" or another

> formulation

Thank you for this useful comment, we have revised the manuscript accordingly.

> Abstract, 1st sentence : the authors state that the Pliocene climate has "supported the same magnitude of global warmth as that which has been projected for the climate at the end of the 21st century when $p\mathrm{CO_2}$". This sentence gives the impression that the Pliocene was a period during which temperatures have risen, which is not true, as the authors explain later in the introduction. Please rephrase accordingly, stating something like "during which temperatures were 2° to 3°C higher than the preindustrial, which is of the same magnitude as the warmth expected for the end of the 21st century". 2nd sentence : "these mechanisms explore changes to the ..."? This formulation is not understandable. You can explore the mechanisms of change; but can the mechanisms explore the changes? Did you mean "these mechanisms explain changes"? In which case, explain what mechanisms you're talking about, otherwise it's really mysterious.

Regarding the first sentence: We don't see how the reviewer could have read it to imply in any way regarding either an increase or decrease of temperature. The sentence is simply referring to 'supporting/sustaining' a level of warmth and not about any evolution to/from that state.

Regarding the second sentence: We quite frankly see no distinction between what we have said and what the reviewer is suggesting we could have said.

> Page 2, line 1 : "the mid-Pliocene warm period (∼3.3 Mya)". The mid-Pliocene warm period is not centered around 3.3 Ma which is precisely the date of the MIS M2 glaciation. The mid-Pliocene warm period has been defined in PlioMIP1 and 2 as lasting from roughly 3.3 to 3 Ma. The specific "time slice" of PlioMIP2 is the warm MIS KM5c interval at 3.205 Ma, not the glacial MIS M2 at 3.3 Ma.

Our original statement simply indicates an approximate time period for the mid-Pliocene. We suggest that no educated reader would associate a phrase such as "the mid-Pliocene warm period" with a single, sharp marine isotope stage. Indeed the mid-Pliocene does not seem to be specifically defined by the International Commission on Stratigraphy and while 3.0–3.3 Mya has come to be called the mid-Pliocene (mainly due to that interval having being chosen for study by the original PlioMIP program), some authors have referred to fossils as old as 3.5 Mya as belonging to the mid-Pliocene [*Rybczynski et al.*, 2013]. Furthermore, even though the PlioMIP2 program is manifestly 'focused on the mid-Pliocene' the specific interest of the project is only on the KM5c interval at 3.205 Mya! Lastly, we are well aware of the details of what the reviewer has pointed out as all of it has been mentioned previously in *Chandan and Peltier* [2017].

> Page 2, line 24 : "the onset of Pleistocene glacial-interglacial cycles ∼1Ma". The onset of glacial-interglacial cycles is the beginning of the Pleistocene at 2.6 Ma. The transition from 40-kyr glacial-interglacial cycles to 100-kyr glacial interglacial cycles occurs at roughly 1

> Ma.

We greatly appreciate the reviewer bringing this to our attention. We have modified this to say "at the onset of the 100 kyr Pleistocene glacial-interglacial cycles at $\sim 1$ Ma".

> Page 3, line 33 : "what are the mechanisms through which the Pliocene warms ?" You cannot say that because the Pliocene doesn't warm. Replace by : "What are the mechanisms through which the UofT model warms in response the mid-Pliocene boundary conditions ?" (replace also accordingly in the discussion or, better, avoid the repetition of the questions...)

We concur that there is a need for improved clarity here. We have revised the sentence to "What are the mechanisms sustaining mid-Pliocene warmth?"

> Good luck.

Thank you, and good luck to you too!

**References**

Brierley, C.M., Burls, N.J., Ravelo, A.C., Fedorov, A.V., 2015. Pliocene warmth and gradients. Nature Geosci 8, 419–420. doi:10.1038/ngeo2444

Chandan, D., Peltier, W.R., 2017. Regional and global climate for the mid-Pliocene using the University of Toronto version of CCSM4 and PlioMIP2 boundary conditions. Clim. Past 13, 919–942. doi:10.5194/cp-13-919-2017

DeConto, R.M., Pollard, D., Wilson, P.A., Pälike, H., Lear, C.H., Pagani, M., 2008. Thresholds for Cenozoic bipolar glaciation. Nature 455, 652–656. doi:10.1038/nature07337

Dowsett, H.J., Dolan, A., Rowley, D.B., Moucha, R., Forte, A.M., Mitrovica, J.X., Pound, M.J., Salzmann, U., Robinson, M.M., Chandler, M.A., Foley, K., Haywood, A.M., 2016. The PRISM4 (mid-Piacenzian) paleoenvironmental reconstruction. Clim. Past 12, 1519–1538. doi:10.5194/cp-12-1519-2016

Dowsett, H.J., Foley, K.M., Stoll, D.K., Chandler, M.A., Sohl, L.E., Bentsen, M., Otto-Bliesner, B.L., Bragg, F.J., Chan, W.-L., Contoux, C., Dolan, A.M., Haywood, A.M., Jonas, J.A., Jost, A., Kamae, Y., Lohmann, G., Lunt, D.J., Nisancioglu, K.H., Ramstein, G., Abe-Ouchi, A., Riesselman, C.R., Robinson, M.M., Rosenbloom, N.A., Salzmann, U., Stepanek, C., Strother, S.L., Ueda, H., Yan, Q., Zhang, Z., 2013. Sea Surface Temperature of the mid-Piacenzian Ocean: A Data-Model Comparison. Sci. Rep. 3. doi:10.1038/srep02013

Dowsett, H.J., Robinson, M.M., Haywood, A.M., Hill, D.J., Dolan, A.M., Stoll, D.K., Chan, W.-L., Abe-Ouchi, A., Chandler, M.A., Rosenbloom, N.A., Otto-Bliesner, B.L., Bragg, F.J., Lunt, D.J., Foley, K.M., Riesselman, C.R., 2012. Assessing confidence in Pliocene sea surface temperatures to evaluate predictive models. Nature Climate Change 2, 365–371. doi:10.1038/nclimate1455

Haywood, A.M., Dowsett, H.J., Dolan, A.M., Rowley, D.B., Abe-Ouchi, A., Otto-Bliesner, B.L., Chandler, M.A., Hunter, S.J., Lunt, D.J., Pound, M.J., Salzmann, U., 2016. The Pliocene Model Intercomparison Project (PlioMIP) Phase 2: scientific objectives and experimental design. Clim. Past 12, 663–675. doi:10.5194/cp-12-663-2016

Haywood, A.M., Dowsett, H.J., Dolan, A.M., 2016. Integrating geological archives and climate models for the mid-Pliocene warm period. Nature Communications 6, 1–14. doi:10.1038/ncomms10646

Hyde, W.T., Kim, K.-Y., Crowley, T.J., North, G.R., 1990. On the Relation Between Polar Continentality and Climate: Studies With a Nonlinear Seasonal Energy Balance Model. J. Geophys. Res. 95, 18,653–18,668.

Hill, D.J., Haywood, A.M., Lunt, D.J., Hunter, S.J., Bragg, F.J., Contoux, C., Stepanek, C., Sohl, L.E., Rosenbloom, N.A., Chan, W.-L., Kamae, Y., Zhang, Z., Abe-Ouchi, A., Chandler, M.A., Jost, A., Lohmann, G., Otto-Bliesner, B.L., Ramstein, G., Ueda, H., 2014. Evaluating the dominant components of warming in Pliocene climate simulations. Clim. Past 10, 79–90. doi:10.5194/cp-10-79-2014

Lunt, D.J., Haywood, A.M., Schmidt, G.A., Salzmann, U., Valdes, P.J., Dowsett, H.J., Loptson, C.A., 2012. On the causes of mid-Pliocene warmth and polar amplification. Earth and Planetary Science Letters 321–322, 128–138. doi:10.1016/j.epsl.2011.12.042

North, G.R., Mengel, J.G., Short, D.A., 1983. Simple Energy Balance Model Resolving the Seasons and the Continents: Application to the Astronomical Theory of the Ice Ages. J. Geophys. Res. 88, 6576–6586.

Pagani, M., Liu, Z., LaRiviere, J.P., Ravelo, A.C., 2010. High Earth-system climate sensitivity determined from Pliocene carbon dioxide concentrations. Nature Geosci 3, 27–30. doi:10.1038/ngeo724

PALAEOSENS Project Members, 2012. Making sense of palaeoclimate sensitivity. Nature 491, 683–691. doi:10.1038/nature11574

Peltier, W.R., Vettoretti, G., 2014. Dansgaard-Oeschger oscillations predicted in a comprehensive model of glacial climate: A "kicked" salt oscillator in the Atlantic. Geophys. Res. Lett. 41, 7306–7313. doi:10.1002/2014GL061413

Solgaard, A.M., Bonow, J.M., Langen, P.L., Japsen, P., Hvidberg, C.S., 2013. Mountain building and the initiation of the Greenland Ice Sheet. Palaeogeography, Palaeoclimatology, Palaeoecology 392, 161–176. doi:10.1016/j.palaeo.2013.09.019

Rybczynski, N., Gosse, J.C., Harington, C.R., Wogelius, R.A., Hidy, A.J., Buckley, M., 2013. Mid-Pliocene warm-period deposits in the High Arctic yield insight into camel evolution 4, 1550. doi:10.1038/ncomms2516

Vettoretti, G., Peltier, W.R., 2015. Interhemispheric air temperature phase relationships in the nonlinear Dansgaard-Oeschger oscillation. Geophys. Res. Lett. 42, 1–10. doi:10.1002/2014GL062898

Vettoretti, G., Peltier, W.R., 2016. Thermohaline instability and the formation of glacial North Atlantic super polynyas at the onset of Dansgaard-Oeschger warming events. Geophys. Res. Lett. 43, 5336–5344. doi:10.1002/ 2016GL068891

Vettoretti, G., Peltier, W.R., 2018. Fast Physics and Slow Physics in the Nonlinear Dansgaard-Oeschger Relaxation Oscillation. Journal of Climate. 31, 3423–3449. doi:10.1175/JCLI-D-17-0559.1

---

## Author Response (AR1)

**Response to the Editor**

We are very grateful to the editor for the time he has taken to review our manuscript and the interactive discussion. His comments and suggestions have enabled us to further improve the clarity of our manuscript.

> Having looked at the manuscript, I have tried to find what the reviewer has referred to as repetitions. One example could be the fact that the three key questions as listed on lines 30–34, page 3 appear again on lines 23–26, page 28, before re-appearing as the titles of subsections 5.1.1–5.1.3. Perhaps the authors can just skip the listing of the questions again on page 28, writing something to the effect '...so as to address the three key questions posed in the introduction' since the very same questions follow immediately in bold font.

We thank the editor for this helpful comment and we have modified the manuscript as suggested.

> Concerning the introduction, two issues which I have thought long and hard about are the reversal of the climate (the paragraph at the bottom of page 2) and then the part about the ultimate use of stationary boundary conditions (first half of page 4). The authors have gone to great length to justify the inclusion of the whole of those sections in convincing fashion, so I am reluctant to ask for their removal, as the reviewer has suggested. However, I would like to ask the authors to try and somehow simplify /shorten those parts.

We have taken a careful look at the Introduction and eliminated several sentences that we think we can remove safely without affecting the integrity of the overall arguments.

> Reviewer 2 referred to the expression 'these mechanisms explore the changes' on line 4 of the abstract. I, too, could not really understand how a mechanism can 'explore' something, and I think this does need to be changed.

We have replaced the word "explore" with the phrase "allow us to understand the warming in terms of".

> At the beginning of page 2, maybe the authors and reviewer 2 can meet each other halfway and write 'The mid-Pliocene warm period (3.3–3 Mya)', as was done at the start of the introduction in Chandan and Peltier (2017).

In our response to reviewer 2 on this matter, we forgot to mention that we do not have any deep seated preference for indicating the mid-Pliocene as either 3 Mya or 3.3–3 Mya. Therefore we have modified the revised manuscript to indicate the mid-Pliocene as the interval 3.3–3 Mya.

> Line 24, page 1: The values 1370 ppmv and 850 ppmv need to be switched around so that they refer properly to RCP6 and RCP8.5, respectively. Line 15, page 3: delete the comma after the word 'Provided'. Line 23, page 23: Add an 's' and move the comma for better readability: 'But two additional components of the mid-Pliocene orography, (i) that due...'

We are very thankful to the editor for finding these typographical errors. We have fixed all of them in the revised manuscript.

[revised manuscript text omitted]

---

## Editor Decision (ED1)

Many thanks to the reviewers and the authors for their reviews and responses. I think the authors have managed to respond well and meticulously to the technical queries and I do not foresee any problems in that area after the agreed-upon revisions have been made.

Clearly, the second reviewer had some issues with the style of the paper. Repetition, unnecessary parts and the introduction, in particular, seem to be a bone of contention between the reviewer and the authors. Having looked at the manuscript, I have tried to find what the reviewer has referred to as repetitions. One example could be the fact that the three key questions as listed on lines 30-34, page 3 appear again on lines 23-26, page 28, before re-appearing as the titles of subsections 5.1.1-5.1.3. Perhaps the authors can just skip the listing of the questions again on page 28, writing something to the effect '….so as to address the three key questions posed in the introduction' since the very same questions follow immediately in bold font.

Concerning the introduction, two issues which I have thought long and hard about are the reversal of the climate (the paragraph at the bottom of page 2) and then the part about the ultimate use of stationary boundary conditions (first half of page 4). The authors have gone to great length to justify the inclusion of the whole of those sections in convincing fashion, so I am reluctant to ask for their removal, as the reviewer has suggested. However, I would like to ask the authors to try and somehow simplify /shorten those parts.

Reviewer 2 referred to the expression 'these mechanisms explore the changes' on line 4 of the abstract. I, too, could not really understand how a mechanism can 'explore' something, and I think this does need to be changed.

At the beginning of page 2, maybe the authors and reviewer 2 can meet each other halfway and write 'The mid-Pliocene warm period (3.3-3 Mya)', as was done at the start of the introduction in Chandan and Peltier (2017).

As for splitting the last section into separate discussion and conclusion sections (which, admittedly, I am more used to in long papers), I think this is more down to personal taste or style and doing so here would most likely add repetition to an already lengthy paper. Therefore, I am fine with leaving that part as it is, considering that the authors have divided the section clearly into three subsections, each one answering one of their key questions.

I would like to file this manuscript under 'Reconsider after major revisions' so that I can review this manuscript again before the final decision. However, I view these revisions more as 'major tweaks', including the changes the authors themselves have already agreed to make. I look forward to seeing the revised manuscript as it contains a wealth of very interesting results.

In addition, I would like to make some other minor points.

- Line 24, page 1: The values 1370 ppmv and 850 ppmv need to be switched around so that they refer properly to RCP6 and RCP8.5, respectively.
- Line 15, page 3: delete the comma after the word 'Provided'.
- Line 23, page 23: Add an 's' and move the comma for better readability: 'But two additional components of the mid-Pliocene orography, (i) that due…'

---

## Editor Decision (ED2)

**Final revisions recommended to the authors**

Page 3, line 27: for better readability, please insert commas: …..to, for the first time, reasonably……

Page 4, line 5: 'Pliocene' is misspelt.

Page 9, line 7: 'more than one factor' or 'several factors'?

Page18, line 22: 'on its own', ie no apostrophe.

Page 18, line 34: no hyphen in 'in turn'.

Page 25, line 30: no hyphen in 'in turn'.

Page 26, line 15: replace 'and' with a comma.

Page 29, lines 25-27: Remove the numbers.

Page 30, line 29, 'Pliocene' is misspelt.

Page 34, line 2: Remove this citation which is no longer referred to in the latest manuscript.